# THE COMPUTATIONAL COMPLEXITY OF CIRCUIT DISCOVERY FOR INNER INTERPRETABILITY

**Federico Adolfi**
ESI Neuroscience, Max-Planck Society
& University of Bristol
fede.adolfi@bristol.ac.uk

**Martina G. Vilas**
Department of Computer Science
Goethe University Frankfurt
martinagvilas@em.uni-frankfurt.de

**Todd Wareham**
Department of Computer Science
Memorial University of Newfoundland
harold@mun.ca

## ABSTRACT

Many proposed applications of neural networks in machine learning, cognitive/brain science, and society hinge on the feasibility of inner interpretability via circuit discovery. This calls for empirical and theoretical explorations of viable algorithmic options. Despite advances in the design and testing of heuristics, there are concerns about their scalability and faithfulness at a time when we lack understanding of the complexity properties of the problems they are deployed to solve. To address this, we study circuit discovery with classical and parameterized computational complexity theory: (1) we describe a conceptual scaffolding to reason about circuit finding queries in terms of affordances for description, explanation, prediction and control; (2) we formalize a comprehensive set of queries for mechanistic explanation, and propose a formal framework for their analysis; (3) we use it to settle the complexity of many query variants and relaxations of practical interest on multi-layer perceptrons. Our findings reveal a challenging complexity landscape. Many queries are intractable, remain fixed-parameter intractable relative to model/circuit features, and inapproximable under additive, multiplicative, and probabilistic approximation schemes. To navigate this landscape, we prove there exist transformations to tackle some of these hard problems with better-understood heuristics, and prove the tractability or fixed-parameter tractability of more modest queries which retain useful affordances. This framework allows us to understand the scope and limits of interpretability queries, explore viable options, and compare their resource demands on existing and future architectures.

## 1 INTRODUCTION

As artificial neural networks (ANNs) grow in size and capabilities, *Inner Interpretability* — an emerging field tasked with explaining their inner workings (Räuker et al., 2023; Vilas et al., 2024a) — attempts to devise scalable, automated procedures to understand systems mechanistically. Many proposed applications of neural networks in machine learning, cognitive and brain sciences, and society, hinge on the feasibility of inner interpretability. For instance, we might have to rely on interpretability methods to improve system safety (Bereska & Gavves, 2024), detect and control vulnerabilities (García-Carrasco et al., 2024), prune for efficiency (Hooker et al., 2021), find and use task subnetworks (Zhang et al., 2024), explain internal concepts underlying decisions (Lee et al., 2023), experiment with neuro-cognitive models of language, vision, etc. (Lindsay, 2024; Lindsay & Bau, 2023; Pavlick, 2023), describe determinants of ANN-brain alignment (Feghhi et al., 2024; Oota et al., 2023), improve architectures, and extract domain insights (Räuker et al., 2023). We will have to solve different instances of these interpretability problems, ideally automatically, for increasingly large models. We therefore need efficient interpretability procedures, and this requires empirical and theoretical explorations of viable algorithmic options.

**Circuit discovery and its challenges.** Since top-down approaches to inner interpretability (see Vilas et al., 2024a) work their way down from high-level concepts or algorithmic hypotheses (Lieberum et al., 2023), there is interest in a complementary bottom-up methodology: *circuit discovery* (see Shi et al., 2024; Tigges et al., 2024). It starts from neuron- and circuit-level isolation or description (e.g., Hoang-Xuan et al., 2024; Lepori et al., 2023) and attempts to build up higher-level abstractions. The motivation is the *circuit hypothesis*: models might implement their capabilities via small subnetworks (Shi et al., 2024). Advances in the design and testing of interpretability heuristics (see Shi et al., 2024; Tigges et al., 2024) come alongside interest in the automation of circuit discovery (e.g., Conmy et al., 2023; Ferrando & Voita, 2024; Syed et al., 2023) and concerns about its feasibility (Voss et al., 2021; Räuker et al., 2023). One challenge is scaling up methods to larger networks, more naturalistic datasets, and more complex tasks (e.g., Lieberum et al., 2023; Marks et al., 2024), given their manual-intensive search over large spaces (Voss et al., 2021). A related issue is that current heuristics, though sometimes promising (e.g., Merullo et al., 2024), often yield discrepant results (see e.g., Shi et al., 2024; Niu et al., 2023; Zhang & Nanda, 2023). They often find circuits that are not functionally faithful (Yu et al., 2024a) or lack the expected affordances (e.g., effects on behavior; Shi et al., 2024). This questions whether certain localization methods yield results that inform editing (Hase et al., 2023), and vice versa (Wang & Veitch, 2024). More broadly, we run into 'interpretability illusions' (Friedman et al., 2024) when our simplifications (e.g., circuits) mimic the local input-output behavior of the system but lack global *faithfulness* (Jacovi & Goldberg, 2020).

**Exploring viable algorithmic options.** These challenges come at a time when, despite emerging theoretical frameworks (e.g., Vilas et al., 2024a; Geiger et al., 2024), there are notable gaps in the formalization and analysis of the computational problems that interpretability heuristics attempt to solve (see Wang & Veitch, 2024, §8). Issues around scalability of circuit discovery and faithfulness have a natural formulation in the language of Computational Complexity Theory (Arora & Barak, 2009; Downey & Fellows, 2013). A fundamental source of breakdown of scalability — which lack of faithfulness is one manifestation of — is the intrinsic resource demands of interpretability problems. In order to design efficient and effective solutions, we need to understand the complexity properties of circuit discovery queries and the constraints that might be leveraged to yield the desired results. Although experimental efforts have made promising inroads, the complexity-theoretic properties that naturally impact scalability and faithfulness remain open questions (see e.g., Suber-caseaux, 2020, §6C). We settle them here by complementing these efforts with a systematic study of the computational complexity of circuit discovery for inner interpretability. We present a framework that allows us to (a) understand the scope and limits of interpretability queries for description/explanation and prediction/control, (b) explore viable options, and (c) compare their resource demands among existing and future architectures.

## 1.1 CONTRIBUTIONS

- We present a conceptual scaffolding to reason about circuit finding queries in terms of affordances for description, explanation, prediction and control.
- We formalize a comprehensive set of queries that capture mechanistic explanation, and propose a formal framework for their analysis.
- We use this framework to settle the complexity of many query variants, parameterizations, approximation schemes and relaxations of practical interest on multi-layer perceptrons, relevant to various architectures such as transformers.
- We demonstrate how our proof techniques can also be useful to draw links between interpretability and explainability by using them to improve existing results on the latter.

## 1.2 OVERVIEW OF RESULTS

- We uncover a challenging complexity landscape (see Table 4) where many queries are intractable (NP-hard, $\Sigma_2^p$-hard), remain fixed-parameter intractable (W[1]-hard) when constraining model/circuit features (e.g., depth), and are inapproximable under additive, multiplicative, and probabilistic approximation schemes.
- We prove there exist transformations to potentially tackle some hard problems (NP- vs. $\Sigma_2^p$-complete) with better-understood heuristics, and prove the tractability (PTIME) or fixed-parameter tractability (FPT) of other queries of interest, and we identify open problems.
- We describe a quasi-minimality property of ANN circuits and exploit it to generate tractable queries which retain useful affordances as well as efficient algorithms to compute them.
- We establish a separation between local and global query complexity. Together with quasi-minimality, this explains interpretability illusions of faithfulness observed in experiments.

## 1.3 RELATED WORK

This paper gives the first systematic exploration of the computational complexity of *inner interpretability* problems.[1] An adjacent area is the complexity analysis of *explainability* problems (Bassan & Katz, 2023; Ordyniak et al., 2023). It differs from our work in its focus on *input queries* — aspects of the input that explain model decisions — as we look at the inner workings of neural networks via *circuit queries*. Barceló et al. (2020) study the explainability of multi-layer perceptrons compared to simpler models through a set of input queries. Bassan et al. (2024) extend this idea with a comparison between *local* and *global* explainability. None of these works formalize or analyze circuit queries (although Subercaseaux, 2020, identifies it as an open problem); we adapt the local versus global distinction in our framework and show how our proof techniques can tighten some results on explainability queries. Ramaswamy (2019) and Adolfi & van Rooij (2023) explore a small set of circuit queries and only on abstract biological networks modeled as general graphs, which cannot inform circuit discovery in ANNs. Efforts in characterizing the complexity of learning neural networks (e.g., Song et al., 2017; Chen et al., 2020; Livni et al., 2014) might eventually connect to our work, although a number of differences between the formalizations makes results in one area difficult to predict from those in the other. Likewise, efforts to settle the complexity of finding small circuits consistent with a truth table (Hitchcock & Pavan, 2015) are currently too general to be applicable to interpretability problems. More generally, we join efforts to build a solid theoretical foundation for interpretability (Bassan & Katz, 2023; Geiger et al., 2024; Vilas et al., 2024a).

## 2 MECHANISTIC UNDERSTANDING OF NEURAL NETWORKS

Mechanistic understanding is a contentious topic (Ross & Bassett, 2024), but for our purposes it will suffice to adopt a pragmatic perspective. In many cases of practical interest, we want our interpretability methods to output objects that allow us to, in some limited sense, (1) describe or explain succinctly, and (2) control or predict precisely. Such objects (e.g., circuits) should be 'efficiently queriable'; they are often referred to as "a way of making an explanation tractable" (Cao & Yamins, 2023). Roughly, this means that we would like short descriptions (e.g., small circuits) with useful affordances (e.g., to readily answer questions and perform interventions of interest). Circuits have the potential to fulfill these criteria (Olah et al., 2020). Here we preview some special circuits with useful properties which we formalize and analyze later on. Table 1 maps the main circuits we study to their corresponding affordances for description, explanation, prediction and control. Formal definitions of circuit queries are given alongside results in Section 4 (see also Appendix).

Table 1: Circuit affordances for description, explanation, prediction, and control.

| Circuit | Affordance | |
|---|---|---|
| | *Description / Explanation* | *Prediction / Control* |
| Sufficient Circuit | Which neurons suffice in isolation to cause a behavior? *Minimum:* shortest description. | Inference in isolation. *Minimal:* ablating any neuron breaks behavior of the circuit. |
| Quasi-minimal Sufficient Circuit | Which neurons suffice in isolation to cause a behavior and which is a breaking point? | Ablating the breaking point breaks behavior of the circuit. |
| Necessary Circuit | Which neurons are part of all circuits for a behavior? Key subcomputations? | Ablating the neurons breaks behavior of any sufficient circuit in the network. |
| Circuit Ablation & Clamping | Which neurons are necessary in the current configuration of the network? | Ablating/Clamping the neurons breaks behavior of the network. |
| Circuit Robustness | How much redundancy supports a behavior? Resilience to perturbations. | Ablating any set of neurons of size below threshold does not break behavior. |
| Patched Circuit | Which neurons drive a behavior in a given input context, i.e., are control nodes? | Patching neurons changes network behavior for inputs of interest. Steering; Editing. |
| Quasi-minimal Patched Circuit | Which neurons can drive a behavior in a given input context and which neuron is a breaking point? | Patching neurons causes target behavior for inputs of interest; Unpatching breaking point breaks target behavior. |
| Gnostic Neurons | Which neurons respond preferentially to a certain concept? | Concept editing; guided synthesis. |

---

[1]This work expands on FA's PhD dissertation at University of Bristol (Adolfi, 2023; Adolfi et al., 2024).

## 3 INNER INTERPRETABILITY QUERIES AS COMPUTATIONAL PROBLEMS

We model post-hoc interpretability queries on neural networks as computational problems in order to analyze their intrinsic complexity properties. These *circuit queries* also formalize criteria for desired circuits, including those appearing in the literature, such as 'faithfulness', 'completeness', and 'minimality' (Wang et al., 2022; Yu et al., 2024a).

**Query variants: coverage, size and minimality.** The *coverage* of a circuit is the domain over which it behaves in a certain way (e.g., faithful to the model's prediction). *Local* circuits do so over a finite set of known inputs and *global* circuits do so over all possible inputs. The *size* of a circuit is the number of neurons. Some circuit queries require circuits of *bounded* size whereas others leave the size *unbounded*. A circuit with a certain property (e.g., local sufficiency) is *minimal* if there is no subset of its neurons that also has that property (cf. *minimum* size among all such circuits present in the network; see Figure 1).

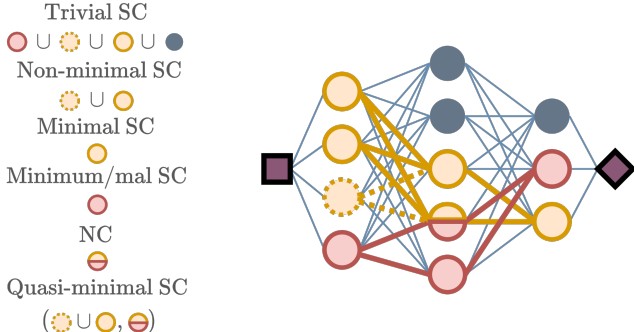

Figure 1: Relationships between circuit types. Sufficient Circuits (SCs) are faithful to the model. The entire network is a trivial SC. Necessary Circuits (NCs) are units shared by all minimal SCs. Quasi-minimal SCs contain a known breaking point (here, NC) and unknown superfluous units.

To fit our comprehensive suite of problems, we explain how to generate problem variants and later on only present one representative definition of each.

**Problem 0.** PROBLEMNAME (PN)
  *Input*:   A multi-layer perceptron $\mathcal{M}$, CoverageIN, SizeIN.
  *Output*:  A Property circuit $\mathcal{C}$ of $\mathcal{M}$, SizeOUT, s.t. CoverageOUT $\mathcal{C}(\mathbf{x}) = \mathcal{M}(\mathbf{x})$, Suffix.

Problem 0 and Table 2 illustrate how to generate problem variants using a template, and ProblemName = SUFFICIENT CIRCUIT as an example (e.g., the Coverage[IN/OUT] variables specify parts of the input/output description that vary according to whether the requested circuit must have global or local faithfulness). Problem definitions will be given for *search* (return specified circuits) or *decision* (answer yes/no circuit queries) versions. Others, including *optimization* (return maximum/minimum-size circuits), can be generated by assigning variables. Problems presented later on are obtained similarly. We also explore various parameterizations, approximation schemes, and relaxations that we explain in the following sections as needed.

Table 2: Generating query variants from problem templates.

| Description variables | Query variants | | | | | |
| --- | --- | --- | --- | --- | --- | --- |
| | *Local* | | | *Global* | | |
| | *Bounded* | *Unbounded* | *Optimal* | *Bounded* | *Unbounded* | *Optimal* |
| CoverageIN | an input $\mathbf{x}$ | an input $\mathbf{x}$ | an input $\mathbf{x}$ | "_" | "_" | "_" |
| CoverageOUT | "_" | "_" | "_" | $\forall_{\mathbf{x}}$ | $\forall_{\mathbf{x}}$ | $\forall_{\mathbf{x}}$ |
| SizeIN | int. $u \leq |\mathcal{M}|$ | "_" | "_" | int. $u \leq |\mathcal{M}|$ | "_" | "_" |
| SizeOUT | size $|\mathcal{C}| \leq u$ | "_" | min. size | size $|\mathcal{C}| \leq u$ | "_" | min. size |
| Property | minimal / "_" | minimal / "_" | "_" | minimal / "_" | minimal / "_" | "_" |
| Suffix | if it exists, otherwise $\perp$ | "_" | "_" | if it exists, otherwise $\perp$ | "_" | "_" |

## 3.1 COMPLEXITY ANALYSES

**Classical and parameterized complexity.** We prove theorems about interpretability queries building on techniques from classical (Garey & Johnson, 1979) and parameterized complexity (Downey & Fellows, 2013). Given our limited knowledge of the problem space of interpretability, *worst-case analysis* is appropriate to explore which problems might be solvable without requiring any additional assumptions (e.g., Bassan et al., 2024; Barceló et al., 2020), and experimental results suggest it captures a lower bound on real-world complexity (e.g., Friedman et al., 2024; Shi et al., 2024; Yu et al., 2024a). Here we give a brief, informal overview of the main concepts underlying our analyses (see Appendix for extensive formal definitions). We will explore beyond classical polynomial-time tractability (PTIME) by studying fixed-parameter tractability (FPT), a more novel and finer-grained look at the *sources of complexity* of problems to test aspects that possibly make interpretability feasible in practice. NP-hard queries are considered intractable because they cannot be computed by polynomial-time algorithms. A relaxation is to allow unreasonable (e.g., exponential) resource demands to be confined to problem parameters that can be kept small in practice. Parameterizing a given ANN and requested circuit leads to parameterized problems (see Table 3 for problem parameters we study later). Parameterized queries in the class FPT admit fixed-parameter tractable algorithms. W-hard queries (by analogy: to FPT as NP-hard is to PTIME), however, do not. We study counting problems via analogous classes #P and #W[1]. We also investigate completeness for NP and classes higher up the polynomial hierarchy such as $\Sigma_2^p$ and $\Pi_2^p$ to identify aspects of hard problems that make them even harder, and to explore the possibility to tackle hard interpretability problems with better-understood methods for well-known NP-complete problems (de Haan & Szeider, 2017). Most proofs involve reductions between computational problems which establish the complexity status of interpretability queries based on the known complexity of canonical problems in other areas.

Table 3: Model and circuit parameterizations.

| Parameter | Model (given) | Circuit (requested) |
|---|---|---|
| Number of layers (depth) | $\hat{L}$ | $\hat{l}$ |
| Maximum layer width | $\hat{L}_w$ | $\hat{l}_w$ |
| Total number of units[2] | $\hat{U} = |\mathcal{M}| \leq \hat{L} \cdot \hat{L}_w$ | $|\mathcal{C}| = \hat{u}$ |
| Number of input units | $\hat{U}_I$ | $\hat{u}_I$ |
| Number of output units | $\hat{U}_O$ | $\hat{u}_O$ |
| Maximum weight | $\hat{W}$ | $\hat{w}$ |
| Maximum bias | $\hat{B}$ | $\hat{b}$ |

**Approximation.** Although sometimes computing optimal solutions is intractable, it is conceivable we could devise tractable interpretability procedures to obtain *approximate* solutions that are useful in practice. We consider 5 notions of approximation: additive, multiplicative, and three probabilistic schemes ($\mathcal{A} = \{c, \text{PTAS}, 3\text{PA}\}$; see Appendix for formal definitions). *Additive approximation* algorithms return solutions at most a fixed distance $c$ away from optimal (e.g., from the minimum-sized circuit), ensuring that errors cannot get impractically large ($c$-approximability). *Multiplicative approximation* returns solutions at most a factor of optimal away. Some hard problems allow for polynomial-time multiplicative approximation schemes (PTAS) where we can get arbitrarily close to optimal solutions as long as we expend increasing compute time (Ausiello et al., 1999). Finally, we consider three types of *probabilistic polynomial-time approximability* (henceforth 3PA) that may be acceptable in situations where always getting the correct output for an input is not required: algorithms that (1) always run in polynomial time and produce the correct output for a given input in all but a small number of cases (Hemaspaandra & Williams, 2012); (2) always run in polynomial time and produce the correct output for a given input with high probability (Motwani & Raghavan, 1995); and (3) run in polynomial time with high probability but are always correct (Gill, 1977).

**Model architecture.** The Multi-Layer Perceptron (MLP) is a natural first step in our exploration because (a) it is proving useful as a stepping stone in current experimental (e.g., Lampinen et al., 2024) and theoretical work (e.g., Rossem & Saxe, 2024; McInerney & Burke, 2023); (b) it exists as a leading standalone architecture (Yu et al., 2024b), as the central element of all-MLP architectures (Tolstikhin et al., 2021), and as a key component of state-of-the-art models such as transformers (Vaswani et al., 2017); (c) it is of active interest to the interpretability community (e.g., Geva et al., 2022; 2021; Dai et al., 2022; Meng et al., 2024; 2022; Niu et al., 2023; Vilas et al., 2024b; Hanna

et al., 2023); and (d) we can relate our findings in *inner interpretability* to those in *explainability*, which also begins with MLPs (e.g., Barceló et al., 2020; Bassan et al., 2024). Although MLP blocks can be taken as units to simplify search, it is recommended to investigate MLPs by treating each neuron as a unit (e.g., Gurnee et al., 2023; Cammarata et al., 2020; Olah et al., 2017), as it better reflects the semantics of computations in ANNs (Lieberum et al., 2023, sec. 2.3.1). We adopt this perspective in our analyses. We write $\mathcal{M}$ for an MLP model and $\mathcal{M}(\mathbf{x})$ for its output on input vector $\mathbf{x}$. Its size $|\mathcal{M}|$ is the number of neurons. A circuit $\mathcal{C}$ is a subset of $|\mathcal{C}|$ neurons which induce a (possibly end-to-end) subgraph of $\mathcal{M}$ (see Appendix for formal definitions).

## 4 RESULTS & DISCUSSION: THE COMPLEXITY OF CIRCUIT QUERIES

In this section we present each circuit query with its computational problem and a discussion of the complexity profile we obtain across variants, relaxations, and parameterizations. For an overview of the results for all queries, see Table 4. Proofs of the theorems can be found in the Appendix.

### 4.1 SUFFICIENT CIRCUIT

Sufficient circuits (SCs) are sets of neurons connected end-to-end that suffice, in isolation, to reproduce some model behavior over an input domain (see *faithfulness*; Wang et al., 2022; Yu et al., 2024a). They are conceptually related to the desired outcome of zero-ablating components that do not contribute to the behavior of interest (small, parameter-efficient subnetworks). Zero-ablation as a method (e.g., to find sufficient circuits) has been criticized on the grounds that the patched value (zero) is somewhat arbitrary and therefore can mischaracterize the functioning of the neuron/circuit when operating in the context of the rest of the network during inference. This gives rise to alternative methods such as activation patching with activation means or specific input activations, which we study later. SCs remain relevant as, despite valid criticisms of zero-ablation (e.g., Conmy et al., 2023), circuit discovery through pruning might be justified at least in some cases (Yu et al., 2024a).

> **Problem 1.** BOUNDED LOCAL SUFFICIENT CIRCUIT (BLSC)
> *Input*: A multi-layer perceptron $\mathcal{M}$, an input vector $\mathbf{x}$, and an integer $u \leq |\mathcal{M}|$.
> *Output*: A circuit $\mathcal{C}$ in $\mathcal{M}$ of size $|\mathcal{C}| \leq u$, such that $\mathcal{C}(\mathbf{x}) = \mathcal{M}(\mathbf{x})$, if it exists, otherwise $\perp$.

We find that many variants of SC are NP-hard (see Table 4). Counterintuitively, this intractability does not depend straightforwardly on parameters such as network depth (W[1]-hard relative to $\mathcal{P}$). Therefore, hardness is not mitigated by keeping models shallow. Given this barrier, we explore the possibility of obtaining approximate solutions but find that hard SC variants are inapproximable relative to all schemes in Section 3.1. An alternative is to consider the membership of these problems in a well-studied class whose solvers are better understood than interpretability heuristics (de Haan & Szeider, 2017). We prove that local versions of SC are NP-complete. This implies there exist efficient transformations from instances of SC to those of the satisfiability problem (SAT; Biere et al., 2021), opening up the possibility to borrow techniques that work reasonably well in practice for SAT that might be suitable for some versions of neural network problems. Interestingly, this is not possible for the global version, which we prove is complete for a class higher up the complexity hierarchy ($\Sigma_2^p$-complete). This result establishes a formal separation between local and global query complexity that partly explains 'interpretability illusions' (Friedman et al., 2024; Yu et al., 2024a), which we conjecture holds for other queries we investigate later. These illusions come about when an interpretability abstraction (e.g., a circuit) seems empirically faithful to its target (e.g., model behavior) by some criterion (e.g., 'local' tests on a dataset), but actually lacks faithfulness in the way it generalizes to other criteria (e.g., tests of its 'global' behavior outside the original distribution).

Next we explore whether we could diagnose if SCs with some desired property (e.g., minimality) are abundant, which would be informative of the ability of heuristic search to stumble upon one of them. We analyze various queries where the output is a count of SCs (i.e., counting problems). We find that both local and global, bounded and unbounded variants are #P-complete and remain intractable (#W[1]-hard) when parameterized by many network features including depth (Table 3).

The hardness profile of SC over all these variants calls for exploring more substantial relaxations. We introduce the notion of *quasi-minimality* for this purpose (similar to Ramaswamy, 2019) and later demonstrate its usefulness beyond this particular problem. Any neuron in a *minimal/minimum* SC is a breaking point in the sense that removing it will break the target behavior. In *quasi-minimal* SCs we are merely guaranteed to know at least one neuron that causes this breakdown. By introducing this relaxation, which gives up some affordances but retains others of interest, we get a feasible interpretability query.

**Problem 2.** UNBOUNDED QUASI-MINIMAL LOCAL SUFFICIENT CIRCUIT (UQLSC)

    *Input*:    A multi-layer perceptron $\mathcal{M}$, and an input vector $\mathbf{x}$.

    *Output*:  A circuit $\mathcal{C}$ in $\mathcal{M}$ and a neuron $v \in \mathcal{C}$ s.t. $\mathcal{C}(\mathbf{x}) = \mathcal{M}(\mathbf{x})$ and $[\mathcal{C} \setminus \{v\}](\mathbf{x}) \neq \mathcal{M}(\mathbf{x})$.

UQLSC is in PTIME. We describe an efficient algorithm to compute it which can be heuristically biased towards finding smaller circuits and combined with techniques that exploit weights and gradients (see Appendix).

## 4.2 GNOSTIC NEURON

Gnostic neurons, sometimes called 'grandmother neurons' in neuroscience (Gale et al., 2020) and 'concept neurons' or 'object detectors' in AI (e.g., Bau et al., 2020), are one of the oldest and still current interpretability queries of interest (see also 'knowledge neurons'; Niu et al., 2023).

**Problem 3.** BOUNDED GNOSTIC NEURONS (BGN)

    *Input*:    A multi-layer perceptron $\mathcal{M}$ and two sets of input vectors $\mathcal{X}$ and $\mathcal{Y}$, an integer $k$, and an activation threshold $t$.

    *Output*:  A set of neurons $V$ in $\mathcal{M}$ of size $|V| \geq k$ such that $\forall_{v \in V}$ it is the case that $\forall_{\mathbf{x} \in \mathcal{X}}$, $\mathcal{M}(\mathbf{x})$ produces activations $A^v_{\mathbf{x}} \geq t$ and $\forall_{\mathbf{y} \in \mathcal{Y}} : A^v_{\mathbf{y}} < t$, if it exists, else $\perp$.

BGN is in PTIME. Alternatives might require GNs to have some behavioral effect when intervened; such variants would remain tractable.

## 4.3 CIRCUIT ABLATION AND CLAMPING

The idea that some neurons perform key subcomputations for certain tasks naturally leads to the hypothesis that ablating them should have downstream effects on the corresponding model behaviors. Searching for neuron sets with this property has been one strategy (i.e., *zero-ablation*) to get at important circuits (Wang & Veitch, 2024). The circuit ablation (CA) problem formalizes this idea.

**Problem 4.** BOUNDED LOCAL CIRCUIT ABLATION (BLCA)

    *Input*:    A multi-layer perceptron $\mathcal{M}$, an input vector $\mathbf{x}$, and an integer $u \leq |\mathcal{M}|$.

    *Output*:  A neuron set $\mathcal{C}$ in $\mathcal{M}$ of size $|\mathcal{C}| \leq u$, s.t. $[\mathcal{M} \setminus \mathcal{C}](\mathbf{x}) \neq \mathcal{M}(\mathbf{x})$, if it exists, else $\perp$.

A difference between CAs and minimal SCs is that the former can be interpreted as a possibly non-minimal breaking set in the context of the whole network whereas the latter is by default a minimal breaking set when the SC is taken in isolation. In this sense, CA can be seen as a less stringent criterion for circuit affordances. A related idea is circuit clamping (CC): fixing the activations of certain neurons to a level that produces a change in the behavior of interest.

**Problem 5.** BOUNDED LOCAL CIRCUIT CLAMPING (BLCC)

    *Input*:    A multi-layer perceptron $\mathcal{M}$, vector $\mathbf{x}$, value $r$, and an integer $u$ s.t. $1 < u \leq |\mathcal{M}|$.

    *Output*:  A subset of neurons $\mathcal{C}$ in $\mathcal{M}$ of size $|\mathcal{C}| \leq u$, such that for the $\mathcal{M}^*$ induced by clamping all $c \in \mathcal{C}$ to value $r$, $\mathcal{M}^*(\mathbf{x}) \neq \mathcal{M}(\mathbf{x})$, if it exists, otherwise $\perp$.

Despite these more modest criteria, we find that both the local and global variants of CA and CC are NP-hard, fixed-parameter intractable W[1]-hard relative to various parameters, and inapproximable in all 5 senses studied. However, we prove these problems are NP-complete, which opens up practical options not available for other problems we study (see remarks in Section 4.1).

## 4.4 CIRCUIT PATCHING

A critique of zero-ablation is the arbitrariness of the value, leading to alternatives such as mean-ablation (e.g., Wang et al., 2022). This contrasts studying circuits in isolation versus embedded in surrounding subnetworks. Activation patching (Ghandeharioun et al., 2024; Zhang & Nanda, 2023; Hanna et al., 2024) and path patching (Goldowsky-Dill et al., 2023) try to pinpoint which activations play an in-context role in model behavior, which inspires the circuit patching (CP) problem.

**Problem 6.** BOUNDED LOCAL CIRCUIT PATCHING (BLCP)

    *Input*:    A multi-layer perceptron $\mathcal{M}$, an integer $k$, an input vector $\mathbf{y}$, and a vector set $\mathcal{X}$.

    *Output*:  A subset $\mathcal{C}$ in $\mathcal{M}$ of size $|\mathcal{C}| \leq k$, such that for the $\mathcal{M}^*$ induced by patching $\mathcal{C}$ with activations from $\mathcal{M}(\mathbf{y})$ and $\mathcal{M} \setminus \mathcal{C}$ with activations from $\mathcal{M}(\mathbf{x})$, $\mathcal{M}^*(\mathbf{x}) = \mathcal{M}(\mathbf{y})$ for all $\mathbf{x} \in \mathcal{X}$, if it exists, otherwise $\perp$.

We find that local/global variants are intractable (NP-hard) in a way that does not depend on parameters such as network depth or size of the patched circuit (W[1]-hard), and are inapproximable ($\{c, \text{PTAS}, 3\text{PA}\}$-inapprox.). Although we also prove the local variant of CP is NP-complete and

therefore approachable in practice with solvers for hard problems not available for the global variants (see remarks in Section 4.1), these complexity barriers motivate exploring further relaxations. With some modifications the idea of *quasi-minimality* can be repurposed to do useful work here.

**Problem 7.** UNBOUNDED QUASI-MINIMAL LOCAL CIRCUIT PATCHING (UQLCP)

*Input*: A multi-layer perceptron $\mathcal{M}$, an input vector $\mathbf{y}$, and a set $\mathcal{X}$ of input vectors.

*Output*: A subset $\mathcal{C}$ in $\mathcal{M}$ and a neuron $v \in \mathcal{C}$, such that for the $\mathcal{M}^*$ induced by patching $\mathcal{C}$ with activations from $\mathcal{M}(\mathbf{y})$ and $\mathcal{M} \setminus \mathcal{C}$ with activations from $\mathcal{M}(\mathbf{x})$, $\forall_{\mathbf{x} \in \mathcal{X}} : \mathcal{M}^*(\mathbf{x}) = \mathcal{M}(\mathbf{y})$, and for $\mathcal{M}'$ induced by patching identically except for $v \in \mathcal{C}$, $\exists_{\mathbf{x} \in \mathcal{X}} : \mathcal{M}'(\mathbf{x}) \neq \mathcal{M}(\mathbf{y})$.

In this way we obtain a tractable query (PTIME) for quasi-minimal patching, sidestepping barriers while retaining some useful affordances (see Table 1). We present an algorithm to compute UQLCP efficiently that can be combined with strategies exploiting weights and gradients (see Appendix).

Table 4: Classical and parameterized complexity results by problem variant.

| Classical & parameterized queries[3] $\mathcal{P} = \mathcal{P}_{\mathcal{M}} \cup \mathcal{P}_{\mathcal{C}}$ $\mathcal{P}_{\mathcal{M}} = \{\hat{L}, \hat{U}_I, \hat{U}_O, \hat{W}, \hat{B}\}$ $\mathcal{P}_{\mathcal{C}} = \{\hat{l}, \hat{l}_w, \hat{u}, \hat{u}_I, \hat{u}_O, \hat{w}, \hat{b}\}$ | Problem variants | | | |
|---|---|---|---|---|
| | Local | | Global | |
| | Decision/Search | Optimization | Decision/Search | Optimization |
| SUFFICIENT CIRCUIT (SC) | NP-complete | $\mathcal{A}$-inapprox. | $\Sigma_2^p$-complete | $\mathcal{A}$-inapprox. |
| $\mathcal{P}$-SC | W[1]-hard | $\mathcal{A}$-inapprox. | W[1]-hard | $\mathcal{A}$-inapprox. |
| Minimal SC | NP-complete | ? | $\in \Sigma_2^p$ ∣ NP-hard | ? |
| $\mathcal{P}$-Minimal SC | W[1]-hard | ? | W[1]-hard | ? |
| Unbounded Minimal SC | ? | N ╱ A | ? | N ╱ A |
| $\mathcal{P}$-Unbounded Minimal SC | ? | | ? | |
| Unbounded Quasi-Minimal SC | PTIME | | ? | |
| Count SC | #P-complete | N ╱ A | #P-hard | N ╱ A |
| $\mathcal{P}$-Count SC | #W[1]-hard | | #W[1]-hard | |
| Count Minimal SC | #P-complete | | #P-hard | |
| $\mathcal{P}$-Count Minimal SC | #W[1]-hard | | #W[1]-hard | |
| Count Unbounded Minimal SC | #P-complete | | #P-hard | |
| GNOSTIC NEURON (GN) | PTIME | N/A | ? | N/A |
| CIRCUIT ABLATION (CA) | NP-complete | $\mathcal{A}$-inapprox. | $\in \Sigma_2^p$ ∣ NP-hard | $\mathcal{A}$-inapprox. |
| $\{\hat{L}, \hat{U}_I, \hat{U}_O, \hat{W}, \hat{B}, \hat{u}\}$-CA | W[1]-hard | $\mathcal{A}$-inapprox. | W[1]-hard | $\mathcal{A}$-inapprox. |
| CIRCUIT CLAMPING (CC) | NP-complete | $\mathcal{A}$-inapprox. | $\in \Sigma_2^p$ ∣ NP-hard | $\mathcal{A}$-inapprox. |
| $\{\hat{L}, \hat{U}_O, \hat{W}, \hat{B}, \hat{u}\}$-CC | W[1]-hard | $\mathcal{A}$-inapprox. | W[1]-hard | $\mathcal{A}$-inapprox. |
| CIRCUIT PATCHING (CP) | NP-complete | $\mathcal{A}$-inapprox. | $\in \Sigma_2^p$ ∣ NP-hard | $\mathcal{A}$-inapprox. |
| $\{\hat{L}, \hat{U}_O, \hat{W}, \hat{B}, \hat{u}\}$-CP | W[2]-hard | $\mathcal{A}$-inapprox. | W[2]-hard | $\mathcal{A}$-inapprox. |
| Unbounded Quasi-Minimal CP | PTIME | N/A | ? | N/A |
| NECESSARY CIRCUIT (NC) | $\in \Sigma_2^p$ ∣NP-hard | $\mathcal{A}$-inapprox. | $\in \Sigma_2^p$ ∣ NP-hard | $\mathcal{A}$-inapprox. |
| $\{\hat{L}, \hat{U}_I, \hat{U}_O, \hat{W}, \hat{u}\}$-NC | W[1]-hard | $\mathcal{A}$-inapprox. | W[1]-hard | $\mathcal{A}$-inapprox. |
| CIRCUIT ROBUSTNESS (CR) | coNP-complete | ? | $\in \Pi_2^p$ ∣ coNP-hard | ? |
| $\{\hat{L}, \hat{U}_I, \hat{U}_O, \hat{W}, \hat{B}, \hat{u}\}$-CR | coW[1]-hard | ? | coW[1]-hard | ? |
| $\{|H|\}$-CR | FPT | FPT | ? | ? |
| $\{|H|, \hat{U}_I\}$-CR | FPT | FPT | FPT | FPT |
| SUFFICIENT REASONS (SR) | $\in \Sigma_2^p$ ∣ NP-hard | 3PA-inapprox. | N ╱ A | |
| $\{\hat{L}, \hat{U}_O, \hat{W}, \hat{B}, \hat{u}\}$-SR | W[1]-hard | 3PA-inapprox. | | |

---

[3]Circuits are bounded-size unless otherwise stated. Each cell contains the complexity of the problem variant in terms of classical and FP (in)tractability, membership in complexity classes, and (in)approximability ($\mathcal{A} = \{c, \mathrm{PTAS}, 3\mathrm{PA}\}$). '?' marks potentially fruitful open problems. 'N/A' stands for not applicable.

## 4.5 NECESSARY CIRCUIT

The criterion of *necessity* is a stringent one, and consequently necessary circuits (NCs) carry powerful affordances (see Table 1). Since neurons in NCs collectively interact with all possible sufficient circuits for a target behavior, they are candidates to describe key task subcomputations and intervening on them is guaranteed to have effects even in the presence of high redundance. This relates to the notion of *circuit overlap* and therefore to efforts in identifying circuits shared by various tasks (e.g., Merullo et al., 2024), and the link between overlap and faithfulness (e.g., Hanna et al., 2024).

**Problem 8.** BOUNDED GLOBAL NECESSARY CIRCUIT (BGNC)

*Input*:  A multi-layer perceptron $\mathcal{M}$, and an integer $k$.

*Output*:  A subset $\mathcal{S}$ of neurons in $\mathcal{M}$ of size $|\mathcal{S}| \leq k$, such that $\mathcal{S} \cap \mathcal{C} \neq \emptyset$ for every circuit $\mathcal{C}$ in $\mathcal{M}$ that is sufficient relative to all possible input vectors, if it exists, otherwise $\bot$.

Unfortunately both local and global versions of NC are NP-hard (in $\Sigma_2^p$; Table 4), remain intractable even when keeping parameters such as network depth, number of input and output neurons, and others small (Table 3), and does not admit any of the available approximation schemes (Section 3.1). Tractable versions of NC are unlikely unless substantial restrictions or relaxations are introduced.

## 4.6 CIRCUIT ROBUSTNESS

A behavior of interest might be over-determined or resilient in the sense that many circuits in the model implement it and one can take over when the other breaks down. This is related to the notion of *redundancy* used in neuroscience (e.g., Nanda et al., 2023). Intuitively, when a model implements a task in this way, the behavior should be more robust to a number of perturbations. The possibility of verifying this property experimentally motivates the circuit robustness (CR) problem, and a related interpretability effort is diagnosing nodes that are excluded by circuit discovery procedures but still have an impact on behavior (false negatives; Kramár et al., 2024).

**Problem 9.** BOUNDED LOCAL CIRCUIT ROBUSTNESS (BLCR)

*Input*:  A multi-layer perceptron $\mathcal{M}$, a subset $H$ of $\mathcal{M}$, an input vector $\mathbf{x}$, and an integer $k$ with $1 \leq k \leq |H|$.

*Output*:  <YES> if for each $H' \subseteq H$, with $|H'| \leq k$, $\mathcal{M}(\mathbf{x}) = [\mathcal{M} \setminus H'](\mathbf{x})$, else <NO>.

We find that Local CR is coNP-complete while Global CR is in $\Pi_2^p$ and coNP-hard. It remains fixed-parameter intractable (coW[1]-hard) relative to model parameters (Table 3). Pushing further, we explore parameterizing CR by $\{|H|\}$ and prove fixed-parameter tractability of $\{|H|\}$-CR which holds both for the local and global versions. There exist algorithms for CR that scale well as long as $|H|$ is reasonable; a scenario that might be useful to probe robustness in practice. This wraps up our results for circuit queries. We briefly digress into *explainability* before discussing some implications.

## 4.7 SUFFICIENT REASONS

Understanding the sufficient reasons (SR) for a model decision in terms of input features consists of knowledge of values of the input components that are enough to determine the output. Given a model decision on an input, the most interesting reasons are those with the least components.

**Problem 10.** BOUNDED LOCAL SUFFICIENT REASONS (BLSR)

*Input*:  A multi-layer perceptron $\mathcal{M}$, an input vector $\mathbf{x}$ of length $|\mathbf{x}| = \hat{u}_I$, and an integer $k$ with $1 \leq k \leq \hat{u}_I$.

*Output*:  A subset $\mathbf{x}^s$ of $\mathbf{x}$ of size $|\mathbf{x}^s| = k$, such that for every possible completion $\mathbf{x}^c$ of $\mathbf{x}^s$ $\mathcal{M}(\mathbf{x^c}) = \mathcal{M}(\mathbf{x})$, if it exists, otherwise $\bot$.

To demonstrate the usefulness of our framework beyond inner interpretability, we show how it links to explainability. Using our techniques for circuit queries, we significantly tighten existing results for SR (Barceló et al., 2020; Wäldchen et al., 2021) by proving that hardness (NP-hard, W[1]-hard, 3PA-inapprox.) holds even when the model has only one hidden layer.

## 5 IMPLICATIONS, LIMITATIONS, AND FUTURE DIRECTIONS

We presented a framework based on parameterized complexity to accompany experiments on inner interpretability with theoretical explorations of viable algorithms. With this grasp of circuit query complexity, we can understand the challenges of scalability and the mixed outcomes of experiments with heuristics for circuit discovery. There is ample complexity-theoretic evidence that there is a limit (often underestimated) to how good the performance of heuristics on intractable problems can

be (Hemaspaandra & Williams, 2012). We can explain 'interpretability illusions' (Friedman et al., 2024) due to lack of faithfulness, minimality (e.g., Shi et al., 2024; Yu et al., 2024a) and other affordances (Wang & Veitch, 2024; Hase et al., 2023), in terms of the kinds of circuits that our current heuristics are well-equipped to discover. For instance, consider the algorithm for automated circuit discovery proposed by Conmy et al. (2023), which eliminates one network component at a time if the consequence on behavior is reasonably small. Since this algorithm runs in polynomial time, it is not likely to solve the problems proven hard here, such as MINIMAL SUFFICIENT CIRCUIT. However, one reason we observe interesting results in some cases is because it is well-equipped to solve QUASI-MINIMAL CIRCUIT problems. As our conceptual and formal analyses show, quasi-minimal circuits can mimic various desirable aspects of sufficient circuits (Table 1), and the former can be found tractably (results for Problem 2 and Problem 7). At the same time, understanding these properties of circuit discovery heuristics helps us explain observed discrepancies: why we often see (1) lack of faithfulness (i.e., global coverage is out of reach for QMC algorithms), (2) non-minimality (i.e., QM circuits can have many non-breaking points), and (3) large variability in performance across tasks and analysis parameters (e.g., Shi et al., 2024; Conmy et al., 2023).

Although we find that many queries of interest are intractable in the general case (and empirical results are in line with this characterization), this should not paralyze efforts to interpret neural network models. As our exploration of the current complexity landscape shows, reasonable relaxations, restrictions and problem variants can yield tractable queries for circuits with useful properties. Consider a few out of many possible avenues to continue these explorations.

**(i) Study query parameters.** Faced with an intractable query, we can investigate which parameters of the problem (e.g., network or circuit aspects) might be responsible for its core hardness. If these problematic parameters can be kept small in real-world applications, this yields a fixed-parameter tractable query. We have explored some, but more are possible as any aspect of the problem can be parameterized. A close dialogue between theorists and experimentalists is important for this, as empirical regularities suggest which parameters might be fruitful to explore theoretically, and experiments test whether theoretically conjectured parameters can be kept small in practice.

**(ii) Generate novel queries.** Our formalization of quasi-minimal circuit problems illustrates the search for viable algorithmic options with examples of tractable problems for inner interpretability. When the use case is well defined, efficient queries that return circuits with useful affordances for applications can be designed. Alternative circuits might also mimic the affordances for prediction/control of ideal circuits while avoiding intractability.

**(iii) Explore network output as axis of approximation.** Some of our constructions use binary input/output (following previous work; e.g., Bassan et al., 2024; Barceló et al., 2020). Although continuous output does not necessarily matter complexity-wise (see Appendix for [counter]examples), this is an interesting direction for future work, as it opens the door to studying the network output as an axis of approximation, which in turn might be a useful relaxation.

**(iv) Design more abstract queries.** A different path is to design queries that partially rely on mid-level abstractions (Vilas et al., 2024a) to bridge the gap between circuits and human-intelligible algorithms (e.g., key-value mechanisms; Geva et al., 2022; Vilas et al., 2024b).

**(v) Characterize actual network structure.** It is in principle possible that some real-world, trained neural networks possess internal structure that is benevolent to general (ideal) circuit queries (e.g., redundancy; see Appendix). In such optimistic scenarios, general-purpose heuristics might work well. The empirical evidence available to date, however, speaks against this. In any case, it will always be important to characterize any such structure to use it explicitly to design algorithms with useful guarantees.

**(vi) Compare resource demands of interpretability/explainability across architectures.** Our results for inner interpretability complement those of explainability (e.g., Barceló et al., 2020; Bassan et al., 2024; Wäldchen et al., 2021). These aspects can be studied together for different architectures to assess their intrinsic interpretability. To some extent our results already transfer to some cases of interest. Since the transformer architecture contains MLPs, the complexity status of our circuit queries bears on neuron-level circuit discovery efforts in transformers (e.g., Large Language/Vision/Audio models).

ACKNOWLEDGMENTS

We thank Ronald de Haan for comments on proving membership using alternating quantifier formulas.

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
