# Appendix: Definitions, Theorems and Proofs

## Table of Contents

# A  PRELIMINARIES

Each section of this appendix is self-contained except for the following definitions. We re-state interpretability query definitions in a more detailed form in each section for convenience. To err here on the side of rigor, here we will use more cumbersome notation that we avoided in the main manuscript for succinctness.

## A.1  COMPUTATIONAL PROBLEMS OF KNOWN COMPLEXITY

Some of our proofs construct reductions from the following computational problems.

CLIQUE (Garey & Johnson, 1979, Problem GT19)
*Input*: An undirected graph $G = (V, E)$ and a positive integer $k$.
*Question*: Does $G$ have a clique of size at least $k$, i.e., a subset $V' \subseteq V, |V'| \geq k$, such that for all pairs $v, v' \in V', (v, v') \in E$?

VERTEX COVER (VC) (Garey & Johnson, 1979, Problem GT1)
*Input*: An undirected graph $G = (V, E)$ and a positive integer $k$.
*Question*: Does $G$ contain a vertex cover of size at most $k$, i.e., a subset $V' \subseteq V, |V'| \leq k$, such that for all $(u, v) \in E$, at least one of $u$ or $v$ is in $V'$?

DOMINATING SET (DS) (Garey & Johnson, 1979, Problem GT2)
*Input*: An undirected graph $G = (V, E)$ and a positive integer $k$.
*Question*: Does $G$ contain a dominating set of size at most $k$, i.e., a subset $V' \subseteq V, |V'| \leq k$, such that for all $v \in V$, either $v \in V'$ or there is at least one $v' \in V'$ such that $(v, v') \in E$?

HITTING SET (HS) (Garey & Johnson, 1979, Problem SP8)
*Input*: A collection of subsets $C$ of a finite set $S$ and a positive integer $k$.
*Question*: Is there a subset $S'$ of $S, |S'| \leq k$, such that $S'$ has a non-empty intersection with each set in $C$?

MINIMUM DNF TAUTOLOGY (3DT) (Schaefer & Umans, 2002, Problem L7)
*Input*: A 3-DNF tautology $\phi$ with $T$ terms over a set of variables $V$ and a positive integer $k$.
*Question*: Is there a 3-DNF formula $\phi'$ made up of $\leq k$ of the terms in $\phi$ that is a also a tautology?

## A.2  CLASSICAL AND PARAMETERIZED COMPLEXITY

**Definition 1** (Polynomial-time tractability). An algorithm is said to run in *polynomial-time* if the number of steps it performs is $O(n^c)$, where $n$ is a measure of the input size and $c$ is some constant. A problem $\Pi$ is said to be *tractable* if it has a *polynomial-time algorithm*. P denotes the class of such problems.

Consider a more fine-grained look at the *sources of complexity* of problems. The following is a relaxation of the notion of tractability, where unreasonable resource demands are allowed as long as they are constrained to a set of problem parameters.

**Definition 2** (Fixed-parameter tractability). Let $\mathcal{P}$ be a set of problem parameters. A problem $\mathcal{P}$-$\Pi$ is *fixed-parameter tractable* relative to $\mathcal{P}$ if there exists an algorithm that computes solutions to instances of $\mathcal{P}$-$\Pi$ of any size $n$ in time $f(\mathcal{P}) \cdot n^c$, where $c$ is a constant and $f(\cdot)$ some computable function. FPT denotes the class of such problems and includes all problems in P.

## A.3  HARDNESS AND REDUCTIONS

Most proof techniques in this work involve reductions between computational problems.

**Definition 3** (Reducibility). A problem $\Pi_1$ is *polynomial-time reducible* to $\Pi_2$ if there exists a polynomial-time algorithm (*reduction*) that transforms instances of $\Pi_1$ into instances of $\Pi_2$ such that solutions for $\Pi_2$ can be transformed in polynomial-time into solutions for $\Pi_1$. This

implies that if a tractable algorithm for $\Pi_2$ exists, it can be used to solve $\Pi_1$ tractably. *Fpt-reductions* transform an instance $(x, k)$ of some problem parameterized by $k$ into an instance $(x', k')$ of another problem, with $k' \leq g(k)$, in time $f(k) \cdot p(|x|)$ where $p$ is a polynomial and $g(\cdot)$ is an arbitrary function. These reductions analogously transfer fixed-parameter tractability results between problems.

Hardness results are generally conditional on two conjectures with extensive theoretical and empirical support. Intractability statements build on these as follows.

**Conjecture 1.**  P $\neq$ NP.

**Definition 4** (Polynomial-time intractability).  The class NP contains all problems in P and more. Assuming Conjecture 1, NP-hard problems lie outside P. These problems are considered *intractable* because they cannot be solved in polynomial-time (unless Conjecture 1 is false; see Fortnow, 2009).

**Conjecture 2.**  FPT $\neq$ W[1].

**Definition 5** (Fixed-parameter intractability).  The class W[1] contains all problems in the class FPT and more. Assuming Conjecture 2, W[1]-hard parameterized problems lie outside FPT. These problems are considered *fixed-parameter intractable*, relative to a given parameter set, because no fixed-parameter tractable algorithm can exist to solve them (unless Conjecture 2 is false; see Downey & Fellows, 2013).

The following two easily-proven lemmas will be useful in our parameterized complexity proofs.

**Lemma 1.**  (Wareham, 1999, Lemma 2.1.30) If problem $\Pi$ is fp-tractable relative to aspect-set $K$ then $\Pi$ is fp-tractable for any aspect-set $K'$ such that $K \subset K'$.

**Lemma 2.**  (Wareham, 1999, Lemma 2.1.31) If problem $\Pi$ is fp-intractable relative to aspect-set $K$ then $\Pi$ is fp-intractable for any aspect-set $K'$ such that $K' \subset K$.

## A.4    APPROXIMATION

Although sometimes computing optimal solutions might be intractable, it is still conceivable that we could devise tractable procedures to obtain *approximate* solutions that are useful in practice. We consider two natural notions of additive and multiplicative approximation and three probabilistic schemes.

### A.4.1    MULTIPLICATIVE APPROXIMATION

For a minimization problem $\Pi$, let $OPT_\Pi(I)$ be an optimal solution for $\Pi$ on instance $I$, $A_\Pi(I)$ be a solution for $\Pi$ returned by an algorithm $A$, and $m(OPT_\Pi(I))$ and $m(A_\Pi(I))$ be the values of these solutions.

**Definition 6** (Multiplicative approximation algorithm).  [Ausiello et al. 1999, Def. 3.5]. Given a minimization problem $\Pi$, an algorithm $A$ is a *multiplicative $\epsilon$-approximation algorithm* for $\Pi$ if for each instance $I$ of $\Pi$, $m(A_\Pi(I)) - m(OPT_\Pi(I)) \leq \epsilon \times m(OPT_\Pi(I))$.

It would be ideal if one could obtain approximate solutions for a problem $\Pi$ that are arbitrarily close to optimal if one is willing to allow extra algorithm runtime.

**Definition 7** (Multiplicative approximation scheme).  [Adapted from Ausiello et al., 1999, Def. 3.10]. Given a minimization problem $\Pi$, a *polynomial-time approximation scheme* (PTAS) for $\Pi$ is a set $\mathcal{A}$ of algorithms such that for each integer $k > 0$, there is a $\frac{1}{k}$-approximation algorithm $A_\Pi^k \in \mathcal{A}$ that runs in time polynomial in $|I|$.

### A.4.2    ADDITIVE APPROXIMATION

It would be useful to have guarantees that an approximation algorithm for our problems returns solutions at most a fixed distance away from optimal. This would ensure errors cannot get impractically large.

**Definition 8** (Additive approximation algorithm).  [Adapted from Ausiello et al., 1999, Def. 3.3]. An algorithm $A_\Pi$ for a problem $\Pi$ is a *d-additive approximation algorithm* (d-AAA)

if there exists a constant $d$ such that for all instances $x$ of $\Pi$ the error between the value $m(\cdot)$ of an optimal solution $optsol(x)$ and the output $A_\Pi(x)$ is such that $\mid m(optsol(x)) - m(A_\Pi(x)) \mid \leq d$.

### A.4.3 PROBABILISTIC APPROXIMATION

Finally, consider three other types of probabilistic polynomial-time approximability (henceforth 3PA) that may be acceptable in situations where always getting the correct output for an input is not required: (1) algorithms that always run in polynomial time and produce the correct output for a given input in all but a small number of cases (Hemaspaandra & Williams, 2012); (2) algorithms that always run in polynomial time and produce the correct output for a given input with high probability (Motwani & Raghavan, 1995); and (3) algorithms that run in polynomial time with high probability but are always correct (Gill, 1977).

### A.5 MODEL ARCHITECTURE

**Definition 9** (Multi-Layer Perceptron). [Adapted from Barceló et al. 2020]. A *multi-layer perceptron* (MLP) is a neural network model $\mathcal{M}$, with $\hat{L}$ layers, defined by sequences of weight matrices $(\mathbf{W}_1, \mathbf{W}_2, \ldots, \mathbf{W}_{\hat{L}})$, $\mathbf{W}_i \in \mathbb{Q}^{d_{i-1} \times d_i}$, bias vectors $(\mathbf{b}_1, \mathbf{b}_2, \ldots, \mathbf{b}_{\hat{L}})$, $\mathbf{b}_i \in \mathbb{Q}^{d_i}$, and (element-wise) ReLU functions $(f_1, f_2, \ldots, f_{\hat{L}-1})$, $f_i(x) := \max(0, x)$. The final function is, without loss of generality, the binary step function $f_{\hat{L}}(x) := 1$ if $x \geq 0$, otherwise $0$. The computation rules for $\mathcal{M}$ are given by $\mathbf{h}_i := f_i(\mathbf{h}_{i-1}\mathbf{W} + \mathbf{b}_i)$, $\mathbf{h}_0 := \mathbf{x}$, where $\mathbf{x}$ is the input. The output of $\mathcal{M}$ on $\mathbf{x}$ is defined as $\mathcal{M}(\mathbf{x}) := \mathbf{h}_{\hat{L}}$. The graph $G_\mathcal{M} = (V, E)$ of $\mathcal{M}$ has a vertex for each component of each $\mathbf{h}_i$. All vertices in layer $i$ are connected by edges to all vertices of layer $i + 1$, with no intra-layer connections. Edges carry weights according to $\mathbf{W}_i$, and vertices carry the components of $\mathbf{b}_i$ as biases. The size of $\mathcal{M}$ is defined as $|\mathcal{M}| := |V|$.

### A.6 PRELIMINARY REMARKS

As is the case for other work in this area ((which might be called "Applied Complexity Theory" Bassan et al., 2024; Barceló et al., 2020), we are not aiming at developing new mathematical techniques but rather deploying existing mathematical tools to answer important questions that connect to applications. Part of the technical challenge we take up is to formalize problems of practical interest in simple (and if possible, elegant) ways that are readily understandable, and to prove their complexity properties efficiently (i.e., obtaining a one to many relation between proof constructions and meaningful results). This allows us to gain insights into the sources of complexity of problems, an investigation where the difficulty/complexity/intricacy of proofs are a liability.

We use 'input queries' to refer to computational problems in *explainability* and 'circuit queries' for circuit discovery in *inner interpretability*. We make no claims as to whether one or the other query relates more to intuitive ideas of explanation or interpretation and merely use the latter as familiar pointers to the literature.

All of our proofs for local problem variants assume a particular input vector I, be it the all-0 or all-1 vector. Note that we can simulate these vectors by having zero weights on the input lines and putting appropriate 0 and 1 biases on the input neurons (a technique developed and used in our later-derived proofs but readily applicable to earlier ones). This causes the input to be 'ignored', which renders our proofs correct under both integer and continuous inputs. Note this construction is in line with previous work (e.g., Bassan et al., 2024; Barceló et al., 2020). Importantly, this highlights that the characteristics of the input are not important but rather there is a combinatorial 'heart' beating at the center of our circuit problems; namely, the selection of a subcircuit from exponential number of subcircuits. This combinatorial core can, if not tamed by appropriate restrictions on network and input structure, give rise to non-polynomial worst-case algorithmic complexity.

Proofs for global problem variants often employ constructions similar to those for local variants, with minor to medium (though crucial) differences. For completeness, the full construction is stated again to minimize errors and the need to check proofs other than those being examined.

**On the issue of real-world structure and formal complexity.** One example scenario where real-world statistics might act as mitigating forces with respect to computational hardness (of the general problems) is the case of high redundancy (related to our Circuit Robustness problem). Redundancy can in some sense make circuit finding easier (as solutions are more abundant), but the benefit comes at a cost for interpretability through introducing identifiably issues. As circuits supporting a particular behavior are more numerous (i.e., there is more redundancy), it might get easier to find them with heuristics, but since they are more numerous, they potentially represent competing explanations, which leads to the issue of identifiability. This redundancy would be important to diagnose and characterize, an issue that our Circuit Robustness problem touches on.

# B    Local and Global Sufficient Circuit

Minimum locally sufficient circuit (MLSC)
*Input*: A multi-layer perceptron $M$ of depth $cd_g$ with $\#n_{tot,g}$ neurons and maximum layer width $cw_g$, connection-value matrices $W_1, W_2, \ldots, W_{cd_g}$, neuron bias vector $B$, a Boolean input vector $I$ of length $\#n_{g,in}$, and integers $d$, $w$, and $\#n$ such that $1 \leq d \leq cd_g$, $1 \leq w \leq cw_g$, and $1 \leq \#n \leq \#n_{tot,g}$.
*Question*: Is there a subcircuit $C$ of $M$ of depth $cd_r \leq d$ with $\#n_{tot,r} \leq \#n$ neurons and maximum layer width $cw_r \leq w$ that produces the same output on input $I$ as $M$?

Minimum globally sufficient circuit (MGSC)
*Input*: A multi-layer perceptron $M$ of depth $cd_g$ with $\#n_{tot,g}$ neurons and maximum layer width $cw_g$, connection-value matrices $W_1, W_2, \ldots, W_{cd_g}$, neuron bias vector $B$, and integers $d$, $w$, and $\#n$ such that $1 \leq d \leq cd_g$, $1 \leq w \leq cw_g$, and $1 \leq \#n \leq \#n_{tot,g}$.
*Question*: Is there a subcircuit $C$ of $M$ of depth $cd_r \leq d$ with $\#n_{tot,r} \leq \#n$ neurons and maximum layer width $cw_r \leq w$ that produces the same output as $M$ on every possible Boolean input vector of length $\#_{in,g}$?

Given a subset $x$ of the neurons in $M$, the subcircuit $C$ of $M$ based on $x$ has the neurons in $x$ and all connections in $M$ among these neurons. Note that in order for the output of $C$ to be equal to the output of $M$ on input $I$, the numbers $\#n_{in,g}$ and $\#n_{out,g}$ of input and output neurons in $M$ must exactly equal the numbers $\#n_{in,r}$ and $\#n_{out,r}$ of input and output neurons in $C$; hence, no input or output neurons can be deleted from $M$ in creating $C$. Following Barceló et al. 2020, page 4, all neurons in $M$ use the ReLU activation function and the output $x$ of each output neuron is stepped as necessary to be Boolean, i.e, $step(x) = 0$ if $x \leq 0$ and is 1 otherwise.

For a graph $G = (V, E)$, we shall assume an ordering on the vertices and edges in $V$ and $E$, respectively. For each vertex $v \in V$, let the complete neighbourhood $N_C(v)$ of $v$ be the set composed of $v$ and the set of all vertices in $G$ that are adjacent to $v$ by a single edge, i.e., $v \cup \{u \mid u \in V \text{ and } (u, v) \in E\}$. Finally, let $VC_B$ be the version of VC in which each vertex in $G$ has degree at most $B$.

We will prove various classical and parameterized results for MLSC and MGSC using reductions from Clique (Theorem 1 and 7). These reductions are summarized in Figure 2 and the parameterized results are proved relative to the parameters in Table 5. Additional reductions from VC and DS (Theorems 5 and 103) use specialized ReLU logic gates described in Barceló et al. 2020, Lemma 13. These gates assume Boolean neuron input and output values of 0 and 1 and are structured as follows:

1. NOT ReLU gate: A ReLU gate with one input connection weight of value $-1$ and a bias of 1. This gate has output 1 if the input is 0 and 0 otherwise.

2. $n$-way AND ReLU gate: A ReLU gate with $n$ input connection weights of value 1 and a bias of $-(n-1)$. This gate has output 1 if all inputs have value 1 and 0 otherwise.

3. $n$-way OR ReLU gate: A combination of an $n$-way AND ReLU gate with NOT ReLU gates on all of its inputs and a NOT ReLU gate on its output that uses DeMorgan's Second Law to implement $(x_1 \vee x_2 \vee \ldots x_n)$ as $\neg(\neg x_1 \wedge \neg x_2 \wedge \ldots \neg x_n)$. This gate has output 1 if any input has value 1 and 0 otherwise.

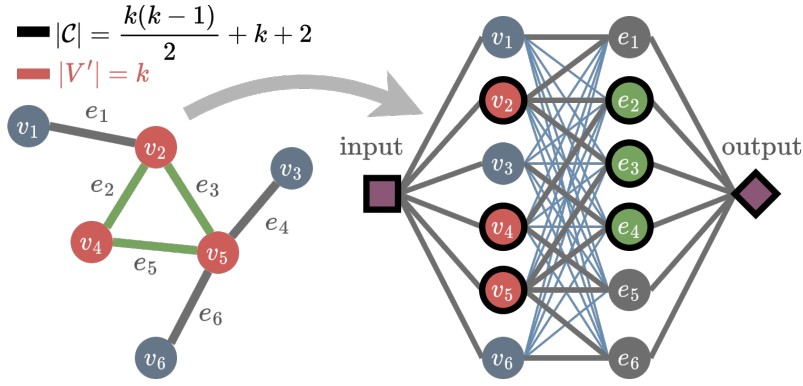

Figure 2: This figure summarizes the reduction from CLIQUE.

Table 5: Parameters for the minimum sufficient subcircuit and reason problems.

| Parameter | Description | Problem |
|---|---|---|
| $cd_g$ | # layers in given MLP | All |
| $cw_g$ | max # neurons in layer in given MLP | All |
| $\#n_{tot,g}$ | total # neurons in given MLP | All |
| $\#n_{in,g}$ | # input neurons in given MLP | All |
| $\#n_{out,g}$ | # output neurons in given MLP | All |
| $B_{\max,g}$ | max neuron bias in given MLP | All |
| $W_{\max,g}$ | max connection weight in given MLP | All |
| $cd_r$ | # layers in requested subcircuit | M{L,G}SC |
| $cw_r$ | max # neurons in layer in requested subcircuit | M{L,G}SC |
| $\#n_{tot,r}$ | total # neurons in requested subcircuit | M{L,G}SC |
| $\#n_{in,r}$ | # input neurons in requested subcircuit | M{L,G}SC |
| $\#n_{out,r}$ | # output neurons in requested subcircuit | M{L,G}SC |
| $B_{\max,r}$ | max neuron bias in requested subcircuit | M{L,G}SC |
| $W_{\max,r}$ | max connection weight in requested subcircuit | M{L,G}SC |
| $k$ | Size of requested subset of input vector | MSR |

## B.1 RESULTS FOR MLSC

Towards proving NP-completeness, we first prove membership and then follow up with hardness. Membership of MLSC in NP can be proven via the definition of the polynomial hierarchy and the following alternating quantifier formula:

$$\exists[\mathcal{C} \subseteq \mathcal{M}] : \mathcal{C}(\mathbf{x}) = \mathcal{M}(\mathbf{x})$$

**Theorem 1.** If MLSC is polynomial-time tractable then $P = NP$.

*Proof.* Consider the following reduction from CLIQUE to MLSC. Given an instance $\langle G = (V, E), k \rangle$ of CLIQUE, construct the following instance $\langle M, I, d, w, \#n \rangle$ of MLSC: Let $M$ be an MLP based on $\#n_{tot,g} = |V| + |E| + 2$ neurons spread across four layers:

1. **Input layer**: The single input neuron $n_{in}$ (bias 0).

2. **Hidden vertex layer**: The vertex neurons $nv_1, nv_2, \ldots nv_{|V|}$ (all with bias 0).

3. **Hidden edge layer**: The edge neurons $ne_1, ne_2, \ldots ne_{|E|}$ (all with bias $-1$).

4. **Output layer**: The single output neuron $n_{out}$ (bias $-(k(k-1)/2 - 1)$).

The non-zero weight connections between adjacent layers are as follows:

- The input neuron $n_{in}$ is connected to each vertex neuron with weight 1.

- Each vertex neuron $nv_i$, $1 \le i \le |V|$, is connected to each edge neuron whose corresponding edge has an endpoint $v_i$ with weight 1.

- Each edge neuron $ne_i$, $1 \le i \le |E|$, is connected to the output neuron $n_{out}$ with weight 1.

All other connections between neurons in adjacent layers have weight 0. Finally, let $I = (1)$, $d = 4$, $w = k(k-1)/2$, and $\#n = k(k-1)/2 + k + 2$. Observe that this instance of MLSC can be created in time polynomial in the size of the given instance of CLIQUE. Moreover, the output behaviour of the neurons in $M$ from the presentation of input $I$ until the output is generated is as follows:

| timestep | neurons (outputs) |
|----------|-------------------|
| 0 | — |
| 1 | $n_{in}(1)$ |
| 2 | $nv_1(1), nv_2(1), \ldots nv_{|V|}(1)$ |
| 3 | $ne_1(1), ne_2(1), \ldots ne_{|E|}(1)$ |
| 4 | $n_{out}(|E| - (k(k-1)/2 - 1))$ |

Note that it is the stepped output of $n_{out}$ in timestep 4 that yields output 1.

We now need to show the correctness of this reduction by proving that the answer for the given instance of CLIQUE is "Yes" if and only if the answer for the constructed instance of MLSC is "Yes". We prove the two directions of this if and only if separately as follows:

$\Rightarrow$ : Let $V' = \{v_1', v_2', \ldots, v_k'\} \subseteq V$ be a clique in $G$ of size $k$. Consider the subcircuit $C$ based on neurons $n_{in}$, $n_{out}$, $\{nv' \mid v' \in V'\}$, and $\{ne' \mid e' = (x, y) \text{ and } vx, vy \in V'\}$. Observe that in this subcircuit, $cd_r = d = 4$, $cw_r = r = k(k-1)/2$, and $\#n_{tot,r} = \#n = k(k-1)/2 + k + 2$. The output behaviour of the neurons in $C$ from the presentation of input $I$ until the output is generated is as follows:

| timestep | neurons (outputs) |
|----------|-------------------|
| 0 | — |
| 1 | $n_{in}(1)$ |
| 2 | $nv_1'(1), nv_2'(1), \ldots nv_{|V'|}'(1)$ |
| 3 | $ne_1'(1), ne_2'(1), \ldots ne_{k(k-1)/2}'(1)$ |
| 4 | $n_{out}(1)$ |

The output of $n_{out}$ in timestep 4 is stepped to 1, which means that $C$ is behaviorally equivalent to $M$ on $I$.

$\Leftarrow$ : Let $C$ be a subcircuit of $M$ that is behaviorally equivalent to $M$ on input $I$ and has $\#n_{tot,r} \le \#n = k(k-1)/2 + k + 2$ neurons. As neurons in all four layers in $M$ must be present in $C$ to produce the required output, $cd_r = d = cd_g$ and both $n_{in}$ and $n_{out}$ are in $C$. In order for $n_{out}$ to produce a non-zero output, there must be at least $k(k-1)/2$ edge neurons in $C$, each of which must be activated by the inclusion of the vertex neurons corresponding to both of their endpoint vertices. This requires the inclusion of at least $k$ vertex neurons in $C$, as a set $V''$ of vertices in graph can have at most $|V''|(|V''| - 1)/2$ distinct edges between them (with this maximum occurring if all pairs of vertices in $V''$ have an edge between them). As $\#n_{tot,r} \le k(k-1)/2 + k + 2$, all of the above implies that there must

be exactly $k(k-1)/2$ edge neurons and exactly $k$ vertex neurons in $C$ and the vertices in $G$ corresponding to these vertex neurons must form a clique of size $k$ in $G$.

As CLIQUE is $NP$-hard (Garey & Johnson, 1979), the reduction above establishes that MLSC is also $NP$-hard. The result follows from the definition of $NP$-hardness. ∎

**Theorem 2.** If $\langle cd_g, \#n_{in,g}, \#n_{out,g}, B_{\max,g}, W_{\max,g}, cd_r, cw_r, \#n_{in,r}, \#n_{out,r}, \#n_{tot,r},$ $B_{\max,r}, W_{\max,r} \rangle$-MLSC is fixed-parameter tractable then $FPT = W[1]$.

*Proof.* Observe that in the instance of MLSC constructed in the reduction in the proof of Theorem 1, $cd_g = cd_r = 4$, $\#n_{in,g} = \#n_{in,r} = \#n_{out,r} = \#n_{out,r} = W_{\max,g} = W_{\max,r} = 1$, and $B_{\max,g}, B_{\max,r}, \#n_{tot,r}$, and $cw_r$ are all functions of $k$ in the given instance of CLIQUE. The result then follows from the fact that $\langle k \rangle$-CLIQUE is $W[1]$-hard (Downey & Fellows, 1999). ∎

**Theorem 3.** $\langle \#n_{tot,g} \rangle$-MLSC is fixed-parameter tractable.

*Proof.* Consider the algorithm that generates each possible subcircuit of $M$ and checks if that sub-circuit is behaviorally equivalent to $M$ on input $I$. If such a subcircuit is found, return "Yes"; otherwise, return "No". As each such subcircuit can be run on $I$ in time polynomial in the size of the given instance of MLSC and the total number of subcircuits that need to be checked is at most $2^{\#n_{tot,g}}$, the above is a fixed-parameter tractable algorithm for MLSC relative to parameter-set $\{\#n_{tot,g}\}$. ∎

**Theorem 4.** $\langle cd_g, cw_g \rangle$-MLSC is fixed-parameter tractable.

*Proof.* Follows from the observation that $\#n_{tot,g} \leq cd_g \times cw_g$ and the algorithm in the proof of Theorem 3. ∎

Though we have already proved the polynomial-time intractability of MLSC in Theorem 1, the re-duction in the proof of the following theorem will be useful in proving a certain type of polynomial-time inapproximability for MLSC (see Figure 3).

**Theorem 5.** If MLSC is polynomial-time tractable then $P = NP$.

*Proof.* Consider the following reduction from VC to MLSC. Given an instance $\langle G = (V, E), k \rangle$ of VC, construct the following instance $\langle M, I, d, w, \#n \rangle$ of MLSC: Let $M$ be an MLP based on $\#n_{tot,g} = |V| + 2|E| + 2$ neurons spread across five layers:

1. **Input layer**: The single input neuron $n_{in}$ (bias 0).

2. **Hidden vertex layer**: The vertex neurons $nvN_1, nvN_2, \ldots nvN_{|V|}$, all of which are NOT ReLU gates.

3. **Hidden edge layer I**: The edge AND neurons $neA_1, neA_2, \ldots neA_{|E|}$, all of which are 2-way AND ReLU gates.

4. **Hidden edge layer II**: The edge NOT neurons $neN_1, neN_2, \ldots neN_{|E|}$, all of which are NOT ReLU gates.

5. **Output layer**: The single output neuron $n_{out}$, which is an $|E|$-way AND ReLU gate.

The non-zero weight connections between adjacent layers are as follows:

- The input neuron $n_{in}$ is connected to each vertex NOT neuron with weight 1.

- Each vertex NOT neuron $nvN_i$, $1 \leq i \leq |V|$, is connected to each edge AND neuron whose corresponding edge has an endpoint $v_i$ with weight 1.

- Each edge AND neuron $neA_i$, $1 \leq i \leq |E|$, is connected to its corresponding edge NOT neuron $neN_i$ with weight 1.

- Each edge NOT neuron $neN_i$, $1 \leq i \leq |E|$, is connected to the output neuron $n_{out}$ with weight 1.

All other connections between neurons in adjacent layers have weight 0. Finally, let $I = (1)$, $d = 5$, $w = |E|$, and $\#n = 2|E| + k + 2$. Observe that this instance of MLSC can be created in time polynomial in the size of the given instance of VC, Moreover, the output behaviour of the neurons in $M$ from the presentation of input $I$ until the output is generated is as follows:

| timestep | neurons (outputs) |
|---|---|
| 0 | — |
| 1 | $n_{in}(1)$ |
| 2 | $nvN_1(0), nvN_2(0), \ldots nvN_{|V|}(0)$ |
| 3 | $neA_1(0), neA_2(0), \ldots neA_{|E|}(0)$ |
| 4 | $neN_1(1), neN_2(1), \ldots neN_{|E|}(1)$ |
| 5 | $n_{out}(1)$ |

We now need to show the correctness of this reduction by proving that the answer for the given instance of VC is "Yes" if and only if the answer for the constructed instance of MLSC is "Yes". We prove the two directions of this if and only if separately as follows:

$\Rightarrow$ : Let $V' = \{v'_1, v'_2, \ldots, v'_k\} \subseteq V$ be a vertex cover in $G$ of size $k$. Consider the subcircuit $C$ based on neurons $n_{in}$, $n_{out}$, $\{nv'N \mid v' \in V'\}$, $\{neA_1, neA_2, \ldots, neA_{|E|}\}$, and $\{neN_1, neN_2, \ldots, neN_{|E|}\}$. Observe that in this subcircuit, $cd_r = d = 5$, $cw_r = w = |E|$, and $\#n_{tot,r} = \#n = 2|E| + k + 2$. The output behaviour of the neurons in $C$ from the presentation of input $I$ until the output is generated is as follows:

| timestep | neurons (outputs) |
|---|---|
| 0 | — |
| 1 | $n_{in}(1)$ |
| 2 | $nvN'_1(0), nvN'_2(0), \ldots nvN'_{|V'|}(0)$ |
| 3 | $neA_1(0), neA_2(0), \ldots neA_{|E|}(0)$ |
| 4 | $neN_1(1), neN_2(1), \ldots neN_{|E|}(1)$ |
| 5 | $n_{out}(1)$ |

This means that $C$ is behaviorally equivalent to $M$ on $I$.

$\Leftarrow$ : Let $C$ be a subcircuit of $M$ that is behaviorally equivalent to $M$ on input $I$ and has $\#n_{tot,r} \leq \#n = 2|E| + k + 2$ neurons. As neurons in all five layers in $M$ must be present in $C$ to produce the required output, $cd_r = cd_g$ and both $n_{in}$ and $n_{out}$ are in $C$. In order for $n_{out}$ to produce a non-zero output, there must be at least $|E|$ edge NOT neurons and $|E|$ AND neurons in $C$, and each of the latter must be connected to at least one of the vertex NOT neurons corresponding to their endpoint vertices. As $\#n_{tot,r} \leq 2|E| + k + 2$, there must be exactly $|E|$ edge NOT neurons, $|E|$ edge AND neurons, and $k$ vertex NOT neurons in $C$ and the vertices in $G$ corresponding to these vertex NOT neurons must form a vertex cover of size $k$ in $G$.

As VC is $NP$-hard (Garey & Johnson, 1979), the reduction above establishes that MLSC is also $NP$-hard. The result follows from the definition of $NP$-hardness. ∎

We now define our two notions of polynomial-time approximation. For a minimization problem $\Pi$, let $OPT_\Pi(I)$ be an optimal solution for $\Pi$, $A_\Pi(I)$ be a solution for $\Pi$ returned by an algorithm $A$, and $m(OPT_\Pi(I))$ and $m(A_\Pi(I))$ be the values of these solutions. Consider the following alternative to approximation algorithms that give solutions that are within an additive factor of optimal.

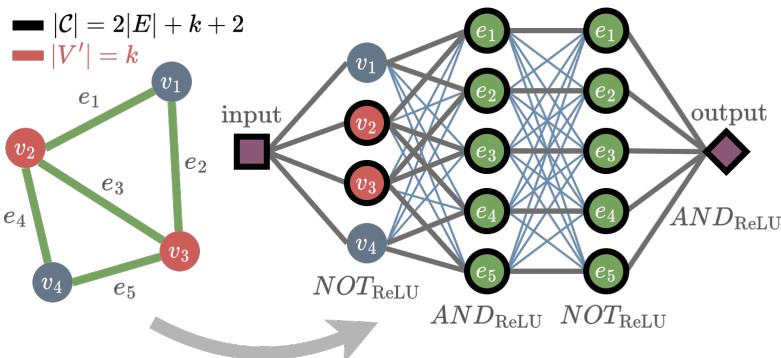

Figure 3: This figure summarizes the reduction from VERTEX COVER

**Definition 10.** (Ausiello et al., 1999, Definition 3.5) Given a minimization problem $\Pi$, an algorithm $A$ is a **(multiplicative) $\epsilon$-approximation algorithm for** $\Pi$ if for each instance $I$ of $\Pi$, $m(A_\Pi(I)) - m(OPT_\Pi(I)) \leq \epsilon \times m(OPT_\Pi(I))$.

It would be ideal if one could obtain approximate solutions for a problem $\Pi$ that are arbitrarily close to optimal if one is willing to allow extra algorithm runtime. This is encoded in the following entity.

**Definition 11.** (Adapted from Definition 3.10 in Ausiello et al. 1999) Given a minimization problem $\Pi$, a **polynomial-time approximation scheme (PTAS) for** $\Pi$ is a set $\mathcal{A}$ of algorithms such that for each integer $k > 0$, there is a $\frac{1}{k}$-approximation algorithm $A_\Pi^k \in \mathcal{A}$ that runs in time polynomial in $|I|$.

The question of whether or not a problem has a PTAS can be answered using the following type of approximation-preserving reducibility.

**Definition 12.** (Papadimitriou & Yannakakis, 1991, page 427) Given two minimization problems $\Pi$ and $\Pi'$, $\Pi$ **L-reduces to** $\Pi'$**, i.e.,** $\Pi \leq_L \Pi'$ if there are polynomial-time algorithms $f$ and $g$ and constants $\alpha, \beta > 0$ such that for each instance $I$ of $\Pi$

(L1) Algorithm $f$ produces an instance $I'$ of $\Pi'$ such that $m(OPT_{\Pi'}(I')) \leq \alpha \times m(OPT_\Pi(I))$; and

(L2) For any solution for $I'$ with value $v'$, algorithm $g$ produces a solution for $I$ of value $v$ such that $v - m(OPT_\Pi(I)) \leq \beta \times (v' - m(OPT_{\Pi'}(I')))$.

**Lemma 3.** (Arora et al., 1998, Theorem 1.2.2) If an optimization problem that is $MAX\ SNP$-hard under L-reductions has a PTAS then $P = NP$.

**Theorem 6.** If MLSC has a PTAS then $P = NP$.

*Proof.* We prove that the reduction from VC to MLSC in the proof of Theorem 5 is also an L-reduction from $VC_B$ to MLSC as follows:

- Observe that $m(OPT_{VC_B}(I)) \geq |E|/B$ (the best case in which $G$ is a collection of $B$-star subgraphs such that each edge is uniquely covered by the central vertex of its associated star) and $m(OPT_{MLSC}(I')) \leq 2|E| + 2|E| + 2 = 4|E| + 2$ (the worst case in which the vertex neurons corresponding to the two endpoints of every edge in $G$ are selected). This gives us

$$
\begin{aligned}
m(OPT_{MLSC}(I')) &\leq & 4Bm(OPT_{VC_B}(I)) + 2 \\
&\leq & 4Bm(OPT_{VC_B}(I)) + 2Bm(OPT_{VC_B}(I)) \\
&\leq & 6Bm(OPT_{VC_B}(I))
\end{aligned}
$$

which satisfies condition L1 with $\alpha = 6B$.

- Observe that any solution $S'$ for for the constructed instance $I'$ of MLSC of value $k + 2|E| + 2$ implies a solution $S$ for the given instance $I$ of $\mathrm{VC}_B$ of size $k$, i.e., the vertices in $V$ corresponding to the selected vertex neurons in $S$. Hence, it is the case that $m(S) - m(OPT_{VC_B}(I)) = m(S') - m(OPT_{MLSC}(I'))$, which satisfies condition L2 with $\beta = 1$.

As $\mathrm{VC}_B$ is $MAX\ SNP$-hard under L-reductions (Papadimitriou & Yannakakis, 1991, Theorem 2(d)), the L-reduction above proves that MLSC is also $MAX\ SNP$-hard under L-reductions. The result follows from Lemma 3. ∎

## B.2 Results for MGSC

**Theorem 7.** If MGSC is polynomial-time tractable then $P = NP$.

*Proof.* Consider the following reduction from CLIQUE to MGSC. Given an instance $\langle G = (V, E), k \rangle$ of CLIQUE, construct an instance $\langle M, d, w, \#n \rangle$ of MGSC as in the reduction in the proof of Theorem 1, omitting input vector $I$. Observe that this instance of MGSC can be created in time polynomial in the size of the given instance of CLIQUE. As $\#n_{in,g} = 1$, there are only two possible Boolean input vectors, $(0)$ and $(1)$. Given input vector $(1)$, as MLP $M$ in this reduction is the same as $M$ in the proof of Theorem 1, the output in timestep 4 is once again 1; moreover, given input vector $(0)$, no vertex or edge neurons can have output 1 and hence the output in timestep 4 is 0.

We now need to show the correctness of this reduction by proving that the answer for the given instance of CLIQUE is "Yes" if and only if the answer for the constructed instance of MGSC is "Yes". We prove the two directions of this if and only if separately as follows:

$\Rightarrow$ : Let $V' = \{v'_1, v'_2, \ldots, v'_k\} \subseteq V$ be a clique in $G$ of size $k$. Consider the subcircuit $C$ based on neurons $n_{in}, n_{out}, \{nv' \mid v' \in V'\}$, and $\{ne' \mid e' = (x, y)$ and $vx, vy \in V'\}$. Observe that in this subcircuit, $cd_r = 4$, $cw_r = k(k-1)/2$, and $\#n_{tot,r} = k(k-1)/2 + k + 2$. Given input $(1)$, the output behaviour of the neurons in $C$ from the presentation of input until the output is generated is as follows:

| timestep | neurons (outputs) |
|:---:|:---|
| 0 | — |
| 1 | $n_{in}(1)$ |
| 2 | $nv'_1(1), nv'_2(1), \ldots nv'_{|V'|}(1)$ |
| 3 | $ne'_1(1), ne'_2(1), \ldots ne'_{k(k-1)/2}(1)$ |
| 4 | $n_{out}(1)$ |

Moreover, given input $(0)$, no vertex or edge neurons in $C$ can have output 1 and the output of $C$ at timestep 4 is 0. This means that $C$ is behaviorally equivalent to $M$ on all possible Boolean input vectors

$\Leftarrow$ : Let $C$ be a subcircuit of $M$ that is behaviorally equivalent to $M$ on all possible Boolean input vectors and has $\#n_{tot,r} \leq \#n = k(k-1)/2 + k + 2$ neurons. Consider the case of input vector $(1)$. This vector must cause $C$ to generate output 1 at timestep 4 as $C$ is behaviorally equivalent to $M$ on all Boolean input vectors. As neurons in all four layers in $M$ must be present in $C$ to produce the required output, $cd_r = cd_g$ and both $n_{in}$ and $n_{out}$ are in $C$. In order for $n_{out}$ to produce a non-zero output, there must be at least $k(k-1)/2$ edge neurons in $C$, each of which must be activated by the inclusion of the vertex neurons corresponding to both of their endpoint vertices. This requires the inclusion of at least $k$ vertex neurons in $C$, as a set $V''$ of vertices in graph can have at most $|V''|(|V''| - 1)/2$ distinct edges between them (with this maximum occurring if all pairs of vertices in $V''$ have an edge between them). As $\#n_{tot,r} \leq k(k-1)/2 + k + 2$, all of the above implies that there must

be exactly $k(k-1)/2$ edge neurons and exactly $k$ vertex neurons in $C$ and the vertices in $G$ corresponding to these vertex neurons must form a clique of size $k$ in $G$.

As CLIQUE is $NP$-hard (Garey & Johnson, 1979), the reduction above establishes that MGSC is also $NP$-hard. The result follows from the definition of $NP$-hardness. ∎

**Theorem 8.** If $\langle cd_g, \#n_{in,g}, \#n_{out,g}, B_{\max,g}, W_{\max,g}, cd_r, cw_r, \#n_{in,r}, \#n_{out,r}, \#n_{tot,r}, B_{\max,r}, W_{\max,r} \rangle$-MGSC is fixed-parameter tractable then $FPT = W[1]$.

*Proof.* Observe that in the instance of MGSC constructed in the reduction in the proof of Theorem 7, $cd_g = cd_r = 4$, $\#n_{in,g} = \#n_{in,r} = \#n_{out,r} = \#n_{out,r} = W_{\max,g} = W_{\max,r} = 1$, and $B_{\max,g}, B_{\max,r}, \#n_{tot,r}$, and $cw_r$ are all functions of $k$ in the given instance of CLIQUE. The result then follows from the fact that $\langle k \rangle$-CLIQUE is $W[1]$-hard (Downey & Fellows, 1999). ∎

**Theorem 9.** $\langle \#n_{tot,g} \rangle$-MGSC is fixed-parameter tractable.

*Proof.* Consider the algorithm that generates each possible subcircuit of $M$ and checks if that subcircuit is behaviorally equivalent to $M$ on all possible Boolean input vectors of length $\#n_{in,g}$. If such a subcircuit is found, return "Yes"; otherwise, return "No". There are $2^{\#n_{in,g}}$ possible Boolean input vectors and the total number of subcircuits that need to be checked is at most $2^{\#n_{tot,g}}$. As $\#n_{in,g} \leq \#n_{tot,g}$ and each such subcircuit can be run on an input vector in time polynomial in the size of the given instance of MGSC, the above is a fixed-parameter tractable algorithm for MGSC relative to parameter-set $\{\#n_{tot,g}\}$. ∎

**Theorem 10.** $\langle cd_g, cw_g \rangle$-MGSC is fixed-parameter tractable.

*Proof.* Follows from the observation that $\#n_{tot,g} \leq cd_g \times cw_g$ and the algorithm in the proof of Theorem 9. ∎

Let us now consider the PTAS-approximability of MGSC. A first thought would be to -reuse the reduction in the proof of Theorem 5 if the given MLP $M$ and VC subcircuit are behaviorally equivalent under both possible input vectors, $(1)$ and $(0)$. We already know the former is true. With respect to the latter, observe that the output behaviour of the neurons in $M$ from the presentation of input $(0)$ until the output is generated is as follows:

| timestep | neurons (outputs) |
|----------|-------------------|
| 0 | — |
| 1 | $n_{in}(0)$ |
| 2 | $nvN_1(1), nvN_2(1), \ldots nvN_{|V|}(1)$ |
| 3 | $neA_1(1), neA_2(1), \ldots neA_{|E|}(1)$ |
| 4 | $neN_1(0), neN_2(0), \ldots neN_{|E|}(0)$ |
| 5 | $n_{out}(0)$ |

However, in the VC subcircuit, we are no longer guaranteed that both endpoint vertex NOT neurons for any edge AND neuron (let alone the endpoint vertex NOT neurons for all edge AND neurons) will be the vertex cover encoded in the subcircuit. This means that all edge AND neurons could potentially output 0, which would cause $M$ to output 1 at timestep 4.

This problem can be fixed if we can modify the given VC graph $G$ to create a graph $G'$ such that

1. we can guarantee that both endpoint vertex NOT neurons for at least one edge AND neuron are present in a VC subcircuit $C$ constructed for $G'$ (which would make at least one edge AND neuron output 1 and cause $C$ to output 0 at timestep 4); and

2. we can easily extract a graph vertex cover of size at most $k$ for $G$ from any vertex cover of a particular size for $G'$.

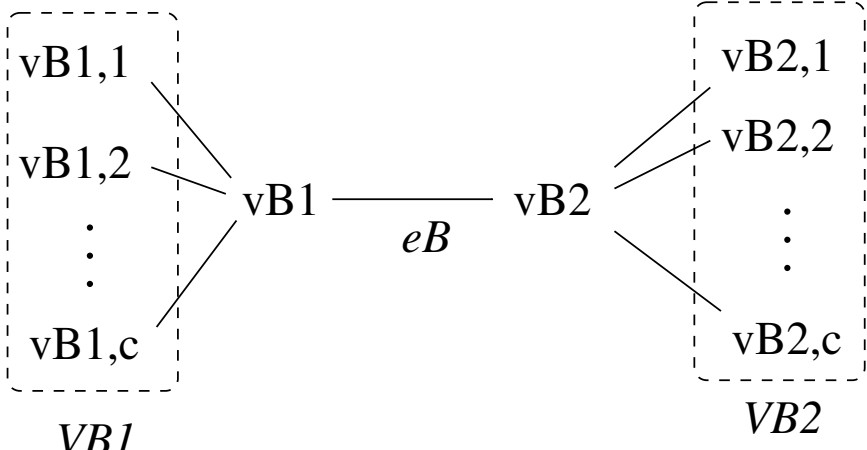

Figure 4: The $c$-way Bowtie Graph $B_c$.

To do this, we shall use the $c$-**way bowtie graph** $B_c$. For $c > 0$, $B_c$ consists of a central edge $e_B$ between vertices $v_{B1}$ and $v_{B2}$ such that $c$ edges radiate outwards from $v_{B1}$ and $v_{B2}$ to the $c$-sized vertex-sets $V_{B1} = \{v_{B1,1}, v_{B1,2}, \ldots, v_{B1,c}\}$ and $V_{B2} = \{v_{B2,1}, v_{B2,2}, \ldots, v_{B2,c}\}$, respectively (see Figure 4). Note that such a graph has $2c+2$ vertices and $2c+1$ edges. Given a graph $G = (V, E)$ with no isolated vertices such that $|V| \leq 2|E|$ (with the minimum occurring in a graph consisting of $|E|$ endpoint-disjoint edges), let $Bow(G) = B_{4|E|} \cup G$. This graph has the following useful property.

**Lemma 4.** Given a graph $G = (V, E)$ and a positive integer $k \leq |V|$, if $Bow(G)$ has a vertex cover $V'$ of size at most $k + 2$ then (1) $\{v_{B1}, v_{B2}\} \in V'$ and (2) $G$ has a vertex cover of size at most $k$.

*Proof.* Let us prove the two consequent clauses as follows:

1. Any vertex cover $V'$ of $Bow(G)$ must cover all the edges in both $B_{4|E|}$ and $G$. Suppose $\{v_{B1}, v_{B2}\} \notin V'$. In order to cover the edges in $B_{4|E|}$, all $8|E|$ vertices in $V_{B1} \cup V_{B2}$ must be in $V'$. This is however impossible as $|V'| \leq k+2 \leq |V|+2 \leq 2|E|+2 \leq 4|E| < 8|E|$. Similarly, suppose only one of $v_{B1}$ and $v_{B2}$ is in $V'$; let us assume it is $v_{B1}$. In that case, all vertices in $V_{B2}$ must be in $V'$. However, this too is impossible as $|V' - \{v_{B1}\}| \leq k+1 \leq |V| + 1 \leq 2|E| + 1 \leq 3|E| < 4|E| = |V_{B2}|$. Hence, both $v_{B1}$ and $v_{B2}$ must be in $V'$.

2. Given (1), $k' \leq k$ vertices remain in $V'$ to cover $G$. All $k'$ of these vertices need not be in $G$, e.g., some may be scattered over $V_{B1}$ and $V_{B2}$. That being said, it is still the case that $G$ must have a vertex cover of size at most $k$.

This concludes the proof. ∎

**Theorem 11.** If MGSC is polynomial-time tractable then $P = NP$.

*Proof.* Consider the following reduction from VC to MGSC. Given an instance $\langle G = (V, E), k \rangle$ of VC, construct the following instance $\langle M, d, w, \#n \rangle$ of MGSC based on $G' = (V', E') = Bow(G)$: Let $M$ be an MLP based on $\#n_{tot,g} = |V'| + 2|E'| + 2$ neurons spread across five layers:

1. **Input layer**: The single input neuron $n_{in}$ (bias 0).

2. **Hidden vertex layer**: The vertex neurons $nvN_1, nvN_2, \ldots nvN_{|V'|}$, all of which are NOT ReLU gates.

3. **Hidden edge layer I**: The edge AND neurons $neA_1, neA_2, \ldots neA_{|E'|}$, all of which are 2-way AND ReLU gates.

4. **Hidden edge layer II**: The edge NOT neurons $neN_1, neN_2, \ldots neN_{|E'|}$, all of which are NOT ReLU gates.

5. **Output layer**: The single output neuron $n_{out}$, which is an $|E'|$-way AND ReLU gate.

The non-zero weight connections between adjacent layers are as follows:

- The input neuron $n_{in}$ is connected to each vertex NOT neuron with weight 1.

- Each vertex NOT neuron $nvN_i$, $1 \leq i \leq |V'|$, is connected to each edge AND neuron whose corresponding edge has an endpoint $v'_i$ with weight 1.

- Each edge AND neuron $neA_i$, $1 \leq i \leq |E'|$, is connected to its corresponding edge NOT neuron $neN_i$ with weight 1.

- Each edge NOT neuron $neN_i$, $1 \leq i \leq |E'|$, is connected to the output neuron $n_{out}$ with weight 1.

All other connections between neurons in adjacent layers have weight 0. Finally, let $d = 5$, $w = |E'|$, and $\#n = 2|E'| + (k+2) + 2 = 2|E'| = k + 4$. Observe that this instance of MGSC can be created in time polynomial in the size of the given instance of VC, the output behaviour of the neurons in $M$ from the presentation of input $(1)$ until the output is generated is as follows:

| timestep | neurons (outputs) |
|----------|-------------------|
| 0 | — |
| 1 | $n_{in}(1)$ |
| 2 | $nvN_1(0), nvN_2(0), \ldots nvN_{|V'|}(0)$ |
| 3 | $neA_1(0), neA_2(0), \ldots neA_{|E'|}(0)$ |
| 4 | $neN_1(1), neN_2(1), \ldots neN_{|E'|}(1)$ |
| 5 | $n_{out}(1)$ |

and the output behaviour of the neurons in $M$ from the presentation of input $(0)$ until the output is generated is as follows:

| timestep | neurons (outputs) |
|----------|-------------------|
| 0 | — |
| 1 | $n_{in}(0)$ |
| 2 | $nvN_1(1), nvN_2(1), \ldots nvN_{|V'|}(1)$ |
| 3 | $neA_1(1), neA_2(1), \ldots neA_{|E'|}(1)$ |
| 4 | $neN_1(0), neN_2(0), \ldots neN_{|E'|}(0)$ |
| 5 | $n_{out}(0)$ |

We now need to show the correctness of this reduction by proving that the answer for the given instance of VC is "Yes" if and only if the answer for the constructed instance of MGSC is "Yes". We prove the two directions of this if and only if separately as follows:

$\Rightarrow$ : Let $V'' = \{v''_1, v''_2, \ldots, v''_k\} \subseteq V$ be a vertex cover in $G$ of size $k$. Consider the subcircuit $C$ based on neurons $n_{in}$, $n_{out}$, $\{nv''N \mid v'' \in V''\} \cup \{nv_{B1}N, nv_{B2}B\}$, $\{neA_1, neA_2, \ldots, neA_{|E'|}\}$, and $\{neN_1, neN_2, \ldots, neN_{|E'|}\}$. Observe that in this subcircuit, $cd_r = d = 5$, $cw_r = w = |E'|$, and $\#n_{tot,r} = \#n = 2|E'| + k + 4$. The output behaviour of the neurons in $C$ from the presentation of input $(1)$ until the output is generated is as follows:

| timestep | neurons (outputs) |
|----------|-------------------|
| 0 | — |
| 1 | $n_{in}(1)$ |
| 2 | All vertex NOT neurons (0) |
| 3 | $neA_1(0), neA_2(0), \ldots neA_{|E'|}(0)$ |
| 4 | $neN_1(1), neN_2(1), \ldots neN_{|E'|}(1)$ |
| 5 | $n_{out}(1)$ |

Moreover, the output behaviour of the neurons in $C$ from the presentation of input $(0)$ until the output is generated is as follows:

| timestep | neurons (outputs) |
|----------|-------------------|
| 0 | — |
| 1 | $n_{in}(0)$ |
| 2 | All vertex NOT neurons (1) |
| 3 | At least one edge AND neuron has output 1, e.g., $ne_BA$ |
| 4 | At least one edge NOT neuron has output 0, e.g., $ne_BN$ |
| 5 | $n_{out}(0)$ |

This means that $C$ is behaviorally equivalent to $M$ on all possible Boolean input vectors.

$\Leftarrow$ : Let $C$ be a subcircuit of $M$ that is behaviorally equivalent to $M$ on all possible Boolean input vectors and has $\#n_{tot,r} \leq \#n = 2|E'| + k + 4$ neurons. As neurons in all five layers in $M$ must be present in $C$ to produce the required output, $cd_r = cd_g$ and both $n_{in}$ and $n_{out}$ are in $C$. In order for $n_{out}$ to produce a non-zero output, there must be at least $|E'|$ edge NOT neurons and $|E'|$ AND neurons in $C$, and each of the latter must be connected to at least one of the vertex NOT neurons corresponding to their endpoint vertices. As $\#n_{tot,r} \leq 2|E'| + k + 4$, there must be exactly $|E|$ edge NOT neurons, $|E'|$ edge AND neurons, and $k + 2$ vertex NOT neurons in $C$ and the vertices in $G'$ corresponding to these vertex NOT neurons must form a vertex cover $V''$ of size $k + 2$ in $G'$. However, as $G' = Bow(G)$, Lemma 4 implies not only that $\{nv_{B1}N, nv_{B2}N\} \in V''$ but that $G$ has a vertex cover of size at most $k$.

As VC is $NP$-hard (Garey & Johnson, 1979), the reduction above establishes that MGSC is also $NP$-hard. The result follows from the definition of $NP$-hardness. ∎

**Theorem 12.** If MGSC has a PTAS then $P = NP$.

*Proof.* We prove that the reduction from VC to MLSC in the proof of Theorem 11 is also an L-reduction from $VC_B$ to MGSC as follows:

- Observe that $m(OPT_{VC_B}(I)) \geq |E|/B$ (the best case in which $G$ is a collection of $B$-star subgraphs such that each edge is uniquely covered by the central vertex of its associated star) and $m(OPT_{MGSC}(I')) \leq 2|E'| + 2|E'| + 2 = 4|E'| + 2$ (the worst case in which the vertex neurons corresponding to the two endpoints of every edge in $G$ are selected). As $|E'| = 9|E| + 1$, this gives us

$$
\begin{aligned}
m(OPT_{MLSC}(I')) &\leq 36|E| + 4 + 2 \\
&\leq 36Bm(OPT_{VC_B}(I)) + 6 \\
&\leq 36Bm(OPT_{VC_B}(I)) + 6Bm(OPT_{VC_B}(I)) \\
&\leq 42Bm(OPT_{VC_B}(I))
\end{aligned}
$$

which satisfies condition L1 with $\alpha = 42B$.

- Observe that any solution $S'$ for for the constructed instance $I'$ of MGSC of value $k + 2|E'| + 4$ implies a solution $S$ for the given instance $I$ of $\text{VC}_B$ of size $k$ in $S$. Hence, it is the case that $m(S) - m(OPT_{VC_B}(I)) = m(S') - m(OPT_{MGSC}(I'))$, which satisfies condition L2 with $\beta = 1$.

As $\text{VC}_B$ is $MAX\ SNP$-hard under L-reductions (Papadimitriou & Yannakakis, 1991, Theorem 2(d)), the L-reduction above proves that MGSC is also $MAX\ SNP$-hard under L-reductions. The result follows from Lemma 3. ∎

## C  SUFFICIENT CIRCUIT SEARCH AND COUNTING PROBLEMS

**Definition 13.** An entity $x$ with property $P$ is *minimal* if there is no non-empty subset of elements in $x$ that can be deleted to create an entity $x'$ with property $P$.

We shall assume here that all subcircuits are non-trivial, i.e., the subcircuit is of size $< |M|$.

Consider the following search problem templates:

**Name** LOCAL SUFFICIENT CIRCUIT (**AccLSC**)
*Input*: A multi-layer perceptron $M$ of depth $cd_g$ with $\#n_{tot,g}$ neurons and maximum layer width $cw_g$, connection-value matrices $W_1, W_2, \ldots, W_{cd_g}$, neuron bias vector $B$, a Boolean input vector $I$ of length $\#n_{g,in}$, integers $d$ and $w$ such that $1 \le d \le cd_g$ and $1 \le w \le cw_g$ **PrmAdd**.
*Output*: A **CType** subcircuit $C$ of $M$ of depth $cd_r \le d$ with maximum layer width $cw_r \le w$ that $C(I) = M(I)$, if such a subcircuit exists, and special symbol $\perp$ otherwise.

**Name** GLOBAL SUFFICIENT CIRCUIT (**AccGSC**)
*Input*: A multi-layer perceptron $M$ of depth $cd_g$ with $\#n_{tot,g}$ neurons and maximum layer width $cw_g$, connection-value matrices $W_1, W_2, \ldots, W_{cd_g}$, neuron bias vector $B$, integers $d$ and $w$ such that $1 \le d \le cd_g$ and $1 \le w \le cw_g$ **PrmAdd**.
*Output*: A **CType** subcircuit $C$ of $M$ of depth $cd_r \le d$ with maximum layer width $cw_r \le w$ such that $C(I) = M(I)$ for every possible Boolean input vector $I$ of length $\#_{in,g}$, if such a subcircuit exists, and special symbol $\perp$ otherwise..

**Name** LOCAL NECESSARY CIRCUIT (**AccLNC**)
*Input*: A multi-layer perceptron $M$ of depth $cd$ with $\#n_{tot}$ neurons and maximum layer width $cw$, connection-value matrices $W_1, W_2, \ldots, W_{cd}$, neuron bias vector $B$, a Boolean input vector $I$ of length $\#n_{in}$ **PrmAdd**.
*Output*: A **CType** subcircuit $C$ of $M$ such that $C \cap C' \ne \emptyset$ for every sufficient circuit $C'$ of $M$ relative to $I$, if such a s subcircuit exists, and special symbol $\perp$ otherwise.

**Name** GLOBAL NECESSARY CIRCUIT (**AccGNC**)
*Input*: A multi-layer perceptron $M$ of depth $cd$ with $\#n_{tot}$ neurons and maximum layer width $cw$, connection-value matrices $W_1, W_2, \ldots, W_{cd}$, neuron bias vector $B$ **PrmAdd**.
*Output*: A **CType** subcircuit $C$ of $M$ such that for for every possible Boolean input vector $I$ of length $\#n_{in}$, $C \cap C' \ne \emptyset$ for every sufficient circuit $C'$ of $M$ relative to $I$, if such a subcircuit exists, and special symbol $\perp$ otherwise.

**Name** LOCAL CIRCUIT ABLATION (**AccLCA**)
*Input*: A multi-layer perceptron $M$ of depth $cd$ with $\#n_{tot}$ neurons and maximum layer width $cw$, connection-value matrices $W_1, W_2, \ldots, W_{cd}$, bias vector $B$, a Boolean input vector $I$ of length $\#n_{in}$ **PrmAdd**.
*Output*: A **CType** subcircuit $C$ of $M$ such that $(M/C)(I) \ne M(I)$, if such a subcircuit exists, and special symbol $\perp$ otherwise.

**Name** GLOBAL CIRCUIT ABLATION (**AccGCA**)
*Input*: A multi-layer perceptron $M$ of depth $cd$ with $\#n_{tot}$ neurons and maximum layer width $cw$,

connection-value matrices $W_1, W_2, \ldots, W_{cd}$, neuron bias vector $B$ **PrmAdd**.
*Output*: A **CType** subcircuit $C$ of $M$ such that $(M/C)(I) \neq M(I)$ every possible Boolean input vector $I$ of length $\#n_{in}$, if such a subcircuit exists, and special symbol $\perp$ otherwise.

Each these templates can be filled out to create six search problem variants as follows:

1. Exact-size Problem:

   - **Name** = "Exact"
   - **Acc** = "Ex"
   - **PrmAdd** = ", and a positive integer $k < |M|$"
   - **CType** = "size-$k$"

2. Bounded-size Problem:

   - **Name** = "Bounded"
   - **Acc** = "B"
   - **PrmAdd** = ", and a positive integer $k < |M|$"
   - **CType** = "size-$\leq k$"

3. Minimal Problem:

   - **Name** = "Minimal"
   - **Acc** = "Mnl"
   - **PrmAdd** = ""
   - **CType** = "minimal"

4. Minimal Exact-size Problem:

   - **Name** = "Minimal exact"
   - **Acc** = "MnlEx"
   - **PrmAdd** = ", and a positive integer $k < |M|$"
   - **CType** = "minimal size-$k$"

5. Minimal Bounded-size Problem:

   - **Name** = "Minimal bounded"
   - **Acc** = "MnlB"
   - **PrmAdd** = ", and a positive integer $k < |M|$"
   - **CType** = "minimal size-$\leq k$"

6. Minimum Problem:

   - **Name** = "Minimum"
   - **Acc** = "Min"
   - **PrmAdd** = ""
   - **CType** = "minimum"

We will use previous results for the following problems to prove our results for the problems above.

EXACT CLIQUE (ExClique)
*Input*: An undirected graph $G = (V, E)$ and a positive integer $k \leq |V|$.
*Output*: A $k$-size vertex subset $V'$ of $G$ that is a clique of size $k$ in $G$, if such a $V'$ exists, and special symbol $\perp$ otherwise.

EXACT VERTEX COVER (ExVC)
*Input*: An undirected graph $G = (V, E)$ and a positive integer $k \leq |V|$.

Table 6: Parameters for the sufficient circuit, necessary circuit, circuit ablation problems. Note that for sufficient circuit problems, there are two versions of each parameter $k$ – namely, those describing the given MLP $M$ and the derived subcircuit $C$ (distinguished by $g$- ands $r$-subscripts, respectively).

| Parameter | Description |
|---|---|
| $cd$ | # layers in given MLP |
| $cw$ | max # neurons in layer in given MLP |
| $\#n_{tot}$ | total # neurons in given MLP |
| $\#n_{in}$ | # input neurons in given MLP |
| $\#n_{out}$ | # output neurons in given MLP |
| $B_{\max}$ | max neuron bias in given MLP |
| $W_{\max}$ | max connection weight in given MLP |
| $k$ | Size of requested neuron subset |

*Output*: A $k$-size vertex subset $V'$ of $G$ that is a vertex cover of size $k$ in $G$, if such a $V'$ exists, and special symbol $\perp$ otherwise.

MINIMAL VERTEX COVER (MnlVC) (Valiant, 1979, Problem 4)
*Input*: An undirected graph $G = (V, E)$.
*Output*: A minimal subset $V'$ of the vertices in $G$ that is a vertex cover of $G$.

Given any search problem **X** above, let **#X** be the problem that returns the number of solution outputs. To assess the complexity of these counting problems, we will use the following definitions in Garey & Johnson 1979, Section 7.3, adapted from those originally given in Valiant 1979.

> **Definition 14.** (Garey & Johnson, 1979, p. 168) A counting problem #Π is in #P if there is a nondeterministic algorithm such that for each input $I$ of Π, (1) the number of 'distinct "guesses" that lead to acceptance of $I$ exactly equals the number solutions of Π for input $I$ and (2) the length of the longest accepting computation is bounded by a polynomial in $|I|$.

#P contains very hard problems, as it is known every class in the Polynomial Hierarchy (which include $P$ and $NP$ as its lowest members) Turing reduces to #P, i.e., $PH \subseteq P^{\#P}$ (Toda, 1991). We use the following type of reduction to isolate problems that are the hardest (#P-complete) and at least as hard as the hardest (#P-hard) in #P.

> **Definition 15.** (Garey & Johnson, 1979, p. 168-169) Given two search problems Π and Π', a *(polynomial time) parsimonious reduction from Π to Π'* is a function $f : I_\Pi \to I_{\Pi'}$ that can be computed in polynomial time such that for every $I \in \Pi$, the number of solutions of Π for input $I$ is exactly equal to the number of Π' for input $f(I)$.

We will also derive parameterized counting results using the framework given in Flum & Grohe 2006, Chapter 14. The definition of class #$W[1]$ (Flum & Grohe, 2006, Definition 14.11) is rather intricate and need not concern us here. We will use the following type of reduction to isolate problems that are at least as hard as the hardest (#$W[1]$-hard) in #$W[1]$.

> **Definition 16.** (Adapted from Flum & Grohe 2006, Definition 14.10.a) Given two parameterized search problems $\langle k \rangle$-Π and $\langle K \rangle$-Π', a *(fpt) parsimonious reduction from $\langle k \rangle$-Π to $\langle K \rangle$-Π'* is a function $f : I_\Pi \to I_{\Pi'}$ computable in fixed-parameter time relative to parameter $k$ such that for every $I \in \Pi$ (1) the number of solutions of Π for input $I$ is exactly equal to the number of Π' for input $f(I)$ and (2) for every parameter $k' \in K$, $k' \leq g_{k'}(k)$ for some function $g_{k'}()$.

Reductions are often established to be parsimonious by proving bijections between solution-sets for $I$ and $f(I)$, i.e., each solution to $I$ corresponds to exactly one solution for $f(I)$ and vice versa.

We will prove various parameterized results for our problems using reductions from ExClique. The parameterized results are proved relative to the parameters in Table 6. Lemmas 1 and 2 will be useful in deriving additional parameterized results from proved ones.

## C.1 RESULTS FOR SUFFICIENT CIRCUIT PROBLEMS

**Theorem 13.** For $\Pi \in L = \{\mathbf{V}LSC, \mathbf{V}GSC \mid \mathbf{V} \in \{Ex, B, MnlEx, MnlB\}\}$, ExClique polynomial-time parsimoniously reduces to $\Pi$.

*Proof.* Consider first the local sufficient circuit problem variants. Observe that for the reduction from Clique to MLSC in the proof of Theorem 1 (1) the reduction is also from ExClique, (2) each clique of size $k$ in the given instance of ExClique has exactly one corresponding sufficient circuit of size $k' = k + k(k-1)/2 + 2$ in the constructed instance of MLSC and vice versa, and (3) courtesy of the bias in neuron $n_{out}$ and the structure of the MLP $M$ in the constructed instance of MLSC, no sufficient circuit can have size $< k'$ and hence problem variants ExLSC, BLSC, MnlExLSC, and MnlBLSC (when $k' = k + k(k-1)/2 + 2$) have the same set of sufficient circuit solutions Hence, this reduction is also a polynomial-time parsimonious reduction from ExClique to each local sufficient circuit problem variant in $L$.

As for the global sufficient circuit problem variants, it was pointed out that in the proof of Theorem 7 that the reduction above is also a reduction from Clique to MGSC; moreover, all three properties above also hold modulo MGSC, ExGSC, BGSC, MnlExGSC, and MnlBGSC. Hence, this reduction is also a polynomial-time parsimonious reduction from ExClique to each global sufficient circuit problem variant in $L$. ∎

**Theorem 14.** For $\Pi \in L = \{\mathbf{V}LSC, \mathbf{V}GSC \mid \mathbf{V} \in \{Ex, B, MnlEx, MnlB\}\}$, $\langle k \rangle$-ExClique fpt parsimoniously reduces to $\langle cd_g, \#n_{in,g}, \#n_{out,g}, B_{\max,g}, W_{\max,g}, cd_r, cw_r, \#n_{in,r}, \#n_{out,r}, \#n_{tot,r}, B_{\max,r}, W_{\max,r} \rangle$-$\Pi$.

*Proof.* Observe that in the instance of MLSC constructed in the reduction in the proof of Theorem 1, $cd_g = cd_r = 4$, $\#n_{in,g} = \#n_{in,r} = \#n_{out,r} = \#n_{out,r} = W_{\max,g} = W_{\max,r} = 1$, and $B_{\max,g}, B_{\max,r}, \#n_{tot,r}$, and $cw_r$ are all functions of $k$ in the given instance of CLIQUE. The result then follows by the reasoning in the proof of Theorem 13. ∎

**Theorem 15.** For $\Pi \in L = \{\mathbf{V}LSC, \mathbf{V}GSC \mid \mathbf{V} \in \{Ex, B, MnlEx, MnlB\}\}$, if $\Pi$ is polynomial-time solvable then $P = NP$.

*Proof.* Suppose there is a polynomial-time algorithm $A$ for some $\Pi \in L$. Let $R$ be the polynomial-time algorithm underlying the polynomial-time parsimonious reduction from ExClique to $\Pi$ specified in the proof of Theorem 13. Construct an algorithm $A'$ for the decision version of ExClique as follows: Given an input $I$ for ExClique$_D$, create input $I'$ for $\Pi$ using $R$, and apply $A$ to $I'$ to create solution $S$. If $S = \bot$, return "No"; otherwise, return "Yes". Algorithm $A'$ is a polynomial-time algorithm for ExClique$_D$; however, as ExClique$_D$ is $NP$-complete (Garey & Johnson, 1979, Problem GT19), this implies that $P = NP$, giving the result. ∎

**Theorem 16.** if MinLSC or MinGSC is polynomial-time solvable then $P = NP$.

*Proof.* Suppose there is a polynomial-time algorithm $A$ for MinLSC (MinGSC). Let $R$ be the polynomial-time algorithm underlying the polynomial-time parsimonious reduction from ExClique to BLSC (BGSC) specified in the proof of Theorem 13. Construct an algorithm $A'$ for the decision version of ExClique as follows: Given an input $I$ for ExClique$_D$, create input $I'$ for BSC )LSC) using $R$, and apply $A$ to $I'$ to create solution $S$. If $|S| \leq k$, return "Yes"; otherwise, return "No". Algorithm $A'$ is a polynomial-time algorithm for ExClique$_D$; however, as ExClique$_D$ is $NP$-complete (Garey & Johnson, 1979, Problem GT19), this implies that $P = NP$, giving the result. ∎

**Theorem 17.** For $\Pi \in L = \{\mathbf{V}LSC, \mathbf{V}GSC \mid \mathbf{V} \in \{Ex, B, MnlEx, MnlB\}\}$ and $K = \{cd_g, \#n_{in,g}, \#n_{out,g}, B_{\max,g}, W_{\max,g}, cd_r, cw_r, \#n_{in,r}, \#n_{out,r}, \#n_{tot,r}, B_{\max,r}, W_{\max,r}\}$, if $\langle K \rangle$-$\Pi$ is fixed-parameter tractable then $FPT = W[1]$.

*Proof.* Suppose there is a fixed-parameter tractable algorithm $A$ for $\langle K \rangle$-$\Pi$ for some $\Pi \in L$. Let $R$ be the fixed-parameter algorithm underlying the fpt parsimonious reduction from $\langle k \rangle$-ExClique to $\langle K \rangle$-$\Pi$ specified in the proof of Theorem 14. Construct an algorithm $A'$ for the decision version of $\langle k \rangle$-ExClique as follows: Given an input $I$ for $\langle k \rangle$-ExClique$_D$, create input $I'$ for $\Pi$ using $R$, and apply $A$ to $I'$ to create solution $S$. If $S = \bot$, return "No"; otherwise, return "Yes". Algorithm $A'$ is a fixed-parameter tractable algorithm for $\langle k \rangle$-ExClique$_D$; however, as $\langle k \rangle$-ExClique$_D$ is $W[1]$-complete (Downey & Fellows, 1999), this implies that $FPT = W[1]$, giving the result. ∎

**Theorem 18.** For $K = \{cd_g, \#n_{in,g}, \#n_{out,g}, B_{\max,g}, W_{\max,g}, cd_r, cw_r, \#n_{in,r}, \#n_{out,r}, \#n_{tot,r}, B_{\max,r}, W_{\max,r}\}$, if $\langle K \rangle$-MinLSC or $\langle K] \rangle$-MinGSC is fixed-parameter tractable then $FPT = W[1]$.

*Proof.* Suppose there is a fixed-parameter tractable algorithm $A$ for $\langle K \rangle$-MinLSC ($\langle K \rangle$-MinGSC). Let $R$ be the fixed-parameter algorithm underlying the fpt parsimonious reduction from $\langle k \rangle$-ExClique to $\langle K \rangle$-BLSC ($\langle K \rangle$-BGSC) specified in the proof of Theorem 14. Construct an algorithm $A'$ for the decision version of $\langle k \rangle$-ExClique as follows: Given an input $I$ for $\langle k \rangle$-ExClique$_D$, create input $I'$ for $\Pi$ using $R$, and apply $A$ to $I'$ to create solution $S$. If $|S| \leq k$, return "Yes"; otherwise, return "No". Algorithm $A'$ is a fixed-parameter tractable algorithm for $\langle k \rangle$-ExClique$_D$; however, as $\langle k \rangle$-ExClique$_D$ is $W[1]$-complete (Downey & Fellows, 1999), this implies that $FPT = W[1]$, giving the result. ∎

**Theorem 19.** For $\Pi \in L = \{\mathbf{V}LSC \mid \mathbf{V} \in \{Ex, B, MnlEx, MnlB\}\}$, #$\Pi$ is #P-complete.

*Proof.* As #ExVC is #P-complete (Provan & Ball 1983, Page 781; see also Garey & Johnson 1979, Page 169), #ExClique is #P-hard by the polynomial-time parsimonious reduction from ExVC to ExClique implicit in Garey & Johnson 1979, Lemma 3.1. The #P-hardness of #$\Pi$ then follows from the appropriate polynomial-time parsimonious reduction from ExClique to $\Pi$ specified in the proof of Theorem 13. Membership of #$\Pi$ in #P and the result follows from the nondeterministic algorithm for #$\Pi$ that, on each computation path, guesses a subcircuit $C$ of $M$ and then verified that $C$ satisfies the properties required by $\Pi$ relative to MLP $M$ input $I$. ∎

**Theorem 20.** For $\Pi \in L = \{\mathbf{V}GSC \mid \mathbf{V} \in \{Ex, B, MnlEx, MnlB\}\}$, #$\Pi$ is #P-hard.

*Proof.* The result follows from the #P-hardness of #ExClique noted in the proof of Theorem 20 and the appropriate polynomial-time parsimonious reduction from ExClique to $\Pi$ specified in the proof of Theorem 13. ∎

**Theorem 21.** #MnlLSC is #P-complete.

*Proof.* Consider the reduction from MnlVC to MnlLSC created by modifying the reduction from VC to MLSC given in the proof of Theorem 5 such that the bias and input-line weight of input neuron $n_{in}$ are changed to 0 and 1 to ensure that all possible input vectors (namely, $\{0\}$ and $\{1\}$) cause $n_{in}$ to output 1. Observe that this modified reduction runs in time polynomial in the size of the given instance of MnlVC. We now need to show that this reduction is parsimonious, i.e., this reduction creates a bijection between the solution-sets of the given instance of MnlVC and the constructed instance of MnlLSC. We prove the two directions of this bijection separately as follows:

$\Rightarrow$ : Let $V' = \{v'_1, v'_2, \ldots, v'_k\} \subseteq V$ be a minimal vertex cover in $G$. Consider the subcircuit $C$ based on neurons $n_{in}$, $n_{out}$, $\{nv'N \mid v' \in V'\}$, $\{neA_1, neA_2, \ldots, neA_{|E|}\}$, and $\{neN_1, neN_2, \ldots, neN_{|E|}\}$. As shown in the $\Rightarrow$-portion of the proof of correctness of the reduction in the proof of Theorem 5, $C$ is behaviorally equivalent to $M$ on $I$. As $V'$ is minimal and only vertex NOT neurons can be deleted from $M$ to create $C$, $C$ must itself be minimal. Moreover, note that any such set $V'$ in $G$ is associated with exactly one set of vertex NOT neurons in (and thus exactly one sufficient circuit of) $M$.

$\Leftarrow$ : Let $C$ be a minimal subcircuit of $M$ that is behaviorally equivalent to $M$ on input $I$. As neurons in all five layers in $M$ must be present in $C$ to produce the required output, both $n_{in}$ and $n_{out}$ are in $C$. In order for $n_{out}$ to produce a non-zero output, all $|E|$ edge NOT neurons and all $|E|$ AND neurons must also be in $C$, and each of the latter must be connected to at least one of the vertex NOT neurons corresponding to their endpoint vertices. Hence, the vertices in $G$ corresponding to the vertex NOT neurons in $C$ must form a vertex cover in $G$. As $C$ is minimal and only vertex NOT neurons can be deleted from $M$ to create $C$, this vertex cover must itself be minimal. Moreover, note that any such set of vertex NOT neurons in $M$ is associated with exactly one set of vertices in (and hence exactly one vertex cover of) $G$.

As #MnlVC is #P-complete (Valiant, 1979, Theorem 1(4)), the reduction above establishes that #MnlLSC is #P-hard. Membership of #MnlLSC in #P and hence the result follows from the non-deterministic algorithm for #MnlLSC that, on each computation path, guesses a subcircuit $C$ of $M$ and then verifies that $C$ is minimal and $C(I) = M(I)$. ∎

**Theorem 22.** #MnlGSC is #P-hard.

*Proof.* Recall that the parsimonious reduction from MnlVC to MnlLSC in the proof of Theorem 21 creates an instance of MnlLSC whose MLP $M$ has the same output for every possible input vector; hence, this reduction is also a parsimonious reduction from MnlVC to MnlGSC. As #MnlVC is #P-complete (Valiant, 1979, Theorem 1(4)), this reduction establishes that #MnlGSC is #P-hard, giving the result. ∎

**Theorem 23.** For $\Pi \in L = \{\mathbf{V}LSC, \mathbf{V}GSC \mid \mathbf{V} \in \{Ex, B, MnlEx, MnlB\}\}$, if #$\Pi$ is polynomial-time solvable then $P = NP$.

*Proof.* Suppose there is a polynomial-time algorithm $A$ for #$\Pi$ for some $\Pi \in L$. Let $R$ be the polynomial-time algorithm underlying the polynomial-time parsimonious reduction from ExClique to $\Pi$ specified in the proof of Theorem 13. Construct an algorithm $A'$ for the decision version of ExClique as follows: Given an input $I$ for ExClique$_D$, create input $I'$ for $\Pi$ using $R$, and apply $A$ to $I'$ to create solution $S$. If $S = 0$, return "No"; otherwise, return "Yes". Algorithm $A'$ is a polynomial-time algorithm for ExClique$_D$; however, as ExClique$_D$ is $NP$-complete (Garey & Johnson, 1979, Problem GT19), this implies that $P = NP$, giving the result. ∎

**Theorem 24.** For $\Pi \in L = \{\mathbf{V}LSC, \mathbf{V}GSC \mid \mathbf{V} \in \{Ex, B, MnlEx, MnlB\}\}$ and $K = \{cd_g, \#n_{in,g}, \#n_{out,g}, B_{\max,g}, W_{\max,g}, cd_r, cw_r, \#n_{in,r}, \#n_{out,r}, \#n_{tot,r}, B_{\max,r}, W_{\max,r}\}$, $\langle K \rangle$-#$\Pi$ is #$W[1]$-hard.

*Proof.* The result follows from the #$W[1]$-hardness of $\langle k \rangle$-#ExClique (Flum & Grohe, 2006, Theorem 14.18) and the appropriate fpt parsimonious reduction from $\langle k \rangle$-ExClique to $\langle K \rangle$-$\Pi$ specified in the proof of Theorem 14. ∎

**Theorem 25.** For $\Pi \in L = \{\mathbf{V}LSC, \mathbf{V}GSC \mid \mathbf{V} \in \{Ex, B, MnlEx, MnlB\}\}$ and $K = \{cd_g, \#n_{in,g}, \#n_{out,g}, B_{\max,g}, W_{\max,g}, cd_r, cw_r, \#n_{in,r}, \#n_{out,r}, \#n_{tot,r}, B_{\max,r}, W_{\max,r}\}$, if $\langle K \rangle$-#$\Pi$ is fixed-parameter tractable then $FPT = \#W[1]$.

*Proof.* Suppose there is a fixed-parameter tractable algorithm $A$ for $\langle K \rangle$-#$\Pi$ for some $\Pi \in L$. Let $R$ be the fixed-parameter algorithm underlying the fpt parsimonious reduction from $\langle k \rangle$-ExClique to $\langle K \rangle$-$\Pi$ specified in the proof of Theorem 14. Construct an algorithm $A'$ for $\langle k \rangle$-#ExClique as follows: Given an input $I$ for $\langle k \rangle ra$-#ExClique, create input $I'$ for #$\Pi$ using $R$, and apply $A$ to $I'$ to create solution $S$. If $S = 0$, return "No"; otherwise, return "Yes". Algorithm $A'$ is a fixed-parameter tractable algorithm for $\langle k \rangle$-#ExClique$_D$; however, as $\langle k \rangle$-#ExClique$_D$ is #$W[1]$-complete (Flum & Grohe, 2006, Theorem 14.18), this implies that $FPT = \#W[1]$, giving the result. ∎

# D  GLOBAL SUFFICIENT CIRCUIT PROBLEM (SIGMA COMPLETENESS)

MINIMUM GLOBAL SUFFICIENT CIRCUIT (MGSC)
*Input*: A multi-layer perceptron $M$ of depth $cd$ with $\#n_{tot}$ neurons and maximum layer width $cw$, connection-value matrices $W_1, W_2, \ldots, W_{cd_g}$, neuron bias vector $B$, and a positive integer $k$.
*Question*: Is there a subcircuit $C$ of $M$ based on $\leq k$ neurons from $M$ such that for every possible input $I$ of $M$, $C(I) = M(I)$?

Given a subset $N$ of the neurons in $M$, the subcircuit $C$ of $M$ based on $x$ has the neurons in $x$ and all connections in $M$ among these neurons. Note that in order for the output of $C$ to be equal to the output of $M$ on input $I$, the numbers $\#n_{in}$ and $\#n_{out}$ of input and output neurons in $M$ must exactly equal the numbers of input and output neurons in $C$; hence, no input or output neurons can be deleted from $M$ in creating $C$. Following Barceló et al. 2020, page 4, all neurons in $M$ use the ReLU activation function and the output $x$ of each output neuron is stepped as necessary to be Boolean, i.e, $step(x) = 0$ if $x \leq 0$ and is 1 otherwise.

We will prove our result for MGSC using a polynomial-time reduction from the problem MINIMUM DNF TAUTOLOGY. Given a DNF formula $\phi$ over a set $V$ of variables, $\phi$ is a **tautology** if $\phi$ evaluates to $True$ for every possible truth-assignment to the variables in $V$.

Our reduction will use specialized ReLU logic gates described in Barceló et al. 2020, Lemma 13. These gates assume Boolean neuron input and output values of 0 and 1 and are structured as follows:

1. NOT ReLU gate: A ReLU gate with one input connection weight of value $-1$ and a bias of 1. This gate has output 1 if the input is 0 and 0 otherwise.

2. $n$-way AND ReLU gate: A ReLU gate with $n$ input connection weights of value 1 and a bias of $-(n-1)$. This gate has output 1 if all inputs have value 1 and 0 otherwise.

3. $n$-way OR ReLU gate: A combination of an $n$-way AND ReLU gate with NOT ReLU gates on all of its inputs and a NOT ReLU gate on its output that uses DeMorgan's Second Law to implement $(x_1 \vee x_2 \vee \ldots x_n)$ as $\neg(\neg x_1 \wedge \neg x_2 \wedge \ldots \neg x_n)$. This gate has output 1 if any input has value 1 and 0 otherwise.

**Theorem 26.** MGSC is $\Sigma_2^p$-complete.

*Proof.* Let us first show the membership of MGSC in $\Sigma_2^p$. Using the alternating-quantifier definition of classes in the polynomial hierarchy, membership of a decision problem $\Pi$ in $\Sigma_2^p$ can be proved by showing that solving $\Pi$ for input $I$ is equivalent to solving a quantified formula of the form $\exists(x)\forall(y) : p(x, y)$ where both the sizes of $x$ and $y$ and the evaluation time of predicate formula $p()$ are upper-bounded by polynomials in $|I|$. Such a formula for MGSC is

$$\exists(\mathcal{C} \subseteq \mathcal{M})\forall\mathbf{x} \in \{0, 1\}^{\#n_{in}} : \mathcal{C}(\mathbf{x}) = \mathcal{M}(\mathbf{x})$$

We now show the $\Sigma_2^p$-hardness of MGSC. Consider the following reduction from 3DT to MGSC. Given an instance $\langle \phi, T, V, k \rangle$ of 3DT, construct the following instance $\langle M, k' \rangle$ of MGSC: Let $M$ be an MLP based on $3|V| + 2T + 2$ neurons spread across five layers:

1. **Input neuron layer**: The input neurons $ni_1, ni_2, \ldots, ni_{|V|}$ (all with bias 0).

2. **Hidden layer I**: The unnegated variable identity neurons $nvU_1, nvU_2, \ldots, nvU_{|V|}$ (all with bias 0) and negated variable NOT neurons $nvN_1, nvN_2, \ldots, nvN_{|V|}$ (all with bias 1).

3. **Hidden layer II**:
   (a) The term 3-way AND neurons $nT_1, xT_2, \ldots, xT_{|T|}$ (all with bias -2).
   (b) The gadget neuron $n_g$ (bias $|V| - 1$).

4. **Hidden layer III**: The modified term 2-way AND neurons $nTm_1, nTM_2, \ldots, nTM_{|T|}$ (all with bias $-1$).

5. **Output layer**: The stepped output neuron $n_{out}$.

The non-zero weight connections between adjacent layers are as follows:

- Each input neuron $ni_i$, $1 \leq i \leq |V|$, is input-connected with weight 1 to its corresponding input line and output-connected with weights 1 and -1 to unnegated and negated variable neurons $nvU_i$ and $nvN_i$, respectively.

- Each term neuron $nT_i$, $1 \leq i \leq |T|$, is input-connected with weight 1 to each of the 3 variable neurons corresponding to that term's literals.

- The gadget neuron $n_g$ is input-connected with weight 1 to all of the unnegated and negated variable neurons.

- Each modified term neuron $nTm_i$, $1 \leq i \leq |T|$, is input-connected with weight 1 to the term neuron $nT_i$ and the gadget neuron $n_g$ and output-connected with weight 1 to the output neuron $n_{out}$.

All other connections between neurons in adjacent layers have weight 0. Finally, let $k' = 3|V| + 2k + 2$. Observe that this instance of MGSC can be constructed time polynomial in the size of the given instance of 3DT.

The following observations about the MLP $M$ constructed above will be of use:

- The input to $M$ is exactly that of the 3-DNF formula $\phi$.

- The output neuron of $M$ outputs 1 if and only if one or more of the modified term neurons output 1.

- Modified term neuron $nTm_i$ outputs 1 if and only both the term neuron $nT_i$ and the gadget neuron $n_g$ output 1.

- The gadget neuron $n_g$ outputs 1 for input $I$ to a subcircuit $C$ of $M$ if and only if the negated and unnegated variable neurons corresponding to $I$ each output 1; hence, $n_g$ outputs 1 for all possible inputs to $C$ if and only if all negated and unnegated variable neurons in hidden layer I are part of $C$.

As $\phi$ is a tautology, the above implies that (1) $M$ outputs 1 for every possible input and (2) every global sufficient circuit of $M$ must include all input and variable neurons, the gadget and output neurons, and at least one term / modified term neuron-pair.

We now need to show the correctness of this reduction by proving that the answer for the given instance of 3DT is "Yes" if and only if the answer for the constructed instance of MGSC is "Yes". We prove the two directions of this if and only if separately as follows:

$\Rightarrow$ : Let $T'$, $|T'| = k$ be a subset of the terms in $\phi$ that is a tautology. As noted above, any global sufficient circuit $C$ for the constructed MLP $M$ must include all input and variable neurons, the gadget and output neurons, and at least one term / modified term neuron-pair. Let $C$ contain the term / modified term neuron-pairs corresponding to the terms in $T'$. Such a $C$ is therefore a global sufficient circuit for $M$ of size $k' = 3|V| = 2k + 2$.

$\Leftarrow$ : Let $C$ be a global sufficient circuit for $M$ of size $k' = 3|V| + 2k + 2$. As noted above, $C$ must include all input and variable neurons, the gadget and output neurons, and $k$ term / modified term neuron-pairs. Let $T'$ be the subset of $k$ terms in $\phi$ corresponding to these neuron-pairs. Given the input-output equivalence $\phi$ and $M$ (and hence any sufficient circuit for $M$), the disjunction of the $k$ terms in $T'$ must be a tautology.

As 3DT is $\Sigma_2^p$-hard (Schaefer & Umans, 2002, Problem L7), the reduction above establishes that MGSC is also $\Sigma_2^p$-hard. The result then follows from the membership of MGSC in $\Sigma_2^p$ shown at the beginning of this proof. ∎

# E    Quasi-Minimal Sufficient Circuit Problem

Quasi-Minimal Sufficient Circuit (QMSC)
*Input*: A multi-layer perceptron $M$ of depth $cd$ with $\#n_{tot}$ neurons and maximum layer width $cw$, connection-value matrices $W_1, W_2, \ldots, W_{cd}$, neuron bias vector $B$, a set $\mathcal{X}$ of input vectors of length $\#n_{in}$.
*Output*: a circuit $\mathcal{C}$ in $\mathcal{M}$ and a neuron $v \in \mathcal{C}$ such that $\mathcal{C}(\mathbf{x}) = \mathcal{M}(\mathbf{x})$ and $[\mathcal{C} \setminus \{v\}](\mathbf{x}) \neq \mathcal{M}(\mathbf{x})$

**Theorem 27.**   QMSC is in PTIME (i.e., polynomial-time tractable).

*Proof.* Consider the following algorithm for QMSC. Build a sequence of MLPs by taking $\mathcal{M}$ with all neurons labeled 1, and generating subsequent $\mathcal{M}_i$ in the sequence by labeling an additional neuron with 0 each time (this choice can be based on any heuristic strategy, for instance, one based on gradients). The first MLP, $\mathcal{M}_1$, obtained by removing all neurons labeled 0 (i.e., none) is such that $\mathcal{M}_1(x) = \mathcal{M}(x)$, and the last $\mathcal{M}_n$ is guaranteed to give $\mathcal{M}_n(x) \neq \mathcal{M}(x)$ because all neurons are removed. Label the first MLP YES, and the last NO. Perform a variant of binary search on the sequence as follows. Evaluate the $\mathcal{M}_i$ halfway between YES and NO while removing all its neurons labeled 0. If it satisfies the condition, label it YES, and repeat the same strategy with the sequence starting from the YES just labeled until the last $\mathcal{M}_n$. If it does *not* satisfy the condition, label it NO and repeat the same strategy with the sequence starting from the YES at the beginning of the original sequence until the NO just labeled. This iterative procedure halves the sequence each time. Halt when you find two adjacent $\langle \text{YES}, \text{NO} \rangle$ circuits (guaranteed to exist), and return the circuit set of the YES network V and the single neuron difference between YES and NO (the breaking point), $v \in V$. The complexity of this algorithm is roughly $O(n \log n)$.
∎

# F    Gnostic Neurons Problem

Gnostic Neurons (GN)
*Input*: A multi-layer perceptron $M$ of depth $cd$ with $\#n_{tot}$ neurons and maximum layer width $cw$, connection-value matrices $W_1, W_2, \ldots, W_{cd}$, neuron bias vector $B$, and two sets $\mathcal{X}$ and $\mathcal{Y}$ of input vectors of length $\#n_{in}$, and a positive integer $k$ such that $1 \leq k \leq \#n_{tot}$.
*Output*: a subset of neurons $V$ in $M$ of size $|V| \geq k$ such that $\forall_{v \in V}$ it is the case that $\forall_{\mathbf{x} \in \mathcal{X}}$ computing $M(\mathbf{x})$ produces activations $A_{\mathbf{x}}^v \geq t$ and $\forall_{\mathbf{y} \in \mathcal{Y}} : A_{\mathbf{y}}^v < t$.

**Theorem 28.**   GN is in PTIME (i.e., polynomial-time tractable).

*Proof.* Consider the complexity of the following subroutines of an algorithm for GN. Computing the activations of all neurons of $M$ for all $\mathbf{x} \in \mathcal{X}$ and all $\mathbf{y} \in \mathcal{Y}$ takes polynomial time in $|M|, |\mathcal{X}|$ and $|\mathcal{Y}|$. Labeling neurons that pass or not the activation threshold takes time polynomial in $|M|, |\mathcal{X}|$ and $|\mathcal{Y}|$. Finally, checking whether the set of neurons that fulfils the condition is of size at least $k$ can be done in polynomial time in $|M|$. These subroutines can be put together to yield a polynomial-time algorithm for GN.
∎

*Remark* 1   One could also add to the output of the computational problem the requirement that if we silence (or activate) the neuron, we should elicit (or abolish) a behavior. Note that checking these effects can be done in polynomial time in all of the input parts given above and also in the size of the behavior set (which should be added to the input in these variants).

# G    Necessary Circuit Problem

Minimum Local Necessary Circuit (MLNC)
*Input*: A multi-layer perceptron $M$ of depth $cd$ with $\#n_{tot}$ neurons and maximum layer width $cw$, connection-value matrices $W_1, W_2, \ldots, W_{cd}$, neuron bias vector $B$, a Boolean input vector $I$ of length $\#n_{in}$, and a positive integer $k$ such that $1 \leq k \leq \#n_{tot}$.

Table 7: Parameters for the minimum necessary circuit problem.

| Parameter | Description |
|---|---|
| $cd$ | # layers in given MLP |
| $cw$ | max # neurons in layer in given MLP |
| $\#n_{tot}$ | total # neurons in given MLP |
| $\#n_{in}$ | # input neurons in given MLP |
| $\#n_{out}$ | # output neurons in given MLP |
| $B_{\max}$ | max neuron bias in given MLP |
| $W_{\max}$ | max connection weight in given MLP |
| $k$ | Size of requested neuron subset |

*Question*: Is there a subset $N'$, $|N'| \leq k$, of the $|N|$ neurons in $M$ such that $N' \cap C \neq \emptyset$ for every sufficient circuit $C$ of $M$ relative to $I$?

MINIMUM GLOBAL NECESSARY CIRCUIT (MGNC)
*Input*: A multi-layer perceptron $M$ of depth $cd$ with $\#n_{tot}$ neurons and maximum layer width $cw$, connection-value matrices $W_1, W_2, \ldots, W_{cd}$, neuron bias vector $B$, and a positive integer $k$ such that $1 \leq k \leq \#n_{tot}$.
*Question*: Is there a subset $N'$, $|N'| \leq k$, of the $|N|$ neurons in $M$ such that for for every possible Boolean input vector $I$ of length $\#n_{in}$, $N' \cap C \neq \emptyset$ for every sufficient circuit $C$ of $M$ relative to $I$?

We will use reductions from the Hitting Set problem to prove our results for the problems above.

Regarding the Hitting Set problem, we shall assume an ordering on the sets and elements in $C$ and $S$, respectively.

Our parameterized results are proved relative to the parameters in Table 7. Lemmas 1 and 2 will be useful in deriving additional parameterized results from proved ones.

Our reductions will use specialized ReLU logic gates described in Barceló et al. 2020, Lemma 13. These gates assume Boolean neuron input and output values of 0 and 1 and are structured as follows:

1. NOT ReLU gate: A ReLU gate with one input connection weight of value $-1$ and a bias of 1. This gate has output 1 if the input is 0 and 0 otherwise.

2. $n$-way AND ReLU gate: A ReLU gate with $n$ input connection weights of value 1 and a bias of $-(n-1)$. This gate has output 1 if all inputs have value 1 and 0 otherwise.

3. $n$-way OR ReLU gate: A combination of an $n$-way AND ReLU gate with NOT ReLU gates on all of its inputs and a NOT ReLU gate on its output that uses DeMorgan's Second Law to implement $(x_1 \vee x_2 \vee \ldots x_n)$ as $\neg(\neg x_1 \wedge \neg x_2 \wedge \ldots \neg x_n)$. This gate has output 1 if any input has value 1 and 0 otherwise.

### G.1 RESULTS FOR MLNC

Membership of MLNC in $\Sigma_2^p$ can be proven via the definition of the polynomial hierarchy and the following alternating quantifier formula:

$$\exists[N \subseteq \mathcal{M}] \, \forall[\mathcal{C} \subseteq \mathcal{M}] : [\mathcal{C}(\mathbf{x}) = \mathcal{M}(\mathbf{x})] \implies N \cap \mathcal{C} \neq \emptyset$$

**Theorem 29.** *If MLNC is polynomial-time tractable then $P = NP$.*

*Proof.* Consider the following reduction from HS to MLNC. Given an instance $\langle C, S, k \rangle$ of HS, construct the following instance $\langle M, I, k' \rangle$ of MLNC: Let $M$ be an MLP based on $\#n_{tot} = |S| + |C| + 2$ neurons spread across four layers:

1. **Input neuron layer**: The single input neuron $n_{in}$ (bias $+1$).

2. **Hidden element layer**: The element neurons $ns_1, ns_2, \ldots ns_{|S|}$ (all with bias 0).

3. **Hidden set layer**: The set AND neurons $nc_1, nc_2, \ldots, nc_{|C|}$ (such that neuron $nc_i$ has bias $-|c_i|$).

4. **Output layer**: The single stepped output neuron $n_{out}$ (bias 0).

The non-zero weight connections between adjacent layers are as follows:

- The input neuron has an edge of weight 0 coming from its input and is in turn connected to each of the element neurons with weight 1.

- Each element neuron $ns_i$, $1 \leq i \leq |S|$, is connected to each set neuron $nc_j$, $1 \leq j \leq |C|$, such that $s_i \in c_j$ with weight 1.

- Each set neuron $nc_i$, $1 \leq i \leq |C|$, is connected to the output neuron with weight 1.

All other connections between neurons in adjacent layers have weight 0. Finally, let $I = (0)$ and $k' = k$. Observe that this instance of MLNC can be created in time polynomial in the size of the given instance of HS. Moreover, the output behaviour of the neurons in $M$ from the presentation of input $I$ until the output is generated is as follows:

| timestep | neurons (outputs) |
|----------|-------------------|
| 0 | — |
| 1 | $n_{in}(1)$ |
| 2 | $ns_1, ns_2, \ldots, ns_{|S|}$ (1) |
| 3 | $nc_1, nc_2, \ldots, nc_{|C|}$ (1) |
| 4 | $n_{out}(1))$ |

Note the following about the behavior of $M$:

**Observation 1.** For any set neuron $nc_i$ to output 1, it must receive input 1 from all of its incoming element neurons connected with weight 1.

**Observation 2.** For the output neuron to output 1, it is sufficient to get input 1 from any of its incoming set neurons with weight 1.

Observations 1 and 2 imply that any sufficient circuit for $M$ must contain at least one set neuron and all of its associated element neurons.

We now need to show the correctness of this reduction by proving that the answer for the given instance of HS is "Yes" if and only if the answer for the constructed instance of MLNC is "Yes". We prove the two directions of this if and only if separately as follows:

$\Rightarrow$ : Let $S' = \{s'_1, s'_2, \ldots, s'_k\} \subseteq S$ be a hitting set of size $k$ for $C$. By the construction above, the $k$ element neurons corresponding to the elements in $S'$ collectively connect with weight 1 to all set neurons in $M$. By Observations 1 and 2, this means that the set $N$ of these element neurons has a non-empty intersection with every sufficient circuit for $M$, and hence that $N$ is a necessary circuit for $M$ of size $k = k'$.

$\Leftarrow$ : Let $N$ be a necessary circuit for $M$ of size $k'$. Let $N_S$ and $N_C$ be the subsets of $N$ that are element and set neurons. We can create a set $N'$ consisting only of element neurons by replacing each set neuron $n_c$ in $N_C$ with an arbitrary element neuron that is not already in $N_S$ and is connected to $n_c$ with weight 1. Observe that $N'$ (whose size may be less than $k'$ if any $n_c$ already had an associated element neuron in $N_S$) remains a necessary circuit for $M$. Moreover, as the element neurons in $N'$ by definition have a non-empty intersection with each sufficient circuit for $M$, by Observations 1 and 2 above, the set $S'$ of elements in

$S$ corresponding to the element neurons in $N'$ has a non-empty intersection with each set in $C$ and hence is a hitting set of size $N' \le k' = k$

As HS is $NP$-hard (Garey & Johnson, 1979), the reduction above establishes that MLNC is also $NP$-hard. The result follows from the definition of $NP$-hardness. ∎

**Theorem 30.** If $\langle cd, \#n_{in}, \#n_{out}, W_{\max}, k \rangle$-MLNC is fixed-parameter tractable then $FPT = W[1]$.

*Proof.* Observe that in the instance of MLNC constructed in the reduction in the proof of Theorem 29, $\#n_{in} = \#n_{out} = W_{\max} = 1$, $cd = 4$, and $k'$ is a function of $k$ in the given instance of HS. The result then follows from the facts that $\langle k \rangle$-HS is $W[2]$-hard (by a reduction from $\langle k \rangle$-DOMINATING SET; Downey & Fellows 1999) and $W[1] \subseteq W[2]$. ∎

**Theorem 31.** $\langle \#n_{tot} \rangle$-MLNC is fixed-parameter tractable.

*Proof.* Consider the algorithm that generates every possible subset $N'$ of size at most $k$ of the neurons in MLP $M$ and for each such subset, generates every possible subset $N''$ of $M$, checks if $N''$ is a sufficient circuit for $M$ relative to $I$ and, if so, checks if $N'$ has a non-empty intersection with $N''$. If an $N'$ is found that has a non-empty intersection with each sufficient circuit for $M$ relative to $I$, return "Yes"; otherwise, return "No". The number of possible subsets $N'$ and $N''$ are both at most $2^{\#n_{tot}}$. As all subsequent checking operations can be done in time polynomial in the size of the given instance of MLNC, the above is a fixed-parameter tractable algorithm for MLNC relative to parameter-set $\{\#n_{tot}\}$. ∎

**Theorem 32.** $\langle cw, cd \rangle$-MLNC is fixed-parameter tractable.

*Proof.* Follows from the algorithm in the proof of Theorem 31 and the observation that $\#n_{tot} \le cw \times cd$. ∎

Observe that the results in Theorems 30–32 in combination with Lemmas 1 and 2 suffice to establish the parameterized complexity status of MLNC relative to many subsets of the parameters listed in Table 7.

Let us now consider the polynomial-time cost approximability of MLNC.

**Theorem 33.** If MLNC has a polynomial-time $c$-approximation algorithm for any constant $c > 0$ then $P = NP$.

*Proof.* Recall from the proof of correctness of the reduction in the proof of Theorem 29 that a given instance of HS has a hitting set of size $k$ if and only if the constructed instance of MLNC has a necessary circuit of size $k' = k$. This implies that, given a polynomial-time $c$-approximation algorithm $A$ for MLNC for some constant $c > 0$, we can create a polynomial-time $c$-approximation algorithm for HS by applying the reduction to the given instance $x$ of HS to construct an instance $x'$ of MLNC, applying $A$ to $x'$ to create an approximate solution $y'$, and then using $y'$ to create an approximate solution $y$ for $x$ that has the same cost as $y'$. The result then follows from Ausiello et al. 1999, Problem SP7, which states that if HS has a polynomial-time $c$-approximation algorithm for any constant $c > 0$ a and is hence in approximation problem class APX then $P = NP$. ∎

Note that this theorem also renders MLNC PTAS-inapproximable unless $FPT = W[1]$.

### G.2 RESULTS FOR MGNC

Membership of MLNC in $\Sigma_2^p$ can be proven via the definition of the polynomial hierarchy and the following alternating quantifier formula:

$$\exists [N \subseteq \mathcal{M}] \, \forall [\mathcal{C} \subseteq \mathcal{M} \, s.t. \, \forall (\mathbf{x} \in \{0,1\}^{\#n_{in}})] : N \cap \mathcal{C} \neq \emptyset$$

**Theorem 34.** If MGNC is polynomial-time tractable then $P = NP$.

*Proof.* Observe that in the instance of MLNC constructed by the reduction in the proof of Theorem 29, the input-connection weight 0 and bias 1 of the input neuron force this neuron to output 1 for both of the possible input vectors $(1)$ and $(0)$. Hence, with slight modifications to the proof of reduction correctness, this reduction also establishes the $NP$-hardness of MGNC. ∎

**Theorem 35.** If $\langle cd, \#n_{in}, \#n_{out}, W_{\max}, k \rangle$-MGNC is fixed-parameter tractable then $FPT = W[1]$.

*Proof.* Observe that in the instance of MGNC constructed in the reduction in the proof of Theorem 34, $\#n_{in} = \#n_{out} = W_{\max} = 1$, $cd = 4$, and $k'$ is a function of $k$ in the given instance of HS The result then follows from the facts that $\langle k \rangle$-HS is $W[2]$-hard (by a reduction from $\langle k \rangle$-DOMINATING SET; Downey & Fellows 1999) and $W[1] \subseteq W[2]$. ∎

**Theorem 36.** $\langle \#n_{tot} \rangle$-MGNC is fixed-parameter tractable.

*Proof.* Modify the algorithm in the proof of Theorem 31 such that each potential sufficient circuit $N''$ is checked to ensure that $M(I) = N''(I)$ for every possible Boolean input vector of length $\#n_{in}$. As the number of such vectors is $2^{\#n_{in}} < 2^{\#n_{tot}}$, the above is a fixed-parameter tractable algorithm for MGNC relative to parameter-set $\{\#n_{tot}\}$. ∎

**Theorem 37.** $\langle cw, cd \rangle$-MGNC is fixed-parameter tractable.

*Proof.* Follows from the algorithm in the proof of Theorem 36 and the observation that $\#n_{tot} \leq cw \times cd$. ∎

Observe that the results in Theorems 35–37 in combination with Lemmas 1 and 2 suffice to establish the parameterized complexity status of MGNC relative to many subsets of the parameters listed in Table 7.

Let us now consider the polynomial-time cost approximability of MGNC.

**Theorem 38.** If MGNC has a polynomial-time $c$-approximation algorithm for any constant $c > 0$ then $P = NP$.

*Proof.* As the reduction in the proof of Theorem 34 is essentially the same as the reduction in the proof of Theorem 29, the result follows by the same reasoning as given in the proof of Theorem 33. ∎

Note that this theorem also renders MGNC PTAS-inapproximable unless $FPT = W[1]$.

## H   CIRCUIT ABLATION AND CLAMPING PROBLEMS

Given an MLP $M$ and a subset $N$ of the neurons in $M$, the MLP $M'$ induced by $N$ is said to be *active* if there is at least one path between the the input and output neurons in $M'$; otherwise, $M'$ is *inactive*. As we are interested in inductions that preserve or violate output behaviour, all output neurons of $M$ must be preserved in $M'$; however, we only require that at least one input neuron be so preserved. Unless otherwise stated, all inductions discussed wrt MLCA and MGCA below will be assumed to result in active MLP.

MINIMUM LOCAL CIRCUIT ABLATION (MLCA)
*Input*: A multi-layer perceptron $M$ of depth $cd$ with $\#n_{tot}$ neurons and maximum layer width $cw$, connection-value matrices $W_1, W_2, \ldots, W_{cd}$, neuron bias vector $B$, a Boolean input vector $I$ of length $\#n_{in}$, and a positive integer $k$ such that $1 \le k \le \#n_{tot}$.
*Question*: Is there a subset $N'$, $|N'| \le k$, of the $|N|$ neurons in $M$ such that $M(I) \ne M'(I)$ for the MLP $M'$ induced by $N \setminus N'$?

MINIMUM GLOBAL CIRCUIT ABLATION (MGCA)
*Input*: A multi-layer perceptron $M$ of depth $cd$ with $\#n_{tot}$ neurons and maximum layer width $cw$, connection-value matrices $W_1, W_2, \ldots, W_{cd}$, neuron bias vector $B$, and a positive integer $k$ such that $1 \le k \le \#n_{tot}$.
*Question*: Is there a subset $N'$, $|N'| \le k$, of the $|N|$ neurons in $M$ such that for the MLP $M'$ induced by $N \setminus N'$, $M(I) \ne M'(I)$ for every possible Boolean input vector $I$ of length $\#n_{in}$?

Given an MLP $M$ and a neuron $v$ in $M$, $v$ *is clamped to value* $val$ if the output of $v$ is always $val$ regardless of the inputs to $v$. As one can trivially change the output of an MLP by clamping one or more of its output neurons, we shall not allow the clamping of output neurons in the problems below.

MINIMUM LOCAL CIRCUIT CLAMPING (MLCC)
*Input*: A multi-layer perceptron $M$ of depth $cd$ with $\#n_{tot}$ neurons and maximum layer width $cw$, connection-value matrices $W_1, W_2, \ldots, W_{cd}$, neuron bias vector $B$, a Boolean input vector $I$ of length $\#n_{in}$, a Boolean value $val$, and a positive integer $k$ such that $1 \le k \le \#n_{tot}$.
*Question*: Is there a subset $N'$, $|N'| \le k$, of the $|N|$ neurons in $M$ such that $M(I) \ne M'(I)$ for the MLP $M'$ in which all neurons in $N'$ are clamped to value $val$?

MINIMUM GLOBAL CIRCUIT CLAMPING (MGCC)
*Input*: A multi-layer perceptron $M$ of depth $cd$ with $\#n_{tot}$ neurons and maximum layer width $cw$, connection-value matrices $W_1, W_2, \ldots, W_{cd}$, neuron bias vector $B$, a Boolean value $val$, and a positive integer $k$ such that $1 \le k \le \#n_{tot}$.
*Question*: Is there a subset $N'$, $|N'| \le k$, of the $|N|$ neurons in $M$ such that for the MLP $M'$ in which all neurons in $N'$ are clamped to value $val$, $M(I) \ne M'(I)$ for every possible Boolean input vector $I$ of length $\#n_{in}$?

Following Barceló et al. 2020, page 4, all neurons in $M$ use the ReLU activation function and the output $x$ of each output neuron is stepped as necessary to be Boolean, i.e, $step(x) = 0$ if $x \le 0$ and is 1 otherwise.

For a graph $G = (V, E)$, we shall assume an ordering on the vertices and edges in $V$ and $E$, respectively. For each vertex $v \in V$, let the complete neighbourhood $N_C(v)$ of $v$ be the set composed of $v$ and the set of all vertices in $G$ that are adjacent to $v$ by a single edge, i.e., $v \cup \{u \mid u \in V \text{ and } (u, v) \in E\}$.

We will prove various classical and parameterized results for MLCA, MGCA, MLCC, and MGCC using reductions from CLIQUE. The parameterized results are proved relative to the parameters in Table 8. Lemmas 1 and 2 will be useful in deriving additional parameterized results from proved ones.

Table 8: Parameters for the minimum circuit ablation and clamping problems.

| Parameter | Description |
|---|---|
| $cd$ | # layers in given MLP |
| $cw$ | max # neurons in layer in given MLP |
| $\#n_{tot}$ | total # neurons in given MLP |
| $\#n_{in}$ | # input neurons in given MLP |
| $\#n_{out}$ | # output neurons in given MLP |
| $B_{\max}$ | max neuron bias in given MLP |
| $W_{\max}$ | max connection weight in given MLP |
| $k$ | Size of requested neuron subset |

Additional reductions from DS used to prove polynomial-time cost inapproximability use specialized ReLU logic gates described in Barceló et al. 2020, Lemma 13. These gates assume Boolean neuron input and output values of 0 and 1 and are structured as follows:

1. **NOT ReLU gate**: A ReLU gate with one input connection weight of value $-1$ and a bias of 1. This gate has output 1 if the input is 0 and 0 otherwise.

2. $n$-way AND ReLU gate: A ReLU gate with $n$ input connection weights of value 1 and a bias of $-(n-1)$. This gate has output 1 if all inputs have value 1 and 0 otherwise.

3. $n$-way OR ReLU gate: A combination of an $n$-way AND ReLU gate with NOT ReLU gates on all of its inputs and a NOT ReLU gate on its output that uses DeMorgan's Second Law to implement $(x_1 \vee x_2 \vee \ldots x_n)$ as $\neg(\neg x_1 \wedge \neg x_2 \wedge \ldots \neg x_n)$. This gate has output 1 if any input has value 1 and 0 otherwise.

## H.1 RESULTS FOR MINIMAL CIRCUIT ABLATION

The following hardness and inapproximability results are notable for holding when the given MLP $M$ has three hidden layers.

### H.1.1 RESULTS FOR MLCA

Towards proving NP-completeness, we first prove membership and then follow up with hardness. Membership in NP can be proven via the definition of the polynomial hierarchy and the following alternating quantifier formula:

$$\exists[\mathcal{S} \subseteq \mathcal{M}] : [\mathcal{M} \setminus \mathcal{S}](\mathbf{x}) \neq \mathcal{M}(\mathbf{x})$$

**Theorem 39.** If MLCA is polynomial-time tractable then $P = NP$.

*Proof.* Consider the following reduction from CLIQUE to MLCA. Given an instance $\langle G = (V, E), k \rangle$ of CLIQUE, construct the following instance $\langle M, I, k' \rangle$ of MLCA: Let $M$ be an MLP based on $\#n_{tot} = 3|V| + |E| + 2$ neurons spread across five layers:

1. **Input neuron layer**: The single input neuron $n_{in}$ (bias $+1$).

2. **Hidden vertex pair layer**: The vertex neurons $nvP1_1, nvP1_2, \ldots nvP1_{|V|}$ and $nvP2_1, nvP2_2, \ldots nvP2_{|V|}$ (all with bias 0).

3. **Hidden vertex regulator layer**: The vertex neurons $nvR_1, nvR_2, \ldots nR_{|V|}$ (all with bias 0).

4. **Hidden edge layer**: The edge neurons $ne_1, ne_2, \ldots ne_{|E|}$ (all with bias $-1$).

5. **Output layer**: The single output neuron $n_{out}$ (bias $-(k(k-1)/2 - 1)$).

The non-zero weight connections between adjacent layers are as follows:

- Each input neuron has an edge of weight 0 coming from its corresponding input and is in turn connected to each of the vertex pair neurons with weight 1.

- Each P1 (P2) vertex pair neuron $nvP1_i$ ($nvP2_i$), $1 \leq i \leq |V|$, is connected to vertex regulator neuron $nvR_i$ with weight $-2$ (1).

- Each vertex regulator neuron $nv_i$, $1 \leq i \leq |V|$, is connected to each edge neuron whose corresponding edge has an endpoint $v_i$ with weight 1.

- Each edge neuron $ne_i$, $1 \leq i \leq |E|$, is connected to the output neuron $n_{out}$ with weight 1.

All other connections between neurons in adjacent layers have weight 0. Finally, let $I = (1)$ and $k' = k$. Observe that this instance of MLCA can be created in time polynomial in the size of the given instance of CLIQUE. Moreover, the output behaviour of the neurons in $M$ from the presentation of input $I$ until the output is generated is as follows:

| timestep | neurons (outputs) |
|---|---|
| 0 | — |
| 1 | $n_{in}(1)$ |
| 2 | $nvP1_1(1), nvP1_2(1), \ldots nvP1_{|V|}(1), nvP2_1(1), nvP2_2(1), \ldots nvP2_{|V|}(1)$ |
| 3 | $nvR_1(0), nvR_2(0), \ldots nvR_{|V|}(0)$ |
| 4 | $ne_1(0), ne_2(0), \ldots ne_{|E|}(0)$ |
| 5 | $n_{out}(0))$ |

We now need to show the correctness of this reduction by proving that the answer for the given instance of CLIQUE is "Yes" if and only if the answer for the constructed instance of MLCA is "Yes". We prove the two directions of this if and only if separately as follows:

$\Rightarrow$ : Let $V' = \{v_1', v_2', \ldots, v_k'\} \subseteq V$ be a clique in $G$ of size $k'' \geq k$ and $N'$ be the $k'' \geq k' = k$-sized subset of the P1 vertex pair neurons corresponding to the vertices in $V'$. Let $M'$ be the version of $M$ in which all neurons in $N'$ are ablated. As each of these vertex pair neurons previously forced their associated vertex regulator neurons to output 0 courtesy of their connection-weight of $-2$, their ablation now allows these $k''$ vertex regulator neurons to output 1. As $V'$ is a clique of size $k''$, exactly $k''(k''-1)/2 \geq k(k-1)/2$ edge neurons in $M'$ receive the requisite inputs of 1 on both of their endpoints from the vertex regulator neurons associated with the P1 vertex pair neurons in $N'$. This in turn ensures the output neuron produces output 1. Hence, $M(I) = 0 \neq 1 = M'(I)$.

$\Leftarrow$ : Let $N'$ be a subset of $N$ of size at most $k' = k$ such that for the MLP $M'$ induced by ablating all neurons in $N'$, $M(I) \neq M(I')$. As $M(I) = 0$ and circuit outputs are stepped to be Boolean, $M'(I) = 1$. Given the bias of the output neuron, this can only occur if at least $k(k-1)/2$ edge neurons in $M'$ have output 1 on input $I$, which requires that each of these neurons receives 1 from both of its endpoint vertex regulator neurons. These vertex regulator neurons can only output 1 if all of their associated P1 vertex neurons have been ablated; moreover, there must be exactly $k$ such neurons. This means that the vertices in $G$ corresponding to the P1 vertex pair neurons in $N'$ must form a clique of size $k$ in $G$.

As CLIQUE is $NP$-hard (Garey & Johnson, 1979), the reduction above establishes that MLCA is also $NP$-hard. The result follows from the definition of $NP$-hardness. ∎

**Theorem 40.** If $\langle cd, \#n_{in}.\#n_{out}, W_{\max}, B_{\max}, k \rangle$-MLCA is fixed-parameter tractable then $FPT = W[1]$.

*Proof.* Observe that in the instance of MLCA constructed in the reduction in the proof of Theorem 39, $\#n_{in} = \#n_{out} = W_{\max} = 1$, $cd = 5$, and $B_{\max}$ and $k$ are function of $k$ in the given instance of CLIQUE. The result then follows from the fact that $\langle k \rangle$-CLIQUE is $W[1]$-hard (Downey & Fellows, 1999). ∎

**Theorem 41.** $\langle \#n_{tot} \rangle$-MLCA is fixed-parameter tractable.

*Proof.* Consider the algorithm that generates every possible subset $N'$ of size at most $k$ of the neurons $N$ in MLP $M$ and for each such subset, creates the MLP $M'$ induced from $M$ by ablating the neurons in $N'$ and (assuming $M'$ is active) checks if $M'(I) \neq M(I)$. If such a subset is found, return "Yes"; otherwise, return "No". The number of possible subsets $N'$ is at most $k \times \#n_{tot}^k \leq \#n_{tot} \times \#n_{tot}^{\#n_{tot}}$. As any such $M'$ can be generated from $M$, checked or activity, and run on $I$ in time polynomial in the size of the given instance of MLCA, the above is a fixed-parameter tractable algorithm for MLCA relative to parameter-set $\{\#n_{tot}\}$. ∎

**Theorem 42.** $\langle cw, cd \rangle$-MLCA is fixed-parameter tractable.

*Proof.* Follows from the algorithm in the proof of Theorem 41 and the observation that $\#n_{tot} \leq cw \times cd$. ∎

Observe that the results in Theorems 40–42 in combination with Lemmas 1 and 2 suffice to establish the parameterized complexity status of MLCA relative to many subsets of the parameters listed in Table 8.

Let us now consider the polynomial-time cost approximability of MLCA. As MLCA is a minimization problem, we cannot do this using reductions from a maximization problem like CLIQUE. Hence we will instead use a reduction from another minimization problem, namely DS.

**Theorem 43.** If MLCA is polynomial-time tractable then $P = NP$.

*Proof.* Consider the following reduction from DS to MLCA. Given an instance $\langle G = (V, E), k \rangle$ of DS, construct the following instance $\langle M, I, k' \rangle$ of MLCA: Let $M$ be an MLP based on $\#n_{tot,g} = 3|V| + 1$ neurons spread across four layers:

1. **Input layer**: The input vertex neurons $nv_1, nv_2, \ldots nv_{|V|}$, all of which have bias 1.

2. **Hidden vertex neighbourhood layer I**: The vertex neighbourhood AND neurons $nvnA_1, nvnA_2, \ldots nvnA_{|V|}$, where $nvnA_i$ is an $x$-way AND ReLU gates such that $x = |N_C(v_i)|$.

3. **Hidden vertex neighbourhood layer II**: The vertex neighbourhood NOT neurons $nvnN_1, nvnN_2, \ldots nvnN_{|V|}$, all of which are NOT ReLU gates.

4. **Output layer**: The single output neuron $n_{out}$, which is a $|V|$-way AND ReLU gate.

The non-zero weight connections between adjacent layers are as follows:

- Each input vertex neuron $nv_i$, $1 \leq i \leq |V|$, is connected to its input line with weight 0 and to each vertex neighbourhood AND neuron $nvnA_j$ such that $v_i \in N_C(v_j)$ with weight 1.

- Each vertex neighbourhood AND neuron $nvnA_i$, $1 \leq i \leq |V|$, is connected to its corresponding vertex neighbourhood NOT neuron $nvnN_i$ with weight 1.

- Each vertex neighbourhood NOT neuron $nvnN_i$, $1 \leq i \leq |V|$, is connected to the output neuron $n_{out}$ with weight 1.

All other connections between neurons in adjacent layers have weight 0. Finally, let $I$ be the $|V|$-length one-vector and $k' = k$. Observe that this instance of MLCA can be created in time polynomial in the size of the given instance of DS, Moreover, the output behaviour of the neurons in $M$ from the presentation of input $I$ until the output is generated is as follows:

| timestep | neurons (outputs) |
|---|---|
| 0 | — |
| 1 | $nv_1(1), nv_2(1), \ldots nv_{|V|}(1)$ |
| 2 | $nvnA_1(1), nvnA_2(1), \ldots nvnA_{|V|}(1)$ |
| 3 | $nvnN_1(0), nvnN_2(0), \ldots nvnN_{|V|}(0)$ |
| 4 | $n_{out}(0)$ |

We now need to show the correctness of this reduction by proving that the answer for the given instance of DS is "Yes" if and only if the answer for the constructed instance of MLCA is "Yes". We prove the two directions of this if and only if separately as follows:

$\Rightarrow$ : Let $V' = \{v'_1, v'_2, \ldots, v'_k\} \subseteq V$ be a dominating set in $G$ of size $k$ and $N'$ be the $k' = k$-sized subset of the input vertex neurons in $M$ corresponding to the vertices in $V'$. Create MLP $M'$ by ablated in $M$ the neurons in $N'$. As $V'$ is a dominating set, each vertex neighbourhood AND neuron in $M'$ is missing a i-input from at least one input vertex neuron in $N'$, which in turn ensures that each vertex neighbourhood AND neuron in $M'$ has output 0. This in turn ensures that $M$ produces output 1 on input $I$ such that $M(I) = 0 \neq 1 = M'(I)$.

$\Leftarrow$ : Let $N'$ be a $k'' \leq k' = k$-sized subset of the set $N$ of neurons in $M$ whose ablation in $M$ creates an MLP $M'$ such that $M(I) = 0 \neq M'(I)$. As all MLP outputs are stepped to be Boolean, this implies that $M'(I) = 1$. This can only happen if all vertex neighbourhood NOT neurons output 1, which in turn can happen only if all vertex neighbourhood AND gates output 0. As $I = 1^{|V|}$, this can only happen if for each vertex neighbourhood AND neuron, at least one input vertex neuron previously producing a 1-input to that vertex neighbourhood AND neuron has been ablated in creating $M'$. This in turn implies that the $k''$ vertices in $G$ corresponding to the elements of $N'$ form a dominating set of size $k'' \leq k$ for $G$.

As DS is $NP$-hard (Garey & Johnson, 1979), the reduction above establishes that MLCA is also $NP$-hard. The result follows from the definition of $NP$-hardness. ∎

**Theorem 44.** If MLCA has a polynomial-time $c$-approximation algorithm for any constant $c > 0$ then $FPT = W[1]$.

*Proof.* Recall from the proof of correctness of the reduction in the proof of Theorem 43 that a given instance of DS has a dominating set of size $k$ if and only if the constructed instance of MLCA has a subset $N'$ of size $k' = k$ of the neurons in given MLP $M$ such that the ablation in $M$ of the neurons in $N'$ creates an MLP $M'$ such that $M(I) \neq M'(I)$. This implies that, given a polynomial-time $c$-approximation algorithm $A$ for MLCA for some constant $c > 0$, we can create a polynomial-time $c$-approximation algorithm for DS by applying the reduction to the given instance $x$ of DS to construct an instance $x'$ of MLCA, applying $A$ to $x'$ to create an approximate solution $y'$, and then using $y'$ to create an approximate solution $y$ for $x$ that has the same cost as $y'$. The result then follows from Chen & Lin 2019, Corollary 2, which implies that if DS has a polynomial-time $c$-approximation algorithm for any constant $c > 0$ then $FPT = W[1]$. ∎

Note that this theorem also renders MLCA PTAS-inapproximable unless $FPT = W[1]$.

### H.1.2 RESULTS FOR MGCA

Membership in in $\Sigma_2^p$ can be proven via the definition of the polynomial hierarchy and the following alternating quantifier formula:

$$\exists[\mathcal{S} \subseteq \mathcal{M}] \; \forall[\mathbf{x} \in \{0,1\}^{\#n_{in}}] : [\mathcal{M} \setminus \mathcal{S}](\mathbf{x}) \neq \mathcal{M}(\mathbf{x})$$

**Theorem 45.** If MGCA is polynomial-time tractable then $P = NP$.

*Proof.* Observe that in the instance of MLCA constructed by the reduction in the proof of Theorem 39, the input-connection weight 0 and bias 1 of the input neuron force this neuron to output 1 for both of the possible input vectors $(1)$ and $(0)$. Hence, with slight modifications to the proof of reduction correctness, this reduction also establishes the $NP$-hardness of MGCA. ∎

**Theorem 46.** If $\langle cd, \#n_{in}, \#n_{out}, W_{\max}, B_{\max}, k \rangle$-MGCA is fixed-parameter tractable then $FPT = W[1]$.

*Proof.* Observe that in the instance of MGCA constructed in the reduction in the proof of Theorem 45, $\#n_{in} = \#n_{out} = W_{\max} = 1$, $cd = 5$, and $B_{\max}$ and $k$ are function of $k$ in the given instance of CLIQUE. The result then follows from the fact that $\langle k \rangle$-CLIQUE is $W[1]$-hard (Downey & Fellows, 1999). ∎

**Theorem 47.** $\langle \#n_{tot} \rangle$-MGCA is fixed-parameter tractable.

*Proof.* Modify the algorithm in the proof of Theorem 41 such that each created MLP $M$ is checked to ensure that $M(I) \neq M'(I)$ for every possible Boolean input vector of length $\#n_{in}$. As the number of such vectors is $2^{\#n_{in}} \leq 2^{\#n_{tot}}$, the above is a fixed-parameter tractable algorithm for MGCA relative to parameter-set $\{\#n_{tot}\}$. ∎

**Theorem 48.** $\langle cw, cd \rangle$-MGCA is fixed-parameter tractable.

*Proof.* Follows from the algorithm in the proof of Theorem 47 and the observation that $\#n_{tot} \leq cw \times cd$. ∎

Observe that the results in Theorems 46–48 in combination with Lemmas 1 and 2 suffice to establish the parameterized complexity status of MGCA relative to many subsets of the parameters listed in Table 8.

Let us now consider the polynomial-time cost approximability of MGCA. As MGCA is a minimization problem, we cannot do this using reductions from a maximization problem like CLIQUE. Hence we will instead use a reduction from another minimization problem, namely DS.

**Theorem 49.** If MGCA is polynomial-time tractable then $P = NP$.

*Proof.* Observe that in the instance of MLCA constructed by the reduction in the proof of Theorem 43, the input-connection weight 0 and bias 1 of the vertex input neurons force each such neuron to output 1 for input value 0 or 1. Hence, with slight modifications to the proof of reduction correctness, this reduction also establishes the $NP$-hardness of MGCA. ∎

**Theorem 50.** If MGCA has a polynomial-time $c$-approximation algorithm for any constant $c > 0$ then $FPT = W[1]$.

*Proof.* As the reduction in the proof of Theorem 49 is essentially the same as the reduction in the proof of Theorem 43, the result follows by the same reasoning as given in the proof of Theorem 44. ∎

Note that this theorem also renders MGCA PTAS-inapproximable unless $FPT = W[1]$.

## H.2 RESULTS FOR MINIMAL CIRCUIT CLAMPING

Towards proving NP-completeness, we first prove membership and then follow up with hardness. Membership in NP can be proven via the definition of the polynomial hierarchy and the following alternating quantifier formula:

$$\exists [\mathcal{S} \subseteq \mathcal{M}] : \mathcal{M}_{\mathcal{S}}(\mathbf{x}) \neq \mathcal{M}(\mathbf{x})$$

The following hardness results are notable for holding when the given MLP $M$ has only one hidden layer.

### H.2.1 RESULTS FOR MLCC

**Theorem 51.** If MLCC is polynomial-time tractable then $P = NP$.

*Proof.* Consider the following reduction from CLIQUE to MLCC. Given an instance $\langle G = (V, E), k \rangle$ of CLIQUE, construct the following instance $\langle M, I, val, k' \rangle$ of MLCC: Let $M$ be an MLP based on $\#n_{tot} = |V| + |E| + 1$ neurons spread across three layers:

1. **Input vertex layer**: The vertex neurons $nv_1, nv_2, \ldots nv_{|V|}$ (all with bias $-2$).

2. **Hidden edge layer**: The edge neurons $ne_1, ne_2, \ldots ne_{|E|}$ (all with bias $-1$).

3. **Output layer**: The single output neuron $n_{out}$ (bias $-(k(k-1)/2 - 1)$).

Note that this MLP has only one hidden layer. The non-zero weight connections between adjacent layers are as follows:

- Each vertex neuron $nv_i, 1 \leq i \leq |V|$, is connected to each edge neuron whose corresponding edge has an endpoint $v_i$ with weight 1.

- Each edge neuron $ne_i, 1 \leq i \leq |E|$, is connected to the output neuron $n_{out}$ with weight 1.

All other connections between neurons in adjacent layers have weight 0. Finally, let $I = 0^{\#n_{in}}$), $val = 1$, and $k' = k$. Observe that this instance of MLCC can be created in time polynomial in the size of the given instance of CLIQUE. Moreover, the output behaviour of the neurons in $M$ from the presentation of input $I$ until the output is generated is as follows:

| timestep | neurons (outputs) |
|----------|-------------------|
| 0 | — |
| 1 | $nv_1(0), nv_2(0), \ldots nv_{|V|}(0)$ |
| 2 | $ne_1(0), ne_2(0), \ldots ne_{|E|}(0)$ |
| 3 | $n_{out}(0)$ |

We now need to show the correctness of this reduction by proving that the answer for the given instance of CLIQUE is "Yes" if and only if the answer for the constructed instance of MLCC is "Yes". We prove the two directions of this if and only if separately as follows:

$\Rightarrow$ : Let $V' = \{v'_1, v'_2, \ldots, v'_k\} \subseteq V$ be a clique in $G$ of size $k'' \geq k$ and $N'$ be the $k'' \geq k' = k$-sized subset of the input vertex neurons corresponding to the vertices in $V'$. Let $M'$ be the version of $M$ in which all neurons in $N'$ are clamped to value $val = 1$. As $V'$ is a clique of size $k''$, exactly $k''(k''-1)/2 \geq k(k-1)/2$ edge neurons in $M'$ receive the requisite

inputs of 1 on both of their endpoints from the vertex neurons in $N'$. This in turn ensures the output neuron produces output 1. Hence, $M(I) = 0 \neq 1 = M'(I)$.

$\Leftarrow$ : Let $N'$ be a subset of $N$ of size at most $k' = k$ such that for the MLP $M'$ induced by clamping all neurons in $N'$ to value $val = 1$, $M(I) \neq M(I')$. As $M(I) = 0$ and circuit outputs are stepped to be Boolean, $M'(I) = 1$. Given the bias of the output neuron, this can only occur if at least $k(k-1)/2$ edge neurons in $M'$ have output 1 on input $I$, which requires that each of these neurons receives 1 from both of its endpoint vertex neurons. As $I = 0^{\#n_{in}}$, these 1-inputs could only have come from the clamped vertex neurons in $N'$; moreover, there must be exactly $k$ such neurons. This means that the vertices in $G$ corresponding to the vertex neurons in $N'$ must form a clique of size $k$ in $G$.

As CLIQUE is $NP$-hard (Garey & Johnson, 1979), the reduction above establishes that MLCC is also $NP$-hard. The result follows from the definition of $NP$-hardness. ∎

**Theorem 52.** If $\langle cd, \#n_{out}, W_{\max}, B_{\max}, k \rangle$-MLCC is fixed-parameter tractable then $FPT = W[1]$.

*Proof.* Observe that in the instance of MLCC constructed in the reduction in the proof of Theorem 51, $\#n_{out} = W_{\max} = 1$, $cd = 3$, and $B_{\max}$ and $k$ are function of $k$ in the given instance of CLIQUE. The result then follows from the fact that $\langle k \rangle$-CLIQUE is $W[1]$-hard (Downey & Fellows, 1999). ∎

**Theorem 53.** $\langle \#n_{tot} \rangle$-MLCC is fixed-parameter tractable.

*Proof.* Consider the algorithm that generates every possible subset $N'$ of size at most $k$ of the neurons $N$ in MLP $M$ and for each such subset, creates the MLP $M'$ induced from $M$ by clamping the neurons in $N'$ to $val$ and checks if $M'(I) \neq M(I)$. If such a subset is found, return "Yes"; otherwise, return "No". The number of possible subsets $N'$ is at most $k \times \#n_{tot}^k \leq \#n_{tot} \times \#n_{tot}^{\#n_{tot}}$. As any such $M'$ can be generated from $M$ and $M'$ can be run on $I$ in time polynomial in the size of the given instance of MLCC, the above is a fixed-parameter tractable algorithm for MLCC relative to parameter-set $\{\#n_{tot}\}$. ∎

**Theorem 54.** $\langle cw, cd \rangle$-MLCC is fixed-parameter tractable.

*Proof.* Follows from the algorithm in the proof of Theorem 53 and the observation that $\#n_{tot} \leq cw \times cd$. ∎

Observe that the results in Theorems 52–54 in combination with Lemmas 1 and 2 suffice to establish the parameterized complexity status of MLCC relative to many subsets of the parameters listed in Table 8.

Let us now consider the polynomial-time cost approximability of MLCC. As MLCC is a minimization problem, we cannot do this using reductions from a maximization problem like CLIQUE. Hence we will instead use a reduction from another minimization problem, namely DS.

**Theorem 55.** If MLCC is polynomial-time tractable then $P = NP$.

*Proof.* Consider the following reduction from DS to MLCC. Given an instance $\langle G = (V, E), k \rangle$ of DS, construct the following instance $\langle M, I, val, k' \rangle$ of MLCC: Let $M$ be an MLP based on $\#n_{tot,g} = 3|V| + 1$ neurons spread across four layers:

1. **Input layer**: The input vertex neurons $nv_1, nv_2, \ldots nv_{|V|}$, all of which have bias 1.

2. **Hidden vertex neighbourhood layer I**: The vertex neighbourhood AND neurons $nvnA_1, nvnA_2, \ldots nvnA_{|V|}$, where $nvnA_i$ is an $x$-way AND ReLU gates such that $x = |N_C(v_i)|$.

3. **Hidden vertex neighbourhood layer II**: The vertex neighbourhood NOT neurons $nvnN_1, nvnN_2, \ldots nvnN_{|V|}$, all of which are NOT ReLU gates.

4. **Output layer**: The single output neuron $n_{out}$, which is a $|V|$-way AND ReLU gate.

The non-zero weight connections between adjacent layers are as follows:

- Each input vertex neuron $nv_i$, $1 \leq i \leq |V|$, is connected to its input line with weight 0 and to each vertex neighbourhood AND neuron $nvnA_j$ such that $v_i \in N_C(v_j)$ with weight 1.

- Each vertex neighbourhood AND neuron $nvnA_i$, $1 \leq i \leq |V|$, is connected to its corresponding vertex neighbourhood NOT neuron $nvnN_i$ with weight 1.

- Each vertex neighbourhood NOT neuron $nvnN_i$, $1 \leq i \leq |V|$, is connected to the output neuron $n_{out}$ with weight 1.

All other connections between neurons in adjacent layers have weight 0. Finally, let $I$ be the $|V|$-length one-vector, $val = 0$, and $k' = k$. Observe that this instance of MLCC can be created in time polynomial in the size of the given instance of DS, Moreover, the output behaviour of the neurons in $M$ from the presentation of input $I$ until the output is generated is as follows:

| timestep | neurons (outputs) |
|---|---|
| 0 | — |
| 1 | $nvN_1(1), nvN_2(1), \ldots nvN_{|V|}(1)$ |
| 2 | $nvnA_1(1), nvnA_2(1), \ldots nvnA_{|V|}(1)$ |
| 3 | $nvnN_1(0), nvnN_2(0), \ldots nvnN_{|V|}(0)$ |
| 4 | $n_{out}(0)$ |

We now need to show the correctness of this reduction by proving that the answer for the given instance of DS is "Yes" if and only if the answer for the constructed instance of MLCC is "Yes". We prove the two directions of this if and only if separately as follows:

$\Rightarrow$ : Let $V' = \{v'_1, v'_2, \ldots, v'_k\} \subseteq V$ be a dominating set in $G$ of size $k$ and $N'$ be the $k' = k$-sized subset of the input vertex neurons in $M$ corresponding to the vertices in $V'$. Create MLP $M'$ by clamping in $M$ the neurons in $N'$ to $val = 0$. As $V'$ is a dominating set, each vertex neighbourhood AND neuron in $M'$ is now missing a i-input from at least one input vertex neuron in $N'$, which in turn ensures that each vertex neighbourhood AND neuron in $M'$ has output 0. This in turn ensures that $M$ produces output 1 on input $I$ such that $M(I) = 0 \neq 1 = M'(I)$..

$\Leftarrow$ : Let $N'$ be a $k'' \leq k' = k$-sized subset of the set $N$ of neurons in $M$ whose clamping to $val = 0$ in $M$ creates an MLP $M'$ such that $M(I) = 0 \neq M'(I)$. As all MLP outputs are stepped to be Boolean, this implies that $M'(I) = 1$. This can only happen if all vertex neighbourhood NOT neurons output 1, which in turn can happen only if all vertex neighbourhood AND gates output 0. As $I = 1^{|V|}$, this can only happen if for each vertex neighbourhood AND neuron, at least one input vertex neuron previously producing a 1-input to that vertex neighbourhood AND neuron has been clamped to 0 in creating $M'$. This in turn implies that the $k''$ vertices in $G$ corresponding to the elements of $N'$ form a dominating set of size $k'' \leq k$ for $G$.

As DS is $NP$-hard (Garey & Johnson, 1979), the reduction above establishes that MLCC is also $NP$-hard. The result follows from the definition of $NP$-hardness. ∎

**Theorem 56.** If MLCC has a polynomial-time $c$-approximation algorithm for any constant $c > 0$ then $FPT = W[1]$.

*Proof.* Recall from the proof of correctness of the reduction in the proof of Theorem 55 that a given instance of DS has a dominating set of size $k$ if and only if the constructed instance of MLCC has a subset $N'$ of size $k' = k$ of the neurons in given MLP $M$ such that the clamping to $val = 0$ in $M$ of the neurons in $N'$ creates an MLP $M'$ such that $M(I) \neq M'(I)$. This implies that, given a polynomial-time $c$-approximation algorithm $A$ for MLCC for some constant $c > 0$, we can create a polynomial-time $c$-approximation algorithm for DS by applying the reduction to the given instance $x$ of DS to construct an instance $x'$ of MLCC, applying $A$ to $x'$ to create an approximate solution $y'$, and then using $y'$ to create an approximate solution $y$ for $x$ that has the same cost as $y'$. The result then follows from Chen & Lin 2019, Corollary 2, which implies that if DS has a polynomial-time $c$-approximation algorithm for any constant $c > 0$ then $FPT = W[1]$. ∎

Note that this theorem also renders MLCC PTAS-inapproximable unless $FPT = W[1]$.

### H.2.2 RESULTS FOR MGCC

Membership in in $\Sigma_2^p$ can be proven via the definition of the polynomial hierarchy and the following alternating quantifier formula:

$$\exists[\mathcal{S} \subseteq \mathcal{M}] \, \forall[\mathbf{x} \in \{0,1\}^{\#n_{in}}] : \mathcal{M}_{\mathcal{S}}(\mathbf{x}) \neq \mathcal{M}(\mathbf{x})$$

**Theorem 57.** If MGCC is polynomial-time tractable then $P = NP$.

*Proof.* Observe that in the instance of MLCC constructed by the reduction in the proof of Theorem 51, the biases of $-2$ in the input vertex neurons force these neurons to map any given Boolean input vector onto $0^{\#n_{in}}$. Hence, with slight modifications to the proof of reduction correctness, this reduction also establishes the $NP$-hardness of MGCC. ∎

**Theorem 58.** If $\langle cd, \#n_{out}, W_{\max}, B_{\max}, k \rangle$-MGCC is fixed-parameter tractable then $FPT = W[1]$.

*Proof.* Observe that in the instance of MGCC constructed in the reduction in the proof of Theorem 57, $\#n_{out} = W_{\max} = 1$, $cd = 3$, and $B_{\max}$ and $k$ are function of $k$ in the given instance of CLIQUE. The result then follows from the fact that $\langle k \rangle$-CLIQUE is $W[1]$-hard (Downey & Fellows, 1999). ∎

**Theorem 59.** $\langle \#n_{tot} \rangle$-MLCC is fixed-parameter tractable.

*Proof.* Modify the algorithm in the proof of Theorem 53 such that each created MLP $M$ is checked to ensure that $M(I) \neq M'(I)$ for every possible Boolean input vector of length $\#n_{in}$. As the number of such vectors is $2^{\#n_{in}} \leq 2^{\#n_{tot}}$, the above is a fixed-parameter tractable algorithm for MGCC relative to parameter-set $\{\#n_{tot}\}$. ∎

**Theorem 60.** $\langle cw, cd \rangle$-MGCC is fixed-parameter tractable.

*Proof.* Follows from the algorithm in the proof of Theorem 59 and the observation that $\#n_{tot} \leq cw \times cd$. ∎

Observe that the results in Theorems 58–60 in combination with Lemmas 1 and 2 suffice to establish the parameterized complexity status of MGCC relative to many subsets of the parameters listed in Table 8.

Let us now consider the polynomial-time cost approximability of MGCC. As MGCC is a minimization problem, we cannot do this using reductions from a maximization problem like CLIQUE. Hence we will instead use a reduction from another minimization problem, namely DS.

**Theorem 61.** If MGCC is polynomial-time tractable then $P = NP$.

*Proof.* Observe that in the instance of MLCC constructed by the reduction in the proof of Theorem 55, the input-connection weight 0 and bias 1 of the vertex input neurons force each such neuron to output 1 for input value 0 or 1. Hence, with slight modifications to the proof of reduction correctness, this reduction also establishes the $NP$-hardness of MGCC. ∎

**Theorem 62.** If MGCC has a polynomial-time $c$-approximation algorithm for any constant $c > 0$ then $FPT = W[1]$.

*Proof.* As the reduction in the proof of Theorem 61 is essentially the same as the reduction in the proof of Theorem 55, the result follows by the same reasoning as given in the proof of Theorem 56. ∎

Note that this theorem also renders MGCC PTAS-inapproximable unless $FPT = W[1]$.

## I  CIRCUIT PATCHING PROBLEM

MINIMUM LOCAL CIRCUIT PATCHING (MLCP)
*Input*: A multi-layer perceptron $M$ of depth $cd$ with $\#n_{tot}$ neurons and maximum layer width $cw$, connection-value matrices $W_1, W_2, \ldots, W_{cd}$, neuron bias vector $B$, Boolean input vectors $x$ and $y$ of length $\#n_{in}$, and a positive integer $k$ such that $1 \leq k \leq (\#n_{tot} - (\#n_{in} + \#n_{out}))$.
*Question*: Is there a subset $C$, $|C| \leq k$, of the internal neurons in $M$ such that for the MLP $M'$ created when $M$ is $y$-patched wrt $C$, i.e., $M'$ is created when $M/C$ is patched with activations from $M(x)$ and $C$ is patched with activations from $M(y)$, $M'(x) = M(y)$?

MINIMUM GLOBAL CIRCUIT PATCHING (MGCP)
*Input*: A multi-layer perceptron $M$ of depth $cd$ with $\#n_{tot}$ neurons and maximum layer width $cw$, connection-value matrices $W_1, W_2, \ldots, W_{cd}$, neuron bias vector $B$, Boolean input vector $y$ of length $\#n_{in}$, and a positive integer $k$ such that $1 \leq k \leq (\#n_{tot} - (\#n_{in} + \#n_{out}))$.
*Question*: Is there a subset $C$, $|C| \leq k$, of the internal neurons in $M$ such that, for all possible input vectors x, for the MLP $M'$ created when $M$ is $y$-patched wrt $C$, i.e., $M'$ is created when $M/C$ is patched with activations from $M(x)$ and $C$ is patched with activations from $M(y)$, $M'(x) = M(y)$?

Following Barceló et al. 2020, page 4, all neurons in $M$ use the ReLU activation function and the output $x$ of each output neuron is stepped as necessary to be Boolean, i.e, $step(x) = 0$ if $x \leq 0$ and is 1 otherwise.

For a graph $G = (V, E)$, we shall assume an ordering on the vertices and edges in $V$ and $E$, respectively. For each vertex $v \in V$, let the complete neighbourhood $N_C(v)$ of $v$ be the set composed of $v$ and the set of all vertices in $G$ that are adjacent to $v$ by a single edge, i.e., $v \cup \{u \mid u \in V \text{ and } (u, v) \in E\}$.

We will prove various classical and parameterized results for MLCP and MGCP using reductions from DOMINATING SET. The parameterized results are proved relative to the parameters in Table 9. Our reductions (Theorems 63 and 67) uses specialized ReLU logic gates described in Barceló et al. 2020, Lemma 13. These gates assume Boolean neuron input and output values of 0 and 1 and are structured as follows:

1. NOT ReLU gate: A ReLU gate with one input connection weight of value $-1$ and a bias of 1. This gate has output 1 if the input is 0 and 0 otherwise.

2. $n$-way AND ReLU gate: A ReLU gate with $n$ input connection weights of value 1 and a bias of $-(n - 1)$. This gate has output 1 if all inputs have value 1 and 0 otherwise.

3. $n$-way OR ReLU gate: A combination of an $n$-way AND ReLU gate with NOT ReLU gates on all of its inputs and a NOT ReLU gate on its output that uses DeMorgan's Second Law to implement $(x_1 \vee x_2 \vee \ldots x_n)$ as $\neg(\neg x_1 \wedge \neg x_2 \wedge \ldots \neg x_n)$. This gate has output 1 if any input has value 1 and 0 otherwise.

Table 9: Parameters for the minimum circuit patching problem.

| Parameter | Description |
|---|---|
| $cd$ | # layers in given MLP |
| $cw$ | max # neurons in layer in given MLP |
| $\#n_{tot}$ | total # neurons in given MLP |
| $\#n_{in}$ | # input neurons in given MLP |
| $\#n_{out}$ | # output neurons in given MLP |
| $B_{\max}$ | max neuron bias in given MLP |
| $W_{\max}$ | max connection weight in given MLP |
| $k$ | Size of requested patching-subset of $M$ |

## I.1 RESULTS FOR MLCP

Towards proving NP-completeness, we first prove membership and then follow up with hardness. Membership in NP can be proven via the definition of the polynomial hierarchy and the following alternating quantifier formula:

$$\exists [\mathcal{S} \subseteq \mathcal{M}] : \mathcal{M}_{\mathcal{S}}(\mathbf{x}) = \mathcal{M}(\mathbf{y})$$

**Theorem 63.** If MLCP is polynomial-time tractable then $P = NP$.

*Proof.* Consider the following reduction from DS to MLCP adapted from the reduction from DS to MSR in Theorem 103. Given an instance $\langle G = (V, E), k \rangle$ of DS, construct the following instance $\langle M, x, y, k' \rangle$ of MLCP: Let $M$ be an MLP based on $\#n_{tot} = 4|V| + 1$ neurons spread across five layers:

1. **Input layer**: The input vertex neurons $nv_1, nv_2, \ldots nv_{|V|}$, all of which have bias 0.

2. **Hidden vertex layer**: The hidden vertex neurons $nhv_1, nhv_2, \ldots nhv_{|V|}$, all of which are identity ReLU gates with bias 0.

3. **Hidden vertex neighbourhood layer I**: The vertex neighbourhood AND neurons $nvnA_1, nvnA_2, \ldots nvnA_{|V|}$, where $nvnA_i$ is an $x$-way AND ReLU gates such that $x = |N_C(v_i)|$.

4. **Hidden vertex neighbourhood layer II**: The vertex neighbourhood NOT neurons $nvnN_1, nvnN_2, \ldots nvnN_{|V|}$, all of which are NOT ReLU gates.

5. **Output layer**: The single output neuron $n_{out}$, which is a $|V|$-way AND ReLU gate.

The non-zero weight connections between adjacent layers are as follows:

- Each input vertex neuron $nv_i$, $1 \leq i \leq |V|$, is connected to its associated hidden vertex neuron $nhv_i$ with weight 1.

- Each hidden vertex neuron $nhv_i$, $1 \leq i \leq |V|$, is connected to each vertex neighbourhood AND neuron $nvnA_j$ such that $v_i \in N_C(v_j)$ with weight 1.

- Each vertex neighbourhood AND neuron $nvnA_i$, $1 \leq i \leq |V|$, is connected to its corresponding vertex neighbourhood NOT neuron $nvnN_i$ with weight 1.

- Each vertex neighbourhood NOT neuron $nvnN_i$, $1 \leq i \leq |V|$, is connected to the output neuron $n_{out}$ with weight 1.

All other connections between neurons in adjacent layers have weight 0. Finally, let $x$ and $y$ be the $|V|$-length one- and zero-vectors and $k' = k$. Observe that this instance of MLCP can be created in time polynomial in the size of the given instance of DS, Moreover, the output behaviour of the neurons in $M$ from the presentation of input $x$ until the output is generated is

| timestep | neurons (outputs) |
|---|---|
| 0 | — |
| 1 | $nv_1(1), nv_2(1), \ldots nv_{|V|}(1)$ |
| 2 | $nhv_1(1), nhv_2(1), \ldots nhv_{|V|}(1)$ |
| 3 | $nvnA_1(1), nvnA_2(1), \ldots nvnA_{|V|}(1)$ |
| 4 | $nvnN_1(0), nvnN_2(0), \ldots nvnN_{|V|}(0)$ |
| 5 | $n_{out}(0)$ |

and the output behaviour of the neurons in $M$ from the presentation of input $y$ until the output is generated is

| timestep | neurons (outputs) |
|---|---|
| 0 | — |
| 1 | $nv_1(0), nv_2(0), \ldots nv_{|V|}(0)$ |
| 2 | $nhv_1(0), nhv_2(0), \ldots nhv_{|V|}(0)$ |
| 3 | $nvnA_1(0), nvnA_2(0), \ldots nvnA_{|V|}(0)$ |
| 4 | $nvnN_1(1), nvnN_2(1), \ldots nvnN_{|V|}(1)$ |
| 5 | $n_{out}(1)$ |

We now need to show the correctness of this reduction by proving that the answer for the given instance of DS is "Yes" if and only if the answer for the constructed instance of MLCP is "Yes". We prove the two directions of this if and only if separately as follows:

$\Rightarrow$ : Let $V' = \{v'_1, v'_2, \ldots, v'_k\} \subseteq V$ be a dominating set in $G$ of size $k$ and $C$ be the $k' = k$-sized subset of the hidden vertex neurons in $M$ corresponding to the vertices in $V'$. As $V'$ is a dominating set, each vertex neighbourhood AND neuron receives input 0 from at least one hidden vertex neuron in $C$ when $M$ is $y$-patched wrt $C$. This ensures that each vertex neighbourhood AND neuron has output 0, which in turn ensures that each vertex neighbourhood NOT neuron has output 1 and for $M'$ created by $y$-patching $M$ wrt $C$, $M'(x) = M(y) = 1$.

$\Leftarrow$ : Let $C$ be a $k' = k$-sized subset of the internal neurons of $M$ such that when $M'$ is created by $y$-patching $M$ wrt $C$, $M'(x) = M(y) = 1$. The output of $M'$ on $X$ can be 1 (and hence equal to the output of $M$ on $y$) only if all vertex neighbourhood NOT neurons output 1, which in turn can happen only if all vertex neighbourhood AND gates output 0. However, as all elements of $x$ have value 1, this means that each vertex neighbourhood AND neuron must be connected to at least one patched hidden vertex neuron (all of which have output 0 courtesy of $y$), which in turn implies that the $k' = k$ vertices in $G$ corresponding to the patched hidden vertex neurons in $C$ form a dominating set of size $k$ for $G$.

As DS is $NP$-hard (Garey & Johnson, 1979), the reduction above establishes that MLCP is also $NP$-hard. The result follows from the definition of $NP$-hardness. ∎

**Theorem 64.** If $\langle cd, \#n_{out}, W_{\max}, B_{\max}, k \rangle$-MLCP is fixed-parameter tractable then $FPT = W[1]$.

*Proof.* Observe that in the instance of MLCP constructed in the reduction in the proof of Theorem 63, $\#n_{out} = W_{\max} = 1$, $cd = 4$, and $B_{\max}$ and $k$ are function of $k$ in the given instance of DS. The result then follows from the facts that $\langle k \rangle$-DS is $W[2]$-hard (Downey & Fellows, 1999) and $W[1] \subseteq W[2]$. ∎

**Theorem 65.** $\langle \#n_{tot} \rangle$-MLCP is fixed-parameter tractable.

*Proof.* Consider the algorithm that generates all possible subset $C$ of the internal neurons in $M$ and for each such subset, checks if $M'$ created by $y$-patching $M$ wrt $C$ is such that $M'(x) = M(y)$. If such a $C$ is found, return "Yes"; otherwise, return "No". The number of possible subsets $C$ is at most $2^{(\#n_{tot}}$. Given this, as $M$ can be patched relative to $C$ and run on $x$ and $y$ in time polynomial in the size of the given instance of MLCP, the above is a fixed-parameter tractable algorithm for MLCP relative to parameter-set $\{\#n_{tot}\}$. ∎

Observe that the results in Theorems 64 and 65 in combination with Lemmas 1 and 2 suffice to establish the parameterized complexity status of MLCP relative to many subset of the parameters listed in Table 9.

Let us now consider the polynomial-time cost approximability of MLCP.

**Theorem 66.** If MLCP has a polynomial-time $c$-approximation algorithm for any constant $c > 0$ then $FPT = W[1]$.

*Proof.* Recall from the proof of correctness of the reduction in the proof of Theorem 63 that a given instance of DS has a dominating set of size $k$ if and only if the constructed instance of $MLCP$ has a subset $C$ of the internal neurons in $M$ of size $k' = k$ such that for the MLP $M'$ created from $M$ by $y$-patching $M$ wrt $C$, $M'(x) = M(y)$ This implies that, given a polynomial-time $c$-approximation algorithm $A$ for MLCP for some constant $c > 0$, we can create a polynomial-time $c$-approximation algorithm for DS by applying the reduction to the given instance $I$ of DS to construct an instance $I'$ of MLCP, applying $A$ to $I'$ to create an approximate solution $S'$, and then using $S'$ to create an approximate solution $S$ for $I$ that has the same cost as $S'$. The result then follows from Chen & Lin 2019, Corollary 2, which implies that if DS has a polynomial-time $c$-approximation algorithm for any constant $c > 0$ then $FPT = W[1]$. ∎

Note that this theorem also renders MLCP PTAS-inapproximable unless $FPT = W[1]$.

## I.2 RESULTS FOR MGCP

Membership in in $\Sigma_2^p$ can be proven via the definition of the polynomial hierarchy and the following alternating quantifier formula:

$$\exists[\mathcal{S} \subseteq \mathcal{M}] \, \forall[\mathbf{x} \in \{0,1\}^{\#n_{in}}] : \mathcal{M}_{\mathcal{S}}(\mathbf{x}) = \mathcal{M}(\mathbf{y})$$

**Theorem 67.** If MGCP is polynomial-time tractable then $P = NP$.

*Proof.* Modify the reduction in the proof of Theorem 63 such that each hidden vertex neuron has input weight 0 and bias 1; this will force all hidden vertex neurons to output 1 for all input vectors $x$ instead of just when $x$ is the all-one vector. Hence, with slight modifications to the proof of reduction correctness for the reduction in the proof of Theorem 63, this modified reduction establishes the $NP$-hardness of MGCP. ∎

**Theorem 68.** If $\langle cd, \#n_{out}, W_{\max}, B_{\max}, k \rangle$-MGCP is fixed-parameter tractable then $FPT = W[1]$.

*Proof.* Observe that in the instance of MGCP constructed in the reduction in the proof of Theorem 67, $\#n_{out} = W_{\max} = 1$, $cd = 4$, and $B_{\max}$ and $k$ are function of $k$ in the given instance of DS. The result then follows from the facts that $\langle k \rangle$-DS is $W[2]$-hard (Downey & Fellows, 1999) and $W[1] \subseteq W[2]$. ∎

**Theorem 69.** $\langle \#n_{tot} \rangle$-MGCP is fixed-parameter tractable.

*Proof.* Modify the algorithm in the proof of Theorem 65 such that each circuit $M'$ created by $y$-patching $M$ is checked to ensure that $M'(x) = M(y)$ for every possible Boolean input vector $x$ of length $\#n_{in}$. As the number of such vectors is $2^{\#n_{in}} < 2^{\#n_{tot}}$, the above is a fixed-parameter tractable algorithm for MGCP relative to parameter-set $\{\#n_{tot}\}$. ∎

Observe that the results in Theorems 68 and 69 in combination with Lemmas 1 and 2 suffice to establish the parameterized complexity status of MGCP relative to many subset of the parameters listed in Table 9.

Let us now consider the polynomial-time cost approximability of MGCP.

**Theorem 70.** If MGCP has a polynomial-time $c$-approximation algorithm for any constant $c > 0$ then $FPT = W[1]$.

*Proof.* As the reduction in the proof of Theorem 67 is essentially the same as the reduction in the proof of Theorem 63, the result follows by the same reasoning as given in the proof of Theorem 66. ∎

Note that this theorem also renders MGCP PTAS-inapproximable unless $FPT = W[1]$.

## J QUASI-MINIMAL CIRCUIT PATCHING PROBLEM

QUASI-MINIMAL CIRCUIT PATCHING (QMCP)
*Input*: A multi-layer perceptron $M$ of depth $cd$ with $\#n_{tot}$ neurons and maximum layer width $cw$, connection-value matrices $W_1, W_2, \ldots, W_{cd}$, neuron bias vector $B$, an input vector $\mathbf{y}$ and a set $\mathcal{X}$ of input vectors of length $\#n_{in}$.
*Output*: a subset $\mathcal{C}$ in $\mathcal{M}$ and a neuron $v \in \mathcal{C}$, such that for the $\mathcal{M}^*$ induced by patching $\mathcal{C}$ with activations from $\mathcal{M}(\mathbf{y})$ and $\mathcal{M} \setminus \mathcal{C}$ with activations from $\mathcal{M}(\mathbf{x})$, $\forall_{\mathbf{x} \in \mathcal{X}} : \mathcal{M}^*(\mathbf{x}) = \mathcal{M}(\mathbf{y})$, and for $\mathcal{M}'$ induced by patching identically except for $v \in \mathcal{C}$, $\exists_{\mathbf{x} \in \mathcal{X}} : \mathcal{M}'(\mathbf{x}) \neq \mathcal{M}(\mathbf{y})$.

**Theorem 71.** QMCP is in PTIME (i.e., polynomial-time tractable).

*Proof.* Consider the following algorithm for QMCP. Build a sequence of MLPs by taking $\mathcal{M}$ with all neurons labeled 0, and generating subsequent $\mathcal{M}_i$ in the sequence by labeling an additional neuron with 1 each time (this choice can be based on any heuristic strategy, for instance, one based on gradients). The first MLP, $\mathcal{M}_1$, obtained by patching all neurons labeled 1 (i.e., none) is such that $\mathcal{M}_1(x) \neq \mathcal{M}(y)$, and the last $\mathcal{M}_n$ is guaranteed to give $\mathcal{M}_n(x) = \mathcal{M}(y)$ because all neurons are patched. Label the first MLP NO, and the last YES. Perform a variant of binary search on the sequence as follows. Evaluate the $\mathcal{M}_\rangle$ halfway between NO and YES while patching all its neurons labeled 1. If it satisfies the condition, label it YES, and repeat the same strategy with the sequence starting from the first $\mathcal{M}_i$ until the YES just labeled. If it does *not* satisfy the condition, label it NO and repeat the same strategy with the sequence starting from the NO just labeled until the YES at the end of the original sequence. This iterative procedure halves the sequence each time. Halt when you find two adjacent $\langle$NO, YES$\rangle$ patched networks (guaranteed to exist), and return the patched neuron set of the YES network V and the single neuron difference between YES and NO (the breaking point), $v \in V$. The complexity of this algorithm is roughly $O(n \log n)$.

∎

## K CIRCUIT ROBUSTNESS PROBLEM

**Definition 17.** Given an MLP $M$, a subset $H$ of the elements in $M$, an integer $k \leq |H|$, and an input $I$ to $M$, $M$ is $k$-robust relative to $H$ for $I$ if for each subset $H' \subseteq H$, $|H'| \leq k$, $M(I) = (M/H')(I)$.

Table 10: Parameters for the minimum circuit robustness problem.

| Parameter | Description | Appl. |
|---|---|---|
| $cd$ | # layers in given MLP | All |
| $cw$ | max # neurons in layer in given MLP | All |
| $\#n_{tot}$ | total # neurons in given MLP | All |
| $\#n_{in}$ | # input neurons in given MLP | All |
| $\#n_{out}$ | # output neurons in given MLP | All |
| $B_{\max}$ | max neuron bias in given MLP | All |
| $W_{\max}$ | max connection weight in given MLP | All |
| $k$ | Requested level of robustness | All |
| $|H|$ | Size of investigated region | M{L,G}CR |

MAXIMUM LOCAL CIRCUIT ROBUSTNESS (MLCR)
*Input*: A multi-layer perceptron $M$ of depth $cd$ with $\#n_{tot}$ neurons and maximum layer width $cw$, connection-value matrices $W_1, W_2, \ldots, W_{cd}$, neuron bias vector $B$, a subset $H$ of the neurons in $M$, a Boolean input vector $I$ of length $\#n_{in}$, and a positive integer $k$ such that $1 \leq k \leq |H|$.
*Question*: Is $M$ $k$-robust relative to $H$ for $I$?

RESTRICTED MAXIMUM LOCAL CIRCUIT ROBUSTNESS (MLCR$^*$)
*Input*: A multi-layer perceptron $M$ of depth $cd$ with $\#n_{tot}$ neurons and maximum layer width $cw$, connection-value matrices $W_1, W_2, \ldots, W_{cd}$, neuron bias vector $B$, a Boolean input vector $I$ of length $\#n_{in}$, and a positive integer $k$ such that $1 \leq k \leq |M|$.
*Question*: Is $M$ $k$-robust relative to $H = M$ for $I$?

MAXIMUM GLOBAL CIRCUIT ROBUSTNESS (MGCR)
*Input*: A multi-layer perceptron $M$ of depth $cd$ with $\#n_{tot}$ neurons and maximum layer width $cw$, connection-value matrices $W_1, W_2, \ldots, W_{cd}$, neuron bias vector $B$, a subset $H$ of the neurons in $M$, and a positive integer $k$ such that $1 \leq k \leq |H|$.
*Question*: Is $M$ $k$-robust relative to $H$ for every possible Boolean input vector $I$ of length $\#n_{in}$?

RESTRICTED MAXIMUM GLOBAL CIRCUIT ROBUSTNESS (MGCR$^*$)
*Input*: A multi-layer perceptron $M$ of depth $cd$ with $\#n_{tot}$ neurons and maximum layer width $cw$, connection-value matrices $W_1, W_2, \ldots, W_{cd}$, neuron bias vector $B$, and a positive integer $k$ such that $1 \leq k \leq |M|$.
*Question*: Is $M$ $k$-robust relative to $H = M$ for every possible Boolean input vector $I$ of length $\#n_{in}$?

Following Barceló et al. 2020, page 4, all neurons in $M$ use the ReLU activation function and the output $x$ of each output neuron is stepped as necessary to be Boolean, i.e, $step(x) = 0$ if $x \leq 0$ and is 1 otherwise.

We will use previous results for MINIMUM LOCAL/GLOBAL CIRCUIT ABLATION, CLIQUE and VERTEX COVER to prove our results for the problems above.

For a graph $G = (V, E)$, we shall assume an ordering on the vertices and edges in $V$ and $E$, respectively.

We will prove various classical and parameterized results for MLCR, MLCR$^*$, MGCR, and MGCR$^*$. The parameterized results are proved relative to the parameters in Table 10. Lemmas 1 and 2 will be useful in deriving additional parameterized results from proved ones.

Several of our proofs involve problems that are hard for $coW[1]$ and $coNP$, the complement classes of $W[1]$ and $NP$ (Garey & Johnson, 1979, Section 7.1). Given a decision problem **X** let co-**X** be the complement problem in which all instances answers are switched.

**Observation 3.** MLCR* is the complement of MLCA.

*Proof.* Let $X = \langle M, I, k \rangle$ be a shared input of MLCR* and MLCA. If the the answer to MLCA on input $X$ is "Yes", this means that there is a subset $H$ of size $\leq k$ of the neurons of $M$ that can be ablated such that $M(I) \neq (M/H)(I)$. This implies that the answer to MLCR* on input $X$ is "No". Conversely, if the answer to MLCA on input $X$ is "No", this means that there is no subset $H$ of size $\leq k$ of the neurons of $M$ that can be ablated such that $M(I) \neq (M/H)(I)$. This implies that the answer to MLCR* on input $X$ is "Yes'. The observation follows from the definition of complement problem. ∎

The following lemmas will be of use in the derivation and interpretation of results involving complement problems and classes.

**Lemma 5.** (Garey & Johnson, 1979, Section 7.1) Given a decision problem **X**, if **X** is $NP$-hard then co-**X** is $coNP$-hard.

**Lemma 6.** Given a decision problem **X**, if **X** is $coNP$-hard and **X** is polynomial-time solvable then $P = NP$.

*Proof.* Suppose decision problem **X** is $coNP$-hard and solvable in polynomial-time by algorithm $A$. By the definition of problem, class hardness, for every problem **Y** in $coNP$ there is a polynomial-time many-one reduction $\Pi$ from **Y** to **X**; let $A_\Pi$ be the polynomial-time algorithm encoded in $\Pi$. We can create a polynomial-time algorithm $A'$ for co-**Y** by running $A_\Pi$ on a given input, running $A$, and then complementing the produced output, i.e., "Yes" $\Rightarrow$ "No" and "No" $\Rightarrow$ "Yes". However, as co**X** is $NP$-hard by Lemma 5, this implies that $P = NP$. ∎

**Lemma 7.** (Flum & Grohe, 2006, Lemma 8.23) Let $\mathcal{C}$ be a parameterized complexity class. Given a parameterized decision problem **X**, if **X** is $\mathcal{C}$-hard then co-**X** is $co\mathcal{C}$-hard.

**Lemma 8.** Given a parameterized decision problem **X**, if **X** is fixed-parameter tractable then co-**X** is fixed-parameter tractable.

*Proof.* Given a fixed-parameter tractable algorithm $A$ for **X**, we can create a fixed-parameter tractable algorithm $A'$ for co-**X** by running $A$ on a given input and then complementing the produced output, i.e., "Yes" $\Rightarrow$ "No" and "No" $\Rightarrow$ "Yes". ∎

**Lemma 9.** Given a parameterized decision problem **X**, if **X** is $coW[1]$-hard and fixed-parameter tractable then $FPT = W[1]$.

*Proof.* Suppose decision problem **X** is $coW[1]$-hard and solvable in polynomial-time by algorithm $A$. By the definition of problem class hardness, for every problem **Y** in $coW[1]$ there is a parameterized reduction $\Pi$ from **Y** to **X**; let $A_\Pi$ be the fixed-parameter tractable algorithm encoded in $\Pi$. We can create a polynomial-time algorithm $A'$ for co-**Y** by running $A_\Pi$ on a given input, running $A$, and then complementing the produced output, i.e., "Yes" $\Rightarrow$ "No" and "No" $\Rightarrow$ "Yes". However, as co**X** is $W[1]$-hard by Lemma 7, this implies that $P = NP$. ∎

We will also be deriving polynomial-time inapproximability results for optimization versions of MLCR, MLCR*, MGCR, and MGCR*, i.e.,

- Max-MLCR, which asks for the maximum value $k$ such that $M$ is $k$-robust relative to $H$ for $I$.

- Max-MLCR*, which asks for the maximum value $k$ such that $M$ is $k$-robust relative to $H = M$ for $I$.

- Max-MGCR, which asks for the maximum value $k$ such that $M$ is $k$-robust relative to $H$ for every possible Boolean input vector $I$ of length $\#n_{in}$

- Max-MGCR$^*$, which asks for the maximum value $k$ such that $M$ is $k$-robust relative to $H = M$ for every possible Boolean input vector $I$ of length $\#n_{in}$

The derivation of such results is complicated by both the $coNP$-hardness of the decision versions of these problems (and the scarcity of inapproximability results for $coNP$-hard problems which one could transfer to our problems by $L$-reductions) and the fact that the optimization versions of our problems are evaluation problems that return numbers rather than graph structures (which makes the use of instance-copy and gap inapproximability proof techniques [Garey & Johnson 1979, Chapter 6] extremely difficult). We shall sidestep many of these issues by deriving our results within the $OptP$ framework for analyzing evaluation problems developed in Krentel 1988; Gasarch et al. 1995. In particular, we shall find the following of use.

**Definition 18.** (Adapted from (Krentel, 1988, page 493)) Let $f, g : \Sigma^* \to \mathcal{Z}$. A **metric reduction** from $f$ to $g$ is a pair $(T_1, T_2)$ of polynomial-time computable functions where $T_1 : \Sigma^* \to \Sigma^*$ and $T_2 : \Sigma^* \times \mathcal{Z} \to \mathcal{Z}$ such that $f(x) = T_2(x, g(T_1(x)))$ for all $x \in \Sigma^*$.

**Lemma 10.** (Corollary of (Krentel, 1988, Theorem 4.3)) Given an evaluation problem $\Pi$ that is $OptP[O(\log n)]$-hard under metric reductions, if $\Pi$ has a $c$-additive approximation algorithm for some $c \in o(poly)$[4] then $P = NP$.

Some of our metric reductions use specialized ReLU logic gates described in Barceló et al. 2020, Lemma 13. These gates assume Boolean neuron input and output values of 0 and 1 and are structured as follows:

1. NOT ReLU gate: A ReLU gate with one input connection weight of value $-1$ and a bias of 1. This gate has output 1 if the input is 0 and 0 otherwise.

2. $n$-way AND ReLU gate: A ReLU gate with $n$ input connection weights of value 1 and a bias of $-(n-1)$. This gate has output 1 if all inputs have value 1 and 0 otherwise.

3. $n$-way OR ReLU gate: A combination of an $n$-way AND ReLU gate with NOT ReLU gates on all of its inputs and a NOT ReLU gate on its output that uses DeMorgan's Second Law to implement $(x_1 \vee x_2 \vee \ldots x_n)$ as $\neg(\neg x_1 \wedge \neg x_2 \wedge \ldots \neg x_n)$. This gate has output 1 if any input has value 1 and 0 otherwise.

The hardness (inapproximability) results in this section hold when the given MLP $M$ has three (six) hidden layers.

## K.1 RESULTS FOR MLCR-SPECIAL AND MLCR

Let us first consider problem MLCR$^*$.

**Theorem 72.** If MLCR$^*$ is polynomial-time tractable then $P = NP$.

*Proof.* As MLCR$^*$ is the complement of MLCA by Observation 3 and MLCA is $NP$-hard by the proof of Theorem 39, Lemma 5 implies that MLCR$^*$ is $coNP$-hard. The result then follows from Lemma 6. ∎

**Theorem 73.** If $\langle cd, \#n_{in}, \#n_{out}, W_{\max}, B_{\max}, k \rangle$-MLCR$^*$ is fixed-parameter tractable then $FPT = W[1]$.

*Proof.* The $coW[1]$-hardness of $\langle cd, \#n_{in}, \#n_{out}, W_{\max}, B_{\max}, k \rangle$-MLCR$^*$ follows from the $W[1]$-hardness of $\langle cd, \#n_{in}, \#n_{out}, W_{\max}, B_{\max}, k \rangle$-MLCA (Theorem 40), Observation 3, and Lemma 7. The result then follows from Lemma 9. ∎

**Theorem 74.** $\langle \#n_{tot} \rangle$-MLCR$^*$ is fixed-parameter tractable.

*Proof.* The result follows from the fixed-parameter tractability of $\langle \#n_{tot} \rangle$-MLCA Theorem 41, Observation 3, and Lemma 8. ∎

---

[4] Note that $o(poly)$ is the set of all functions $f$ that are strictly upper bounded by all polynomials of $n$, i.e., $f(n) \le c \times g(n)$ for $n \ge n_0$ for all $c > 0$ and $g(n) \in \cup_k n^k = n^{O(1)}$.

**Theorem 75.** $\langle cw, cd \rangle$-MLCR* is fixed-parameter tractable.

*Proof.* The result follows from the fixed-parameter tractability of $\langle cw, cd \rangle$-MLCA Theorem 42, Observation 3, and Lemma 8. ∎

Observe that the results in Theorems 73–75 in combination with Lemmas 1 and 2 suffice to establish the parameterized complexity status of MLCR* relative to many subsets of the parameters listed in Table 10.

We can derive a polynomial-time additive inapproximability result for Max-MLCR* using the following chain of metric reductions based on two evaluation problems:

- Min-VC, which asks for the minimum value $k$ such that $G$ has a vertex cover of size $k$.

- Min-MLCA, which asks for the minimum value $k$ such that there is a $k$-size subset $N'$ of the $|N|$ neurons in $M$ such that $M(I) \neq (M/N')(I)$.

**Lemma 11.** Min-VC metric reduces to Min-MLCA.

*Proof.* Consider the following reduction from Min-VC to Min-MLCA. Given an instance $X = \langle G = (V, E) \rangle$ of Min-VS, construct the following instance $X' = \langle M, I \rangle$ of Min-MLCA: Let $M$ be an MLP based on $\#n_{tot} = 3|V| + 2|E| + 2$ neurons spread across six layers:

1. **Input neuron layer**: The single input neuron $n_{in}$ (bias +1).

2. **Hidden vertex pair layer**: The vertex neurons $nvP1_1, nvP1_2, \ldots nvP1_{|V|}$ and $nvP2_1, nvP2_2, \ldots nvP2_{|V|}$ (all with bias 0).

3. **Hidden vertex AND layer**: The vertex neurons $nvA_1, nvA_2, \ldots nA_{|V|}$, all of which are 2-way AND ReLU gates.

4. **Hidden edge AND layer**: The edge neurons $neA_1, neA_2, \ldots neA_{|E|}$, all of which are 2-way AND ReLU gates.

5. **Hidden edge NOT layer**: The edge neurons $neN_1, neN_2, \ldots neN_{|E|}$, all of which are NOT ReLU gates.

6. **Output layer**: The single output neuron $n_{out}$, which is an $|E|$-way AND ReLU gate.

The non-zero weight connections between adjacent layers are as follows:

- Each input neuron has an edge of weight 0 coming from its corresponding input and is in turn connected to each of the vertex pair neurons with weight 1.

- Each vertex pair neuron $nvP1_i$ ($nvP2_i$), $1 \leq i \leq |V|$, is connected to vertex AND neuron $nvR_i$ with weight 2 (0).

- Each vertex AND neuron $nvA_i$, $1 \leq i \leq |V|$, is connected to each edge AND neuron whose corresponding edge has an endpoint $v_i$ with weight 1.

- Each edge AND neuron $neA_i$, $1 \leq i \leq |E|$, is connected to edge NOT neuron $neN_i$ with weight 1.

- Each edge NOT neuron $neN_i$, $1 \leq i \leq |E|$, is connected to the output neuron $n_{out}$ with weight 1.

All other connections between neurons in adjacent layers have weight 0. Finally, let $I = (1)$. Observe that this instance of Min-MLCA can be created in time polynomial in the size of the given

instance of Min-VC. Moreover, the output behaviour of the neurons in $M$ from the presentation of input $I$ until the output is generated is as follows:

| timestep | neurons (outputs) |
|---|---|
| 0 | — |
| 1 | $n_{in}(1)$ |
| 2 | $nvP1_1(1), nvP1_2(1), \ldots nvP1_{|V|}(1), nvP2_1(1), nvP2_2(1), \ldots nvP2_{|V|}(1)$ |
| 3 | $nvA_1(1), nvA_2(1), \ldots nvA_{|V|}(1)$ |
| 4 | $neA_1(1), neA_2(1), \ldots neA_{|V|}(1)$ |
| 5 | $neN_1(0), neN_2(0), \ldots neN_{|V|}(0)$ |
| 6 | $n_{out}(0))$ |

Note that, given the 0 (2) connection-weights of P2 (P1) vertex pair neurons to vertex AND neurons, it is the outputs of P1 vertex pair neurons in timestep 2 that enables vertex AND neurons to output 1 in timestep 3.

We now need to show the correctness of this reduction by proving that the answer for the given instance of Min-VC is $k$ if and only if the answer for the constructed instance of Min-MLCA is $k$. We prove the two directions of this if and only if separately as follows:

$\Rightarrow$ : Let $V' = \{v'_1, v'_2, \ldots, v'_k\} \subseteq V$ be a minimum-size vertex cover in $G$ of size $k$ and $N'$ be the $k$-sized sized subset of the P1 vertex pair neurons corresponding to the vertices in $V'$. Let $M'$ be the version of $M$ in which all neurons in $N'$ are ablated. As each of these vertex pair neurons previously allowed their associated vertex AND neurons to output 1, their ablation now allows these $k$ vertex AND neurons to output 0. As $V'$ is a vertex cover of size $k$, all edge AND neurons in $M'$ receive inputs of 0 on at least one of their endpoints from the vertex AND neurons associated with the P1 vertex pair neurons in $N'$. This in turn ensures that all of the edge NOT neurons and the output neuron produces output 1. Hence, $M(I) = 0 \neq 1 = M'(I)$.

$\Leftarrow$ : Let $N'$ be a minimum-sized subset of $N$ of size $k$ such that for the MLP $M'$ induced by ablating all neurons in $N'$, $M(I) \neq M(I')$. As $M(I) = 0$ and circuit outputs are stepped to be Boolean, $M'(I) = 1$. Given the bias of the output neuron, this can only occur if all $|E|$ edge NOT (AND) neurons in $M'$ have output 1 (0) on input $I$, the latter of which requiring that each edge AND neuron receives 0 from at least one of its endpoint vertex AND neurons. These vertex AND neurons can only output 0 if all of their associated P1 vertex neurons have been ablated. This means that the vertices in $G$ corresponding to the P1 vertex pair neurons in $N'$ must form a vertex cover of size $k$ in $G$.

As this proves that Min-VC($X$) = Min-MLCA($X'$), the reduction above is a metric reduction from Min-VC to Min-MLCA. ∎

**Lemma 12.** Min-MLCA metric reduces to Max-MLCR$^*$.

*Proof.* As MLCA is the complement problem of MLCR$^*$ (Observation 3), we already have a trivial reduction from MLCA to MLCR$^*$ on their common input $X = \langle M, I \rangle$. We can then show that $k$ is the minimum value such that $M$ has a $k$ circuit ablation relative to $I$ if and only if $k - 1$ is the maximum value such that $M$ is $k - 1$-robust relative to $I$:

$\Rightarrow$ If $k$ is is the minimum value such that $M$ has a $k$-sized circuit ablation relative to $I$ then no subset of $M$ of size $k - 1$ can be a circuit ablation of $M$ relative to $I$, and $M$ is $(k - 1)$-robust relative to $I$. Moreover, $M$ cannot be $k$-robust relative to $I$ as that would contradict the existence of a $k$-sized circuit ablation for $M$ relative to $I$. Hence, $k - 1$ is the maximum robustness value for $M$ relative to $I$.

$\Leftarrow$ If $k - 1$ is the maximum value such that $M$ is $(k - 1)$-robust relative to $I$ then there must be a subset $H$ of $M$ of size $k$ that ensures $M$ is not $k - 1$-robust, i.e., $(M/H)(I) \neq M(I)$. Such an $H$ is a $k$-sized circuit ablation of $M$ relative to $I$. Moreover, there cannot be a $(k - 1)$-sized circuit ablation of $M$ relative to $I$ as that would contradict the $(k - 1)$-robustness of $M$ relative to $I$. Hence, $k$ is the minimum size of circuit ablations for $M$ relative to $I$.

As this proves that Min-MLCA$(X)$ = Max-MLCR$^*(X)$ +1, the reduction above is a metric reduction from Min-MLCA to Max-MLCR$^*$. ∎

**Theorem 76.** If Max-MLCR$^*$has a $c$-additive approximation algorithm for some $c \in o(poly)$ then $P = NP$.

*Proof.* The $OptP[O(\log n)]$-hardness of Max-MLCR$^*$ follows from the $OptP[O(\log n)]$-hardness of Min-VC (Gasarch et al., 1995, Theorem Theorem 3.3) and the metric reductions in Lemmas 11 and 12. The result then follows from Lemma 10 ∎

Let us now consider problem MLCR.

**Lemma 13.** MLCR$^*$ many-one polynomial-time reduces to MLCR.

*Proof.* Follows from the trivial reduction in which an instance $\langle M, I, k \rangle$ of MLCR$^*$ is transformed into an instance $\langle M, H = M, I, k \rangle$ of MLCR. ∎

**Theorem 77.** If MLCR is polynomial-time tractable then $P = NP$.

*Proof.* Follows from the $coNP$-hardness of MLCR$^*$ (Theorem 72), the reduction in Lemma 13, and Lemma 6. ∎

**Theorem 78.** If $\langle cd, \#n_{in}, \#n_{out}, W_{\max}, B_{\max}, k \rangle$-MLCR is fixed-parameter tractable then $FPT = W[1]$.

*Proof.* Follows from the $coW[1]$-hardness of $\langle cd, \#n_{in}, \#n_{out}, W_{\max}, B_{\max}, k \rangle$-MLCR$^*$ (Theorem 73), the reduction in Lemma 13, and Lemma 9. ∎

**Theorem 79.** $\langle |H| \rangle$-MLCR is fixed-parameter tractable.

*Proof.* Consider the algorithm that generates every possible subset $H'$ of size at most $k$ of the neurons $N$ in $H$ and for each such subset (assuming $M/H'$ is active) checks if $(M/H')(I) \neq M(I)$. If such a subset is found, return "No"; otherwise, return "Yes". The number of possible subsets $H'$ is at most $k \times |H|^k \leq |H| \times |H|^{|H|}$. As any such $M'$ can be generated from $M$, checked or activity, and run on $I$ in time polynomial in the size of the given instance of MLCR, the above is a fixed-parameter tractable algorithm for MLCR relative to parameter-set $\{|H|\}$. ∎

**Theorem 80.** $\langle cw, cd \rangle$-MLCR is fixed-parameter tractable.

*Proof.* Follows from the algorithm in the proof of Theorem 79 and the observation that $|H| \leq cw \times cd$. ∎

**Theorem 81.** $\langle \#n_{tot} \rangle$-MLCR is fixed-parameter tractable.

*Proof.* Follows from the algorithm in the proof of Theorem 79 and the observation that $|H| \leq \#n_{tot}$. ∎

**Theorem 82.** If Max-MLCR has a $c$-additive approximation algorithm for some $c \in o(poly)$ then $P = NP$.

*Proof.* As Max-MLCR$^*$ is a special case of Max-MLCR, if Max-MLCR has a $c$-additive approximation algorithm for some $c$ then so does Max-MLCR$^*$. The result then follows from Theorem 76 ∎

### K.2 Results for MGCR-special and MGCR

Consider the following variant of MGCA:

SPECIAL CIRCUIT ABLATION (SCA)
*Input*: A multi-layer perceptron $M$ of depth $cd$ with $\#n_{tot}$ neurons and maximum layer width $cw$, connection-value matrices $W_1, W_2, \ldots, W_{cd}$, neuron bias vector $B$, and a positive integer $k$ such that $1 \le k \le \#n_{tot}$.
*Question*: Is there a subset $N'$, $|N'| \le k$, of the $|N|$ neurons in $M$ such that for the MLP $M'$ induced by $N \setminus N'$, $M(I) \ne M'(I)$ for some Boolean input vector $I$ of length $\#n_{in}$?

**Theorem 83.** If SCA is polynomial-time tractable then $P = NP$.

*Proof.* Observe that in the instance of MLCA constructed by the reduction from CLIQUE in the proof of Theorem 40, the input-connection weight 0 and bias 1 of the input neuron force this neuron to output 1 for both of the possible input vectors $(1)$ and $(0)$. This means that the answer to the given instance of CLIQUE is "Yes" if and only if the answer to the constructed instance of MLCA relative to any of its possible input vectors is "Yes". Hence, with slight modifications to the proof of reduction correctness, this reduction also establishes the $NP$-hardness of SCA. ∎

**Theorem 84.** If $\langle cd, \#n_{in}, \#n_{out}, W_{\max}, B_{\max}, k \rangle$-SCA is fixed-parameter tractable then $FPT = W[1]$.

*Proof.* Observe that in the instance of SCA constructed in the reduction in the proof of Theorem 83, $\#n_{in} = \#n_{out} = W_{\max} = 1$, $cd = 5$, and $B_{\max}$ and $k$ are function of $k$ in the given instance of CLIQUE. The result then follows from the fact that $\langle k \rangle$-CLIQUE is $W[1]$-hard (Downey & Fellows, 1999). ∎

**Theorem 85.** $\langle \#n_{tot} \rangle$-SCA is fixed-parameter tractable.

*Proof.* Modify the algorithm in the proof of Theorem 41 such that each created MLP $M$ is checked to ensure that $M(I) \ne M'(I)$ for every possible Boolean input vector of length $\#n_{in}$. As the number of such vectors is $2^{\#n_{in}} \le 2^{\#n_{tot}}$, the above is a fixed-parameter tractable algorithm for SCA relative to parameter-set $\{\#n_{tot}\}$. ∎

**Theorem 86.** $\langle cw, cd \rangle$-SCA is fixed-parameter tractable.

*Proof.* Follows from the algorithm in the proof of Theorem 85 and the observation that $\#n_{tot} \le cw \times cd$. ∎

Our results above for SCA above gain importance for us here courtesy of the following observation.

**Observation 4.** MGCR$^*$ is the complement of SCA.

*Proof.* Let $X = \langle M, k \rangle$ be a shared input of MGCR$^*$ and SCA. If the the answer to SCA on input $X$ is "Yes", this means that there is a subset $H$ of size $\leq k$ of the neurons of $M$ that can be ablated such that $M(I) \neq (M/H)(I)$ for some input vector $I$. This implies that the answer to MGCR$^*$ on input $X$ is "No". Conversely, if the answer to SCA on input $X$ is "No", this means that there is no subset $H$ of size $\leq k$ of the neurons of $M$ that can be ablated such that $M(I) \neq (M/H)(I)$ for any input vector $I$. This implies that the answer to MGCR$^*$ on input $X$ is "Yes'. The observation follows from the definition of complement problem. ∎

**Theorem 87.** If MGCR$^*$ is polynomial-time tractable then $P = NP$.

*Proof.* As MGCR$^*$ is the complement of SCA by Observation 4 and SCA is $NP$-hard (Theorem 83), Lemma 5 implies that MGCR$^*$ is $coNP$-hard. The result then follows from Lemma 6. ∎

**Theorem 88.** If $\langle cd, \#n_{in}, \#n_{out}, W_{\max}, B_{\max}, k \rangle$-MGCR$^*$ is fixed-parameter tractable then $FPT = W[1]$.

*Proof.* The $coW[1]$-hardness of $\langle cd, \#n_{in}, \#n_{out}, W_{\max}, B_{\max}, k \rangle$-MGCR$^*$ follows from the $W[1]$-hardness of $\langle cd, \#n_{in}, \#n_{out}, W_{\max}, B_{\max}, k \rangle$-SCA (Theorem 84), Observation 4, and Lemma 7. The result then follows from Lemma 9. ∎

**Theorem 89.** $\langle \#n_{tot} \rangle$-MGCR$^*$ is fixed-parameter tractable.

*Proof.* The result follows from the fixed-parameter tractability of $\langle \#n_{tot} \rangle$-SCA (Theorem 85), Observation 4, and Lemma 8. ∎

**Theorem 90.** $\langle cw, cd \rangle$-MGCR$^*$ is fixed-parameter tractable.

*Proof.* The result follows from the fixed-parameter tractability of $\langle cw, cd \rangle$-SCA (Theorem 86), Observation 4, and Lemma 8. ∎

Observe that the results in Theorems 88–90 in combination with Lemmas 1 and 2 suffice to establish the parameterized complexity status of MGCR$^*$ relative to many subsets of the parameters listed in Table 10.

We can derive a polynomial-time additive inapproximability result for Max-MGCR$^*$ using the following chain of metric reductions based on two evaluation problems:

- Min-VC, which asks for the minimum value $k$ such that $G$ has a vertex cover of size $k$.

- Min-SCA, which asks for the minimum value $k$ such that there is a $k$-size subset $N'$ of the $|N|$ neurons in $M$ such that $M(I) \neq (M/N')(I)$ for some Boolean input vector $I$ of length$\#n_{in}$.

**Lemma 14.** Min-VC metric reduces to Min-SCA.

*Proof.* Observe that in the instance of Min-MLCA constructed by the reduction from Min-VC in the proof of Theorem 11, the input-connection weight 0 and bias 1 of the input neuron force this neuron to output 1 for both of the possible input vectors (1) and (0). This means that the answer to the constructed instance of Min-MLCA is the same relative to any of its possible input vectors. Hence, with slight modifications to the proof of reduction correctness, this reduction is also a metric reduction from Min-VC to Min-SCA such that for given instance $X$ of Min-VC and constructed instance $X'$ of Min-SCA, Min-VC$(X) = $ Min-SCA$(X')$. ∎

**Lemma 15.** Min-SCA metric reduces to Max-MGCR$^*$.

*Proof.* As SCA is the complement problem of MGCR$^*$ (Observation 4), we already have a trivial reduction from SCA to MGCR$^*$ on their common input $X = \langle M \rangle$. We can then show that $k$ is the minimum value such that $M$ has a $k$ circuit ablation relative to some possible $I$ if and only if $k-1$ is the maximum value such that $M$ is $k-1$-robust relative to all possible $I$:

$\Rightarrow$ If $k$ is is the minimum value such that $M$ has a $k$-sized circuit ablation relative to some possible $I$ then no subset of $M$ of size $k-1$ can be a circuit ablation of $M$ relative to any possible $I$, and $M$ is $(k-1)$-robust relative to all possible $I$. Moreover, $M$ cannot be $k$-robust relative to all possible $I$ as that would contradict the existence of a $k$-sized circuit ablation for $M$ relative to some possible $I$. Hence, $k-1$ is the maximum robustness value for $M$ relative to all possible $I$.

$\Leftarrow$ If $k-1$ is the maximum value such that $M$ is $(k-1)$-robust relative to all possible $I$ then there must be a subset $H$ of $M$ of size $k$ that ensures $M$ is not $k-1$-robust relative to all possible $I$, i.e., $(M/H)(I) \neq M(I)$ for some possible $I$. Such an $H$ is a $k$-sized circuit ablation of $M$ relative to that $I$. Moreover, there cannot be a $(k-1)$-sized circuit ablation of $M$ relative to some possible $I$ as that would contradict the $(k-1)$-robustness of $M$ relative to all possible $I$. Hence, $k$ is the minimum size of circuit ablations for $M$ relative to some possible $I$.

As this proves that Min-SCA$(X) = $ Max-MGCR$^*(X) + 1$, the reduction above is a metric reduction from Min-SCA to Max-MGCR$^*$. ∎

**Theorem 91.** If Max-MGCR$^*$ has a $c$-additive approximation algorithm for some $c \in o(poly)$ then $P = NP$.

*Proof.* The $OptP[O(\log n)]$-hardness of Max-MGCR$^*$ follows from the $OptP[O(\log n)]$-hardness of Min-VC (Gasarch et al., 1995, Theorem Theorem 3.3) and the metric reductions in Lemmas 14 and 15. The result then follows from Lemma 10 ∎

Let us now consider problem MGCR.

**Lemma 16.** MGCR$^*$ many-one polynomial-time reduces to MGCR.

*Proof.* Follows from the trivial reduction in which an instance $\langle M, k \rangle$ of MGCR$^*$ is transformed into an instance $\langle M, H = M, k \rangle$ of MGCR. ∎

**Theorem 92.** If MGCR is polynomial-time tractable then $P = NP$.

*Proof.* Follows from the $coNP$-hardness of MGCR$^*$ (Theorem 87), the reduction in Lemma 16, and Lemma 6. ∎

**Theorem 93.** If $\langle cd, \#n_{in}, \#n_{out}, W_{\max}, B_{\max}, k \rangle$-MGCR is fixed-parameter tractable then $FPT = W[1]$.

*Proof.* Follows from the $coW[1]$-hardness of $\langle cd, \#n_{in}, \#n_{out}, W_{\max}, B_{\max}, k \rangle$-MGCR$^*$ (Theorem 88), the reduction in Lemma 16, and Lemma 9. ∎

**Theorem 94.** $\langle \#n_{in}, |H| \rangle$-MGCR is fixed-parameter tractable.

*Proof.* Modify the algorithm in the proof of Theorem 79 such that each created MLP $M$ is checked to ensure that $M(I) \neq (H/H')(I)$ for every possible Boolean input vector of length $\#n_{in}$. As the number of such vectors is $2^{\#n_{in}}$, the above is a fixed-parameter tractable algorithm for MGCR relative to parameter-set $\{\#n_{in}, |H|\}$. ∎

**Theorem 95.** $\langle cw, cd \rangle$-MGCR is fixed-parameter tractable.

*Proof.* Follows from the algorithm in the proof of Theorem 94 and the observations that $\#n_{tot} \leq cw \times cd$ and $|H| \leq cw \times cd$. ∎

**Theorem 96.** $\langle \#n_{tot} \rangle$-MGCR is fixed-parameter tractable.

*Proof.* Follows from the algorithm in the proof of Theorem 94 and the observations that $\#n_{tot} \leq \#n_{tot}$ and $|H| \leq \#n_{tot}$. ∎

**Theorem 97.** If Max-MGCR has a $c$-additive approximation algorithm for some $c \in o(poly)$ then $P = NP$.

*Proof.* As Max-MGCR* is a special case of Max-MGCR, if Max-MGCR has a $c$-additive approximation algorithm for some $c$ then so does Max-MGCR*. The result then follows from Theorem 91 ∎

## L   SUFFICIENT REASONS PROBLEM

MINIMUM SUFFICIENT REASON (MSR)
*Input*: A multi-layer perceptron $M$ of depth $cd$ with $\#n_{tot}$ neurons and maximum layer width $cw$, connection-value matrices $W_1, W_2, \ldots, W_{cd}$, neuron bias vector $B$, a Boolean input vector $I$ of length $\#n_{in}$, and a positive integer $k$ such that $1 \leq k \leq \#n_{in}$.
*Question*: Is there a $k$-sized subset $I'$ of $I$ such that for each possible completion $I''$ of $I'$, $I$ and $I''$ are behaviorally equivalent with respect to $M$?

For a graph $G = (V, E)$, we shall assume an ordering on the vertices and edges in $V$ and $E$, respectively. For each vertex $v \in V$, let the complete neighbourhood $N_C(v)$ of $v$ be the set composed of $v$ and the set of all vertices in $G$ that are adjacent to $v$ by a single edge, i.e., $v \cup \{u \mid u \in V \text{ and } (u, v) \in E\}$.

We will prove various classical and parameterized results for MSR using reductions from CLIQUE. The parameterized results are proved relative to the parameters in Table 11. An additional reduction from DS (Theorem 103) use specialized ReLU logic gates described in Barceló et al. 2020, Lemma 13. These gates assume Boolean neuron input and output values of 0 and 1 and are structured as follows:

1. NOT ReLU gate: A ReLU gate with one input connection weight of value $-1$ and a bias of 1. This gate has output 1 if the input is 0 and 0 otherwise.

2. $n$-way AND ReLU gate: A ReLU gate with $n$ input connection weights of value 1 and a bias of $-(n-1)$. This gate has output 1 if all inputs have value 1 and 0 otherwise.

3. $n$-way OR ReLU gate: A combination of an $n$-way AND ReLU gate with NOT ReLU gates on all of its inputs and a NOT ReLU gate on its output that uses DeMorgan's Second Law to implement $(x_1 \vee x_2 \vee \ldots x_n)$ as $\neg(\neg x_1 \wedge \neg x_2 \wedge \ldots \neg x_n)$. This gate has output 1 if any input has value 1 and 0 otherwise.

### L.1   RESULTS FOR MSR

The following hardness results are notable for holding when the given MLP $M$ has only one hidden layer. As such, they complement and significantly tighten results given in (Barceló et al., 2020; Wäldchen et al., 2021), respectively.

**Theorem 98.** If MSR is polynomial-time tractable then $P = NP$.

*Proof.* Consider the following reduction from CLIQUE to MSR. Given an instance $\langle G = (V, E), k \rangle$ of CLIQUE, construct the following instance $\langle M, I, k' \rangle$ of MSR: Let $M$ be an MLP based on $\#n_{tot} = |V| + |E| + 1$ neurons spread across three layers:

1. **Input vertex layer**: The vertex neurons $nv_1, nv_2, \ldots nv_{|V|}$ (all with bias 0).

2. **Hidden edge layer**: The edge neurons $ne_1, ne_2, \ldots ne_{|E|}$ (all with bias $-1$).

3. **Output layer**: The single output neuron $n_{out}$ (bias $-(k(k-1)/2 - 1)$).

Note that this MLP has only one hidden layer. The non-zero weight connections between adjacent layers are as follows:

- Each vertex neuron $nv_i$, $1 \leq i \leq |V|$, is connected to each edge neuron whose corresponding edge has an endpoint $v_i$ with weight 1.

- Each edge neuron $ne_i$, $1 \leq i \leq |E|$, is connected to the output neuron $n_{out}$ with weight 1.

All other connections between neurons in adjacent layers have weight 0. Finally, let $I = (1)$ and $k' = k$. Observe that this instance of MSR can be created in time polynomial in the size of the given instance of CLIQUE. Moreover, the output behaviour of the neurons in $M$ from the presentation of input $I$ until the output is generated is as follows:

| timestep | neurons (outputs) |
|----------|-------------------|
| 0 | — |
| 1 | $nv_1(1), nv_2(1), \ldots nv_{|V|}(1)$ |
| 2 | $ne_1(1), ne_2(1), \ldots ne_{|E|}(1)$ |
| 3 | $n_{out}(|E| - (k(k-1)/2 - 1))$ |

Note that it is the stepped output of $n_{out}$ in timestep 3 that yields output 1.

We now need to show the correctness of this reduction by proving that the answer for the given instance of CLIQUE is "Yes" if and only if the answer for the constructed instance of MSR is "Yes". We prove the two directions of this if and only if separately as follows:

$\Rightarrow$ : Let $V' = \{v'_1, v'_2, \ldots, v'_k\} \subseteq V$ be a clique in $G$ of size $k$ and $I'$ be the $k' = k$-sized subset of $I$ corresponding to the vertices in $V'$. As $V'$ is a clique of size $k$, exactly $k(k-1)/2$ edge neurons in the constructed MLP $M$ receive the requisite inputs of 1 on both of their endpoints from the vertex neurons associated with $I'$. This in turn ensures the output neuron produces output 1. No other possible inputs to the vertex neurons not corresponding to elements of $I'$ can change the outputs of these activated edge neurons (and hence the output neuron as well) from 1 to 0. Hence, all completions of $I'$ cause $M$ to output 1 and are behaviorally equivalent to $I$ with respect to $M$.

Table 11: Parameters for the minimum sufficient reason problem.

| Parameter | Description |
|-----------|-------------|
| $cd$ | # layers in given MLP |
| $cw$ | max # neurons in layer in given MLP |
| $\#n_{tot}$ | total # neurons in given MLP |
| $\#n_{in}$ | # input neurons in given MLP |
| $\#n_{out}$ | # output neurons in given MLP |
| $B_{\max}$ | max neuron bias in given MLP |
| $W_{\max}$ | max connection weight in given MLP |
| $k$ | Size of requested subset of input vector |

$\Leftarrow$ : Let $I'$ be a $k' = k$-sized subset of $I$ such that all possible completions of $I'$ are behaviorally equivalent to $I$ with respect to $M$, i.e., all such completions cause $M$ to output 1. Consider the completion $I''$ of $I'$ in which all non-$I'$ elements have value 0. The output of $M$ on $I''$ can be 1 (and hence equal to the output of $M$ on $I$) only if at least $k(k-1)/2$ edge neurons have output 1. As all non-$I'$ elements of $I''$ have value 0, this means that both endpoints of each of these edge neuron must be connected to elements of $I'$ with output 1, which in turn implies that the $k' = k$ vertices in $G$ corresponding to the elements of $I'$ form a clique of size $k$ for $G$.

As CLIQUE is $NP$-hard (Garey & Johnson, 1979), the reduction above establishes that MSR is also $NP$-hard. The result follows from the definition of $NP$-hardness. ∎

**Theorem 99.** If $\langle cd, \#n_{out}, W_{\max}, B_{\max}, k \rangle$-MSR is fixed-parameter tractable then $FPT = W[1]$.

*Proof.* Observe that in the instance of MSR constructed in the reduction in the proof of Theorem 98, $\#n_{out} = W_{\max} = 1$, $cd = 3$, and $B_{\max}$ and $k$ are function of $k$ in the given instance of CLIQUE. The result then follows from the fact that $\langle k \rangle$-CLIQUE is $W[1]$-hard (Downey & Fellows, 1999). ∎

**Theorem 100.** $\langle \#n_{in} \rangle$-MSR is fixed-parameter tractable.

*Proof.* Consider the algorithm that generates each possible subset $I'$ of of $I$ of size $k$ and for each such $I'$, checks if all possible completions of $I'$ are behaviorally equivalent to $I$ with respect to $M$. If such an $I'$ is found, return "Yes"; otherwise, return "No". The number of possible $I'$ is at most $(\#n_{in})^k \leq (\#n_{in})^{\#n_{in}}$ and the number of possible completions of any such $I'$ is less than $2^{\#n_{in}}$. Given this, as $M$ can be run on each completion of $I'$ is time polynomial in the size of the given instance of MSR, the above is a fixed-parameter tractable algorithm for MSR relative to parameter-set $\{\#n_{in}\}$. ∎

**Theorem 101.** $\langle \#n_{tot} \rangle$-MSR is fixed-parameter tractable.

*Proof.* Follows from the algorithm in the proof of Theorem 100 and the observation that $\#n_{in} \leq \#n_{tot}$. ∎

**Theorem 102.** $\langle cw \rangle$-MSR is fixed-parameter tractable.

*Proof.* Follows from the algorithm in the proof of Theorem 100 and the observation that $\#n_{in} \leq cw$. ∎

Observe that the results in Theorems 99–102 in combination with Lemmas 1 and 2 suffice to establish the parameterized complexity status of MSR relative to every subset of the parameters listed in Table 11.

Let us now consider the polynomial-time cost approximability of MSR. As MSR is a minimization problem, we cannot do this using reductions from a maximization problem like CLIQUE. Hence we will instead use a reduction from another minimization problem, namely DS.

**Theorem 103.** If MSR is polynomial-time tractable then $P = NP$.

*Proof.* Consider the following reduction from DS to MSR. Given an instance $\langle G = (V, E), k \rangle$ of DS, construct the following instance $\langle M, I, k' \rangle$ of MSR: Let $M$ be an MLP based on $\#n_{tot,g} = 3|V| + 1$ neurons spread across four layers:

1. **Input layer**: The input vertex neurons $nv_1, nv_2, \ldots nv_{|V|}$, all of which have bias 0.

2. **Hidden vertex neighbourhood layer I**: The vertex neighbourhood AND neurons $nvnA_1, nvnA_2, \ldots nvnA_{|V|}$, where $nvnA_i$ is an $x$-way AND ReLU gates such that $x = |N_C(v_i)|$.

3. **Hidden vertex neighbourhood layer II**: The vertex neighbourhood NOT neurons $nvnN_1, nvnN_2, \ldots nvnN_{|V|}$, all of which are NOT ReLU gates.

4. **Output layer**: The single output neuron $n_{out}$, which is a $|V|$-way AND ReLU gate.

The non-zero weight connections between adjacent layers are as follows:

- Each input vertex neuron $nv_i$, $1 \leq i \leq |V|$, is connected to each vertex neighbourhood AND neuron $nvnA_j$ such that $v_i \in N_C(v_j)$ with weight 1.

- Each vertex neighbourhood AND neuron $nvnA_i$, $1 \leq i \leq |V|$, is connected to its corresponding vertex neighbourhood NOT neuron $nvnN_i$ with weight 1.

- Each vertex neighbourhood NOT neuron $nvnN_i$, $1 \leq i \leq |V|$, is connected to the output neuron $n_{out}$ with weight 1.

All other connections between neurons in adjacent layers have weight 0. Finally, let $I$ be the $|V|$-length zero-vector and $k' = k$. Observe that this instance of MSR can be created in time polynomial in the size of the given instance of DS, Moreover, the output behaviour of the neurons in $M$ from the presentation of input $I$ until the output is generated is as follows:

| timestep | neurons (outputs) |
|---|---|
| 0 | — |
| 1 | $nvN_1(0), nvN_2(0), \ldots nvN_{|V|}(0)$ |
| 2 | $nvnA_1(0), nvnA_2(0), \ldots nvnA_{|V|}(0)$ |
| 3 | $nvnN_1(1), nvnN_2(1), \ldots nvnN_{|V|}(1)$ |
| 4 | $n_{out}(1)$ |

We now need to show the correctness of this reduction by proving that the answer for the given instance of DS is "Yes" if and only if the answer for the constructed instance of MSR is "Yes". We prove the two directions of this if and only if separately as follows:

$\Rightarrow$ : Let $V' = \{v'_1, v'_2, \ldots, v'_k\} \subseteq V$ be a dominating set in $G$ of size $k$ and $I'$ be the $k' = k$-sized subset of $I$ corresponding to the vertices in $V'$. As $V'$ is a dominating set, each vertex neighbourhood AND neuron receives input 0 from at least one input vertex neuron in the set of input vertex neurons associated with $I'$, which in turn ensures that each vertex neighbourhood AND neuron has output 0. This in turn ensures that $M$ produces output 1. No other possible inputs to the vertex neighbourhood AND neurons can change the output of these neurons from 0 to 1. Hence, all completions of $I'$ cause $M$ to output 1 and are behaviorally equivalent to $I$ with respect to $M$.

$\Leftarrow$ : Let $I'$ be a $k' = k$-sized subset of $I$ such that all possible completions of $I'$ are behaviorally equivalent to $I$ with respect to $M$, i.e., all such completions cause $M$ to output 1. Consider the completion $I''$ of $I'$ in which all non-$I'$ elements have value 1. The output of $M$ on $I''$ can be 1 (and hence equal to the output of $M$ on $I$) only if all vertex neighbourhood NOT neurons output 1, which in turn can happen only if all vertex neighbourhood AND gates output 0. However, as all non-$I'$ elements of $I''$ have value 1, this means that each vertex neighbourhood AND neuron must be connected to at least one element of $I'$, which in turn implies that the $k' = k$ vertices in $G$ corresponding to the elements of $I'$ form a dominating set of size $k$ for $G$.

As DS is $NP$-hard (Garey & Johnson, 1979), the reduction above establishes that MSR is also $NP$-hard. The result follows from the definition of $NP$-hardness. ∎

**Theorem 104.** If MSR has a polynomial-time $c$-approximation algorithm for any constant $c > 0$ then $FPT = W[1]$.

*Proof.* Recall from the proof of correctness of the reduction in the proof of Theorem 103 that a given instance of DS has a dominating set of size $k$ if and only if the constructed instance of $MSR$ has a subset $I'$ of $I$ of size $k' = k$ such that every possible completion of $I'$ is behaviorally equivalent to $I$ with respect to $M$. This implies that, given a polynomial-time $c$-approximation algorithm $A$ for MSR for some constant $c > 0$, we can create a polynomial-time $c$-approximation algorithm for DS by applying the reduction to the given instance $x$ of DS to construct an instance $x'$ of MSR, applying $A$ to $x'$ to create an approximate solution $y'$, and then using $y'$ to create an approximate solution $y$ for $x$ that has the same cost as $y'$. The result then follows from Chen & Lin 2019, Corollary 2, which implies that if DS has a polynomial-time $c$-approximation algorithm for any constant $c > 0$ then $FPT = W[1]$. ∎

Note that this theorem also renders MSR PTAS-inapproximable unless $FPT = W[1]$.

## M  PROBABILISTIC APPROXIMATION SCHEMES

Let us now consider three other types of polynomial-time approximability that may be acceptable in situations where always getting the correct output for an input is not required:

1. algorithms that always run in polynomial time but are frequently correct in that they produce the correct output for a given input in all but a small number of cases (i.e., the number of errors for input size $n$ is bounded by function $err(n)$) Hemaspaandra & Williams (2012);

2. algorithms that always run in polynomial time but are frequently correct in that they produce the correct output for a given input with high probability Motwani & Raghavan (1995); and

3. algorithms that run in polynomial time with high probability but are always correct (Gill, 1977).

Unfortunately, none of these options are in general open to us, courtesy of the following result.

**Theorem 105.** None of the hard problems in Table 4 are polynomial-time approximable in senses (1–3).

*Proof.* (Sketch) Holds relative to several strongly-believed or established complexity-class relation conjectures courtesy of the $NP$-hardness of the problems and the reasoning in the proof of (Wareham, 2022, Result E). ∎

## N  SUPPLEMENTARY DISCUSSION

**Search space size versus intrinsic complexity.** Some of our hardness results can be surprising (e.g., fixed-parameter intractability indicating that taming intuitive network and circuit parameters is not enough to make queries feasible). Other findings might be unsurprising/surprising for the wrong reasons. Often intractability is assumed based on observing that a problem of interest has an exponential search space. But this is not a sufficient condition for intractability. For instance, although the Minimum Spanning Tree problem has an exponential search space, there is enough structure in it that can be exploited to get optimal solutions tractably. A more directly relevant example is our Quasi-Minimal Circuit problems, which also have exponential search spaces. This is a scenario where the typical reasoning in the literature would lead us astray. Given our tractability results, jumping to intractability conclusions would miss valuable opportunities to design tractable algorithms with guarantees.

**Worst-case analysis.** Given our limited knowledge of the problem space of interpretability, worst-case analysis is appropriate to explore what problems might be solvable without requiring any additional assumptions (e.g., Bassan et al., 2024; Barceló et al., 2020) and experimental results suggest it captures a lower bound on real-world complexity (e.g., Friedman et al., 2024; Shi et al., 2024; Yu et al., 2024a). One possibly fruitful avenue would be to conduct an empirical and formal characterization of learned weights in search of structure that could potentially distinguish conditions of

(in)tractability. This could inform future average-case analyses on plausible distributional assumptions.

**Strategies for exploring the viability of interpretability queries.** Although we find that many queries of interest are intractable in the general case (and empirical results are in line with this characterization), this should not paralyze real-world efforts to interpret models. As our exploration of the current complexity landscape shows, reasonable relaxations, restrictions and problem variants can yield tractable queries for circuits with useful properties. Consider a few out of many possible avenues to continue these explorations.

**(i)** Faced with an intractable query, we can investigate which parameters of the problem (e.g., network, circuit aspects) might be responsible for the core hardness of the general problem. If these problematic parameters can be kept small in real-world applications, this can yield a fixed-parameter tractable query which can be answered efficiently in practice. We have explored some of these parameters, but many more could be, as any aspect of the problem can be parameterized. For this, a close dialogue between theorists and experimentalists will be crucial, as often empirical regularities suggest which parameters might be fruitful to explore theoretically, and experiments can test whether theoretically conjectured parameters are or can be kept small in practice.

**(ii)** Generating altogether different circuit query variants is another way of making interpretability feasible. Our formalization of quasi-minimal circuit problems illustrates the search for viable algorithmic options with examples of tractable problems for inner interpretability. When the use case is well defined, efficient queries that return circuits with useful affordances for applications can be designed. Some circuits (e.g., quasi-minimal circuits, but likely others) might mimic the affordances for prediction/control that ideal circuits have, while shedding the intractability that plagues the latter.

**(iii)** It could be fruitful to investigate properties of the network output. Although for some problems, our constructions use step functions in the output layer (following the literature; Bassan et al., 2024; Barceló et al., 2020), for many problems we do not or provide alternative proofs without them. This suggests this is not likely a significant source of complexity. Another aspect could be the binary input/output, although continuous input/output does not necessarily matter complexity-wise. Sometimes it does, as in the case of Linear Programming (PTIME; Karmarkar, 1984) versus 0-1 Integer Programming (NP-complete Garey & Johnson, 1979), and sometimes it does not, as in Euclidean Steiner Tree (NP-hard, not in NP for technical reasons; Garey & Johnson, 1979) versus Rectilinear Steiner Tree (NP-complete; Garey & Johnson, 1979). Still, this is an interesting direction for future work, as it suggests studying the output as an axis of approximation.

**(iv)** A different path is to design queries that partially rely on mid-level abstractions (Vilas et al., 2024a) to bridge the gap between circuits and human-intelligible algorithms (e.g., key-value mechanisms; Geva et al., 2022; Vilas et al., 2024b).

**(v)** It is in principle possible that real-world trained neural networks possess an internal structure that is somehow benevolent to general (ideal) circuit queries (e.g., redundancy). In such optimistic scenarios, general-purpose heuristics might work well. The empirical evidence available, however, speaks against this possibility. In any case, it will always be important to characterize any 'benevolent structure' in the problems such that we can leverage it explicitly to design algorithms with useful guarantees.