# OpenReview forum: "The Computational Complexity of Circuit Discovery for Inner Interpretability"
_ICLR.cc/2025/Conference — ICLR 2025 Spotlight_

### Official Review · Reviewer_YPDD · 2024-10-28

**Soundness:** 4
**Presentation:** 3
**Contribution:** 2
**Rating:** 6
**Confidence:** 4

**Summary:**

The paper formalizes several problems along the lines of finding the minimal subcircuit in a deep network that matches its computation; this is motivated by finding a small interpretable subcircuit. They show that this and several variants of this problem are intractable (either NP-hard or \Sigma_2^2. For several variants they all show that there is no PTAS.

**Strengths:**

Interpreting formalization and complexity study of several problems motivated by interpretability of deep networks.

**Weaknesses:**

The hardness results are not surprising. The area of finding small circuits is in general very hard (see for example: https://www.cs.uwyo.edu/~jhitchco/papers/npcmcsp.pdf even though it is a related but very different problem). The reduction from CLIQUE is simple.

The worst case hardness may not capture the average case complexity for real world distributions. I do understand proving hardness results for probabilistic settings is more difficult. But the impossibility results may not imply hardness of interpretability in practice.

**Questions:**

The paper is well written and gives nice clean formalization of several problems motivated by interpretability.

In several of the problem formalizations you are trying to find the minimal sub-circuit for a given input x. If the problem is a boolean classification problem then in many circuits there might be some trivial tiny sub-circuits that give the correct output for that given x -- in your reduction you have avoided that by choosing a carefully crafted worst case circuit. Wouldn’t it make more sense to look at a distribution of inputs? I would have expected that one would get a much stronger inapproximability hardness result if you took a set of inputs and insisted that the sub-circuit should output correctly for all inputs which you are doing problem 8.

What is the takeaway here for interpretability in practice? Do you think interpretability is essentially intractable? Please expand on potential practical approaches to interpretability in light of these hardness results with scenarios and reasons as to why interpretability might still be feasible despite the worst-case intractability.

---

> ### Author Response · Authors · 2024-11-18
> **Response to Reviewer YPDD (1)**
>
> Thanks for the summary. We would add that we also study problems other than sufficient circuits (gnostic neurons, circuit robustness, necessary circuits), and explore relaxations such as quasi-minimal sufficient circuits / circuit patching, for which we prove tractability and give algorithms that can be combined with existing (e.g., gradient-based) techniques.
>
> Below we reply to each comment and detail our changes to the manuscript to address each issue.
>
> * __Response to Weakness 1.1 (surprisingness of hardness results):__
>     * Thanks for the comment. We think various clarifications are worth including in the manuscript.
>     * (i) As we write in Related Work and highlighted by Reviewer ZH6m, ours is the first report of this kind.
>         * Among the results we report (which include both tractability and intractability), the hardness results are the first to be proven for inner interpretability queries (the thematically closest existing results being hardness of input queries; Bassan et al., 2024, ICML; Barcelo et al., 2020, NeurIPS).
>     * The paper cited by the reviewer (Hitchcock et al.) is on the complexity status of the problem of finding a bounded size circuit given a truth-table (Minimum Circuit Size Problem; MCSP).
>         * Our problems, as the reviewer correctly points out, are very different.
>             * Circuit query problems have trained neural networks as inputs and the desired output is a circuit with certain properties, typically related to its size, minimality, and faithfulness to the behavior of the network (but see also multiple other problems we formalize with different characteristics).
>         * Our proofs therefore apply to the particular settings where the circuit is a subset of the internals of a trained neural network (and the problem does not blow up the input exponentially as with truth tables), which might intuitively be thought to be an easier problem, since we are not trying to find the smallest possible circuit for a truth table but rather the smallest available in the trained network (though intuitions are usually not a good guide).
>         * Note as well that the literature on MCSP conjectures and gives evidence that the problem is NP-intermediate, in contrast to our results which prove NP-completeness or harder of our circuit queries. This is another sense in which our results are not directly paralleled by other work.
>         * All of this means that the hardness reported in that literature does not transfer directly to our problems, and (in)tractability of our circuit queries cannot be assumed without (our) proofs.
>         * This work will added and disscussed in the Related Work section.
>     * To sum up, as the reviewer states, “finding small circuits is _in general_ very hard” (emphasis ours). The crucial point here is that existing results don’t transfer directly (speacial cases might be easy), whereas our proofs apply to particular cases of practical interest to the interpretability community (i.e., trained MLPs; by extension, transformers), not merely in the most general case (which arguably is of little use to interpretability researchers working with neural networks).

---

> ### Author Response · Authors · 2024-11-18
> **Response to Reviewer YPDD (2)**
>
> __Response to Weakness 1.1 (cont.)__
>
> (ii) Moreover, we would argue
> * (a) Sometimes results can appear non-surprising for the wrong reasons.
>     * It is important to signal not only whether our results contradict/confirm folklore intuitions but also to signal whether these intuitions were arrived at for the wrong reasons, even when they might be nominally correct.
>     * Much of the literature that develops heuristics often assumes intractability without evidence, for example, by alluding to exponential search spaces. But exponential search spaces are not a sufficient condition for intractability.
>         * For instance, the Minimum Spanning Tree problem has an exponential search space.
>             * However, there is enough structure in this space such that it can be exploited to get a minimum spanning tree in quadratic time.
>             * Another example is obtaining an optimal segmentation of a sequence given the value of the possible segments.
>             * A more directly relevant example is our Quasi-Minimal Circuit problems, which is a scenario where the typical reasoning in the literature would clearly go wrong. Faced with the need to design algorithms to solve a Quasi-Minimal Circuit query, interpretability researchers could perhaps observe that the search space is exponential. As is typically done and justified in the literature, this problem would be categorized as “intractable” and this would prompt the development of heuristics with no proven guarantees (we need not single out individual work here so we omit the citations). Given our results (i.e., tractability of Quasi-minimal sufficient circuit and circuit patching, and algorithms to compute them that can be combined with gradient techniques, logit lens techniques, etc.), one can see that jumping to intractability conclusions as above would miss the opportunity to use existing tractable techniques and to leverage the provable guarantees that they provide.
>         * Our work hopefully contributes to move the field forward by encouraging researchers to consider restrictions to the problems and additional assumptions that might yield tractable procedures with known properties, and to gain clarity on what problems current heuristics are well equipped to solve.
>         * Our results might contradict folklore assumptions also in this latter sense. Often intractability is assumed (as above, for the wrong reasons) and heuristics are developed that are again expected intuitively to be good approximations. Some of our results show these heuristics are not likely to be good approximations to the general problems, but to more restricted problems that are nevertheless worth exploring (e.g., our quasi-minimal formulations).
> * (b) Some of our results can be surprising. The following are just some examples.
>     * Our parameterized complexity results indicate that keeping some features like depth small does not help alleviate the complexity of the problems.
>     * Our Quasi-minimal variants show how circuit queries with exponential search spaces cannot be assumed to be hard (as much of the literature currently does), but can have tractable algorithms with guarantees.
>     * Our results also help explain mixed experimental results that are hard to make sense of (as with our separation between global and local queries, and quasi-minimal circuits which can mimic some sufficient circuit properties while being easier to find).

---

> > ### Author Response · Authors · 2024-11-18
> > **Response to Reviewer YPDD (3)**
> >
> > * __Response to Weakness 1.2 (simplicity of reduction from Clique):__
> >     * Thank you for this observation. We would like to make the following clarifications.
> >         * We see this simplicity as a virtue of our proofs that in many cases is hard to obtain.
> >             * Although our proofs could easily be made more complicated (indeed, sometimes they are at first pass); making them readily understandable by practitioners and efficient (in the sense that a single construction can serve to obtain multiple results) is part of the task that we undertook here.
> >             * That said, more intricate constructions can be found in the proofs for other theorems
> >         * Simplicity, in other words, is not an indication of the result’s value.
> >             * This comes from their importance in the context they are deployed to solve problems of interest in the field of interpretability, as pointed out by Reviewer 19hk, Reviewer ZH6m, and Reviewer 8wLc in their summaries of our work and descriptions of strengths.
> >         * Technical significance can come, among other things, from the introduction of new mathematics and also from the deployment of existing math to answer important questions.
> >             * As is the case for other work in this area (which might be called “Applied Complexity Theory”; Bassan et al., 2024, ICML; Barcelo et al., 2020, NeurIPS), we are not aiming at developing groundbreaking mathematics but rather deploying mathematical tools to answer important questions that connect to the practice of interpretability.
> >         * This is an important clarification about the approach that this “Applied Complexity Theory” field  takes, so we add it to the preamble of our Appendix before presenting theorems and proofs.

---

> > > ### Author Response · Authors · 2024-11-18
> > > **Response to Reviewer YPDD (4)**
> > >
> > > * __Response to Weakness 2 (worst-case hardness and real-world complexity):__
> > >     * Note existing empirical results support the idea that our hardness proofs are indeed capturing (a lower bound on) real-world computational complexity.
> > >     * Average-case analysis often proceeds (by necessity) by making implausible assumptions about the distributions.
> > >         * When not enough knowledge is available to model real-world distributions, as is the case here, worst-case analysis is the proper tool as it better reflects current knowledge.
> > >         * This is in line with the remarks by Reviewer 19hk, who correctly summarizes the rationale to interpret our complexity results; namely, as a “guiding light as to what problems are tractable and might be solvable _without requiring any additional assumptions_” (emphasis ours).
> > >         * Consequently, worst-case analysis is adopted in current efforts to understand model interpretability (e.g., Bassan et al., 2024, ICML; Barcelo et al., 2020, NeurIPS), and can help us begin to understand possible gaps between formal and real-world problems (Hemaaspandra and Williams, 2012).
> > >     * Still, average-case analysis will become appropriate in due time, and therefore we discuss it now as a future direction together with potential experimental work that might help gather the necessary knowledge to make plausible distributional assumptions.

---

> > > > ### Author Response · Authors · 2024-11-18
> > > > **Response to Reviewer YPDD (5)**
> > > >
> > > > * __Response to Question 1 (single inputs vs sets of inputs):__
> > > >     * Thank you for this suggestion to strenghten our results.
> > > >         * As it turns out, we can readily use our existing proofs and this suggestion to highlight the scope of our computational problems and the results we obtain.
> > > >     * For most problems, we model both local and global variants (not just Problem 8) and report all results in Table 4. (Note some definitions of problems are local, and others are global, following the procedures in section 3.1 for how to construct problem variants. But we effectively study both variants for all problems where it is applicable).
> > > >         * Crucially, all local versions can be easily transformed to local “dataset” versions, as exemplified by the Circuit Patching problems (see also Gnostic Neuron).
> > > >             * These local dataset versions have the same complexity as the single-input local versions.
> > > >             * That is, hardness and inapproximabillity hold for these versions as well.
> > > >             * So, indeed, the stronger result the reviewer suggests can already be obtained but we had neglected to highlight it.
> > > >                 * We originally defined local circuits as covering a “restricted set of inputs” (Line 172), and gave different variants for the various problems, but we now make this more explicit in the text when we describe how to generate query variants.

---

> > > > > ### Author Response · Authors · 2024-11-18
> > > > > **Response to Reviewer YPDD (6)**
> > > > >
> > > > > * __Response to Question 2.1 (takeaway for interpretability in practice):__
> > > > >     * Our takeaway from our results is very much in line with the summaries of our work and descriptions of its strengths provided by Reviewer 19hk, Reviewer ZH6m, and Reviewer 8wLc.
> > > > >     * We would add the following (but see the responses to later questions for ellaboration).
> > > > >         * Most circuit discovery papers evaluate simple toy tasks — behaviors that can be elicited from small LMs.
> > > > >         * Those that do it on large models and more complex tasks are the exception, and none of them obtain pristine results but rather mixed outcomes that might be promising but are hard to interpret without the theoretical results we provide here.
> > > > >         * The crucial issue with these promising empirical results is we don’t know if these approaches can scale to larger models, more complex tasks, and real-world behaviors.
> > > > >         * Here our results are informative of the restrictions that can be placed on circuit queries to make them tractable for large models and complex tasks without any further assumptions.
> > > > >         * We now elaborate on these high-level takeaways in the last section.

---

> > > > > > ### Author Response · Authors · 2024-11-18
> > > > > > **Response to Reviewer YPDD (7)**
> > > > > >
> > > > > > * __Response to Question 2.2 (interpretability essentially intractable?):__
> > > > > >     * No, we do not think real-world interpretability is essentially intractable. We now clarify this in the manuscript.
> > > > > >     * As our exploration of the current complexity landscape shows, reasonable relaxations, restrictions and problem variants can yield tractable queries that yield circuits with useful properties.
> > > > > >         * Continuing this kind of exploration, with feedback from the experimental community, is likely to lead to tractable queries and algorithms to answer them.
> > > > > >     * Note, however, that empirical evidence supports this conclusion, as it currently does not support the claim that it is typically tractable in the general case (see e.g., interpretability illusions, lack of minimality, lack of faithfulness, etc).

---

> > > > > > > ### Author Response · Authors · 2024-11-18
> > > > > > > **Response to Reviewer YPDD (8)**
> > > > > > >
> > > > > > > * __Response to Question 2.3 (approaches to interpretability given hardness):__
> > > > > > >     * Thank you for encouraging to discuss potential avenues.
> > > > > > >     * We see many viable strategies to deal with the intractability of our more ambitious computational problems and to leverage the tractability of more modest ones.
> > > > > > >         * (i) Faced with an intractable query, we can investigate which parameters of the problem (network characteristics, circuit characteristics, etc.) might be responsible for the core hardness of the general problem. If these problematic parameters can be kept small in practice (e.g., keeping depth constant while increasing width, or viceversa), this can yield a fixed-parameter tractable query which can be answered efficiently in practice. We already explored some of these parameters, but many more could be explored as any aspect of the problem can be parameterized. For this, a close dialogue between theorists and experimentalists will be crucial, as often empirical regularities suggest which parameters might be fruitful to explore theoretically, and experiments can test whether theoretically conjectured parameters are or can be kept small in practice.
> > > > > > >         * (ii) Generating altogether different circuit query variants is another way of making interpretability feasible. Our formalization of quasi-minimal circuit problems illustrates the search for viable algorithmic options with examples of tractable problems for mechanistic interpretability.
> > > > > > >             * When the use case of interpretability efforts is well defined, efficient queries that return circuits with useful affordances for the use case can be designed. Some circuits (e.g., quasi-minimal circuits, but likely many other types) might mimic the affordances for prediction/control that ideal circuits have, while shedding the intractability that plagues the latter.
> > > > > > >         * (iii) It is in principle possible that real-world trained neural networks possess an internal structure that is benevolent to general (ideal) circuit queries. The empirical evidence available, however, speaks against this possibility. In any case, it will always be important to characterize any ‘benevolent structure’ in the problems such that we can leverage it explicitly to desgin algorithms with useful guarantees.
> > > > > > >             * One example scenario where real-world statistics might act as mitigating forces with respect to computational hardness (of the general problems) is the case of high redundancy (related to our Circuit Robustness problem).
> > > > > > >                 * Redundancy can in some sense make circuit finding easier (as solutions are more abundant), but the benefit comes at a cost for interpretability through introducing identifiability issues.
> > > > > > >                 * As circuits supporting a particular behavior are more numerous (i.e., there is more redundancy), it might get easier to find them with heuristics, but since they are more numerous, they potentially represent different explanations, which leads to the issue of identifiability (Annonymous Submission10546; Open Review, ICLR2025).
> > > > > > >         * Overall, there are many possible paths to attempt to deal with the intractability of general problems in practice. Theoretical and experimental efforts can tackle these challenges better together, and we hope our work is a useful first step in reaching across from the theoretical side.
> > > > > > >     * We add all these considerations to the Implications section.

---

> > > > > > > > ### Author Response · Authors · 2024-11-18
> > > > > > > > **Response to Reviewer YPDD (9) - Closing remark and References**
> > > > > > > >
> > > > > > > > * __Note:__ we believe we have addressed all weaknesses and concerns. Please consider raising the score if you believe we have done so satisfactorily, and we welcome requests for further changes.
> > > > > > > >
> > > > > > > > __References__
> > > > > > > >
> > > > > > > > Hemaspaandra, L. A., & Williams, R. (2012). An Atypical Survey of Typical-Case Heuristic Algorithms (No. arXiv:1210.8099). arXiv. http://arxiv.org/abs/1210.8099
> > > > > > > >
> > > > > > > > Barceló, P., Monet, M., Pérez, J., & Subercaseaux, B. (2020). Model Interpretability through the lens of Computational Complexity. Advances in Neural Information Processing Systems, 33, 15487–15498. https://proceedings.nips.cc/paper_files/paper/2020/hash/b1adda14824f50ef24ff1c05bb66faf3-Abstract.html
> > > > > > > >
> > > > > > > > Bassan, S., Amir, G., & Katz, G. (2024). Local vs. Global Interpretability: A Computational Complexity Perspective. Proceedings of the 41st International Conference on Machine Learning, 3133–3167. https://proceedings.mlr.press/v235/bassan24a.html
> > > > > > > >
> > > > > > > > Everything, Everywhere, All at Once: Is Mechanistic Interpretability Identifiable? (2024, October 4). The Thirteenth International Conference on Learning Representations. https://openreview.net/forum?id=5IWJBStfU7&referrer=%5BReviewers%20Console%5D(%2Fgroup%3Fid%3DICLR.cc%2F2025%2FConference%2FReviewers%23assigned-submissions)
> > > > > > > >
> > > > > > > > Hitchcock, J. M., & Pavan, A. (n.d.). On the NP-Completeness of the Minimum Circuit Size Problem.

---

> > > > > > > > > ### Comment · Reviewer_YPDD · 2024-11-26
> > > > > > > > >
> > > > > > > > > Thanks for the detailed responses. I have raised my score.

---

### Official Review · Reviewer_8wLc · 2024-10-31

**Soundness:** 3
**Presentation:** 3
**Contribution:** 3
**Rating:** 8
**Confidence:** 4

**Summary:**

This paper presents a framework for examining the computational complexity of generating various types of "interpretability queries" within the field of mechanistic interpretability. Unlike previous work that explored the complexity of post-hoc explanations for different ML models, primarily focusing on explanations in the input domain, this paper extends these findings by providing similar forms of explanations (specifically for neural networks) at the neural level rather than the input level. The paper also introduces complexity results for different mechanistic interpretability queries, demonstrating the intractability of some queries while proving that others are tractable, especially under certain relaxations.

**Strengths:**

The paper's primary strength lies in bridging two distinct areas of interest: (1) the formal analysis of the computational complexity involved in generating various forms of explanations (as explored in [1] and [2]), which often fall under the sub-domain known as "formal explainable AI," and (2) the more contemporary field of mechanistic interpretability. While prior work has proposed using the computational complexity of post-hoc explanations to shed light on interpretability (e.g., [1], [2]), this paper extends that concept by linking it to mechanistic interpretability queries. I believe this approach has the potential to inspire further research in both domains, facilitating a more rigorous analysis of mechanistic interpretability, an area that is currently somewhat underexplored. Additionally, this work offers a thorough categorization of different mechanistic interpretability queries, providing a solid foundation for future studies.

[1] Model Interpretability through the Lens of Computational Complexity (Barcelo et. al, Neurips 2020)

[2] Local vs. Global Interpretability: A Computational Complexity Perspective (Bassan et. al, ICML 2024)

**Weaknesses:**

1. The paper does not clearly discuss its limitations. I would be interested in hearing from the authors what they consider to be the main limitations of their work. While the paper focuses on neuron-level explanations, one evident (potential) limitation is the lack of clarity around the definition of the input space domain. From my understanding, it appears to be binary, but is that correct? Or is the input space discrete or continuous? Given that this research centers on neural networks, rather than simpler models where binary or discrete domains are more common, it would be reasonable to also consider more complex domains, such as continuous ones. Additionally, transitioning to continuous input settings could have significant implications for the complexity of the problem. This should (at the very least) be discussed by the authors.

2. Many of the proofs presented in the paper appear to be relatively straightforward, and I’m not fully convinced of the *technical* significance of this work. It would be beneficial to clearly indicate when a proof is trivial versus when it is non-trivial. Additionally, specifying when a result confirms well-known folklore assumptions, and when it is genuinely surprising would enhance clarity. This distinction was not always evident in the paper.


3. Some of the "folklore" concepts that this paper seeks to rigorously prove are somewhat unclear. For instance, the explanation of this idea could be improved:

This result establishes a formal separation
 between local and global query complexity that partly explains ‘interpretability illusions’ (Friedman
 etal.,2024;Yuetal.,2024a) arising in practice due to local but not global faithfulness.

the meaning of "interpretability illusions" is not clearly defined.

4. I'm not entirely convinced by the rigid distinction between "interpretability" as being neuron-level and "explainability" as being input-level. Naturally, the efficiency of providing explanations at the input level is also closely tied to the model's interpretability, not just its neuron-level explanations. Perhaps a more appropriate way to differentiate between the outcomes here and those in previous studies is to describe them as explanations or interpretations at the "input level" versus the "neuron level," both of which contribute to the model’s overall interpretability.

5. The main table is excellent and offers a clear summary of the results. However, a more concise version of this table might be preferable for the main text, with the larger, detailed version included in the appendix.

6. The paper suggests that problems “contained” within NP can be efficiently addressed using SAT solvers. However, this is not always the case—when it comes to neural networks, especially for continuous domains, neural network verification models often serve as more appropriate optimization tools. These are usually not based on SAT solvers, but rather on SMT-based solvers, or branch-and-bound methods.

**Questions:**

1. Over what type of input domain are the results defined (binary, discrete, or continuous)?

2. What non-trivial proofs are presented in the paper?

---

> ### Author Response · Authors · 2024-11-18
> **Response to Reviewer 8wLc (1)**
>
> Thank you for a great summary and description of strengths. We agree with this characterization.
>
> Below we reply to each comment and detail our changes to the manuscript to address each issue.
>
> * __Response to Weakness 1 and Question 1 (discussion of limitations; input properties):__
>     * Thank you for raising these issues. We think it’s important to discuss them in the paper. We have now made room to do so more thoroughly as follows.
>         * (i) It is common for analyses to start with simple models such as ours (see e.g., Barcelo et al., 2020, NeurIPS; Bassan et al., 2024, ICML) and only then progress to more complex models.
>             * Given that intractability results for simplified models also apply to models that have such simplified models as special cases (as pointed out by Reviewer ZH6m), simplicity is a useful guideline in the initial stage of computational complexity analysis (recall our work is the first to tackle these problems, as noted by Reviewer ZH6m).
>         * (ii) Continuous input/output does not necessarily imply easier problem variants.
>             * Sometimes it does, as in Linear Programming (PTIME; Karmarkar, 1984) vs 0-1 Integer Programming (NP-complete; Garey and Johnson, 1979), and sometimes it does not, as in the case of Euclidean Steiner Tree (NP-hard, not in NP for technical reasons; Garey and Johnson, 1979) vs Rectilinear Steiner Tree (NP-complete; Garey and Johnson, 1979).
>             * This has to be established on a case-by-case basis with proof, as intutition and analogy are not reliable guides.
>         * (iii) We are operating within a standard discrete setting, established previously (and considered to be appropriate and relevant) by Barcelo et al. (2020; NeurIPS) and Bassan et al. (2024; ICML), among others.
>             *  This makes our results comparable to the literature on the complexity of explainability (a central contribution of our paper, as observed in the Reviewer’s description of our paper’s strengths above).
>         * (iv) Examining our proofs we can give insight into the answer to these concerns. All of our proofs for local problem variants assume a particular input vector I, be it the all-0 or all-1 vector. Note that we can simulate these vectors by having zero weights on the input lines and putting appropriate 0 and 1 biases on the input neurons (a technique developed and used in our later-derived proofs but readily applicable to earlier ones). This causes the input to be “ignored”, which renders our proofs correct under both integer and continuous inputs. Note this construction is in line with previous work such as Barcelo et al. (2020; NeurIPS) and Bassan et al. (2024; ICML).
>             * Importantly, this highlights that the characteristics of the input are not important but rather there is a combinatorial core at the center of our circuit problems; namely, the selection of a subcircuit from exponential number of subcircuits. This combinatorial core can, if not tamed by appropriate restrictions on network and input structure, give rise to non-polynomial worst-case algorithmic complexity.
>     * This insight delivered by our proof constructions is now also included in the text.
>     * (v) These remarks aside, one can reasonably conjecture that considering more complex domains in the definitions of the problems is not likely to make the problems easier to solve. Our results likely represent a lower bound on the complexity of the real-world interpretability problems.
>         * However, such extended analyses are possible, and it is our hope that at least some of the proof techniques we have developed here will be of use in this endeavor, which will be part of our future work.

---

> > ### Author Response · Authors · 2024-11-18
> > **Response to Reviewer 8wLc (2)**
> >
> > * __Response to Weakness 2.1 and Question 2 (technical significance):__
> >     * Thank you for encouraging us to clarify the technical significance of our work.
> >         * There can be technical significance in the introduction of new mathematics and technical significance in deploying existing math to answer important questions.
> >         * As is the case for other work in this area (which might be called “Applied Complexity Theory”; Bassan et al., 2024, ICML; Barcelo et al., 2020, NeurIPS), we are not aiming at developing new mathematics but rather deploying mathematical tools to answer important questions that connect to applications.
> >             * Part of the techanical challenge we take up is to formalize problems of practical interest in simple (and if possible, elegant) ways that are readily understandable, and to prove their complexity properties efficiently (i.e., obtaining a one to many relation between proof constructions and meaningful results). This allows us to gain insights into the sources of complexity of problems, an investigation where the difficulty/complexity/intricacy of proofs are a liability.
> >     * This is an important clarification about the approach that this “Applied Complexity Theory” field  takes, so we add it to the preamble of our Appendix before presenting theorems and proofs.
> >     * Moreover, presenting to a reader the clearest and (perhaps deceptively) simplest path to a result is something we put effort in, as indeed the first attempt at proofs can be overly complicated.
> >     * All of this said, we now indicate where our proofs might contain interesting constructions whose value might go beyond the specific result and its connection to real-world problems.
> >         * Additionally, note that for some problems we provide alternative constructions (e.g., reductions from Clique and Vertex Cover to Sufficient Circuit) that might be useful for different purposes.
> >         * Some of our constructions for circuit queries (inner interpretability) have proven useful to tighten existing results for input queries (explainability; Problem 10).
> >         * Some constructions allow us to go beyond classical complexity to explore parameters of the problems through fixed-parameter complexity. These constructions might be useful to analyze future parameterizations of problems for MLPs and other architectures.
> >     * More generally, much of the technical significance comes from establishing a formal framework that is useful beyond the present questions.
> >         * Both the problem definitions and the proof constructions set up a formal framework to analyze the inner interpretability of existing and future models in a way that aligns well with ongoing efforts (Bassan et al., 2024, ICML; Barcelo et al., 2020, NeurIPS).

---

> > > ### Author Response · Authors · 2024-11-18
> > > **Response to Reviewer 8wLc (3)**
> > >
> > > * __Response to Weakness 2.2 (when results confirm/contradict folklore assumptions):__
> > >     * Thank you for this suggestion. We will add these clarifications at each step of the reporting of the results.
> > >     * We take this opportunity to also clarify that it is important to signal not only whether our results contradict/confirm folklore intuitions but also to signal whether these intuitions were arrived at for the wrong reasons, even when they might be nominally correct.
> > >     * Much of the literature that develops heuristics often assumes intractability without evidence, for example, by alluding to exponential search spaces. But exponential search spaces are not a sufficient condition for intractability.
> > >         * For instance, the Minimum Spanning Tree problem has an exponential search space.
> > >             * However, there is enough structure in this space such that it can be exploited to get a minimum spanning tree in quadratic time.
> > >             * Another example is obtaining an optimal segmentation of a sequence given the value of the possible segments.
> > >             * A more directly relevant example is our Quasi-Minimal Circuit problems, which is a scenario where the typical reasoning in the literature would clearly go wrong. Faced with the need to design algorithms to solve a Quasi-Minimal Circuit query, interpretability researchers could perhaps observe that the search space is exponential. As is typically done and justified in the literature, this problem would be categorized as “intractable” and this would prompt the development of heuristics with no proven guarantees (we need not single out individual work here so we omit the citations). Given our results (i.e., tractability of Quasi-minimal sufficient circuit and circuit patching, and algorithms to compute them that can be combined with gradient techniques, logit lens techniques, etc.), one can see that jumping to intractability conclusions as above would miss the opportunity to use existing tractable techniques and to leverage the provable guarantees that they provide.
> > >         * Our work hopefully contributes to move the field forward by encouraging researchers to consider restrictions to the problems and additional assumptions that might yield tractable procedures with known properties, and to gain clarity on what problems current heuristics are well equipped to solve.
> > >     * Our results might contradict folklore assumptions also in this latter sense. Often intractability is assumed (as above, for the wrong reasons) and heuristics are developed that are again expected intuitively to be good approximations. Some of our results show these heuristics are not likely to be good approximations to the general problems, but only to more restricted problems that are nevertheless worth exploring (e.g., our quasi-minimal formulations).

---

> > > > ### Author Response · Authors · 2024-11-18
> > > > **Response to Reviewer 8wLc (4)**
> > > >
> > > > * __Response to Weakness 3 (explanation of interpretability illusions):__
> > > >     * Thanks for pointing this out. We now explain ‘interpretability illusions’ and how our results contribute to make sense of them in the manuscript as follows.
> > > >         * Interpretability illusions (Friedman et al., 2024) come about when an interpretability abstraction (e.g., a circuit) seems faithful to its target (e.g., model behavior) by some criterion (e.g., ‘local’ tests on a selected dataset), but actually lacks faithfulness in the way it generalizes to other criteria (e.g., tests of its ‘global’ behavior outside the original distribution).
> > > >         * Our results establish a formal separation between local (NP-complete) and global ($\Sigma^p_2$-complete) Sufficient Circuit queries, which we conjecture might hold for the other queries we investigate.
> > > >             * This result puts global queries (whose target is the behavior for a single input or a finite set of inputs) higher in the complexity hierarchy (i.e., the ‘polynomial hiearchy’) than local queries (whose target is the behavior for all possible inputs).
> > > >             * This formal hardness parallels the empirical outcomes characterized as ‘interpretability illusions’ and might be part of their theoretical underpinning.

---

> > > > > ### Author Response · Authors · 2024-11-18
> > > > > **Response to Reviewer 8wLc (5)**
> > > > >
> > > > > * __Response to Weakness 4 (distinction between explainability and interpretability):__
> > > > >     * We agree with the reviewer’s characterization and proposal.
> > > > >     * We inherit this terminology from the literature (we cite several papers using it), but we agree with the reviewer that it is not the best naming scheme.
> > > > >         * This is why when we refer to the queries in a more technical way, we use “input queries” to refer to computational problems in work on complexity of ‘explainability’ and ‘circuit queries’ to refer to those in the field of inner interpretability (specifically, circuit discovery).
> > > > >         * We make no claims as to whether one or the other query relates more to intuitive ideas of explanation or interpretation. Rather, we use these words as pointers to the literature that practitioners are more familiar with.
> > > > >     * We add a footnote to clarify as above, adopting the conceptual organization suggested by the reviewer.

---

> > > > > > ### Author Response · Authors · 2024-11-18
> > > > > > **Response to Reviewer 8wLc (6)**
> > > > > >
> > > > > > * __Response to Weakness 5 (smaller results table):__
> > > > > >     *  Thanks for this suggestion. We have now compressed the table by 30% without loss of information nor drop in font size, and this helps make room for discussing the issues raised by all reviewers.
> > > > > >
> > > > > > * __Response to Weakness 6 (SAT vs SMT-based solvers):__
> > > > > >     * We agree that ‘problems in NP _can_ be efficiently addressed using SAT solvers’ is too strong a statement, as it is not necessarily true in all cases.
> > > > > >         * We now tone down these statements by emphasizing that SAT solvers _may_ be useful in some cases. Indeed, originally we meant to suggest membership in NP _might potentially_ open up the use of heuristics for NP-complete problems that are not available for other problems (de Haan and Szeider, 2017; 2019).
> > > > > >     * We would like to emphasize, for clarity, that the classification of problems beyond membership in NP also aims at identifying sources of added complexity, which is a separate motivation altogether for highlighting completeness for NP vs $\Sigma^p_2$.
> > > > > >
> > > > > > (Questions 1 and 2 are answered above).

---

> > > > > > > ### Author Response · Authors · 2024-11-18
> > > > > > > **Response to Reviewer 8wLc (7) - Closing remarks and References**
> > > > > > >
> > > > > > > * __Note:__ we believe we have addressed all weaknesses and concerns. Please consider raising the score if you believe we have done so satisfactorily, and we welcome requests for further changes.
> > > > > > >
> > > > > > > __References__
> > > > > > >
> > > > > > > Karmarkar, N. (1984). A new polynomial-time algorithm for linear programming. Combinatorica, 4(4), 373–395. https://doi.org/10.1007/BF02579150
> > > > > > >
> > > > > > > Garey, M. R., & Johnson, D. S. (1979). Computers and intractability: A guide to the theory of NP-completeness. W. H. Freeman.
> > > > > > >
> > > > > > > Friedman, D., Lampinen, A., Dixon, L., Chen, D., & Ghandeharioun, A. (2024). Interpretability Illusions in the Generalization of Simplified Models (No. arXiv:2312.03656). https://doi.org/10.48550/arXiv.2312.03656
> > > > > > >
> > > > > > > de Haan, R., & Szeider, S. (2017). Parameterized complexity classes beyond para-NP. Journal of Computer and System Sciences, 87, 16–57. https://doi.org/10.1016/j.jcss.2017.02.002
> > > > > > >
> > > > > > > de Haan, R., & Szeider, S. (2019). A Compendium of Parameterized Problems at Higher Levels of the Polynomial Hierarchy. Algorithms, 12(9), 188. https://doi.org/10.3390/a12090188
> > > > > > >
> > > > > > > Barceló, P., Monet, M., Pérez, J., & Subercaseaux, B. (2020). Model Interpretability through the lens of Computational Complexity. Advances in Neural Information Processing Systems, 33, 15487–15498. https://proceedings.nips.cc/paper_files/paper/2020/hash/b1adda14824f50ef24ff1c05bb66faf3-Abstract.html
> > > > > > >
> > > > > > > Bassan, S., Amir, G., & Katz, G. (2024). Local vs. Global Interpretability: A Computational Complexity Perspective. Proceedings of the 41st International Conference on Machine Learning, 3133–3167. https://proceedings.mlr.press/v235/bassan24a.html

---

> > > > > > > > ### Comment · Reviewer_8wLc · 2024-11-29
> > > > > > > >
> > > > > > > > Dear authors,
> > > > > > > >
> > > > > > > >
> > > > > > > > Thank you for your thorough response.
> > > > > > > >
> > > > > > > > I agree that starting with a "simplified input space" is a sensible approach. However, since the focus here is specifically on neural networks and mechanistic interpretability, where such domains are commonly used, I believe it's important to address this aspect explicitly. Given that the paper is intended for this community, it would be beneficial to include a clear discussion of this matter in the main text. When I initially read the paper, I couldn't find such a discussion, so I think it’s essential to state and elaborate on this point.
> > > > > > > >
> > > > > > > >
> > > > > > > > My initial concern was primarily about local queries, which are tractable under certain relaxations, but I feared they might become intractable in the continuous setting. I now realize this isn’t the case, which significantly reduces the weight of this concern. It was rather silly of me to assume otherwise - of course, strictly local cases wouldn’t depend on the input setting. As for the global setting, after reviewing the proofs further, I agree with the authors' suggestion that hardness results for discrete settings likely extend to the continuous ones. However, this isn't a major issue since these are intractability results regardless. I think it would be valuable for the paper to explicitly state that the complexity of the local queries are independent of the input domain.
> > > > > > > >
> > > > > > > >
> > > > > > > > Here are some additional suggestions I would recommend to the authors:
> > > > > > > >
> > > > > > > > 1. Include a discussion on the input space.
> > > > > > > >
> > > > > > > > 2. Consider breaking down the numerous theorems into smaller components such as claims, lemmas, and propositions. This will make it easier for readers to distinguish between significant implications and those with more straightforward or trivial proofs, a point I also suggested in my initial review.
> > > > > > > >
> > > > > > > > 3. If space allows, adding an illustration in the main text (or maybe only in the appendix) for concepts such as a sufficient circuit, a quasi-minimal circuit, and a subset-minimal circuit could be valuable for improving intuition and understanding.
> > > > > > > >
> > > > > > > > 4. (A very minor point): A quick glance at the references suggests that many are not up to date, missing details like the relevant conference or updated versions.
> > > > > > > >
> > > > > > > >
> > > > > > > > I will raise my score to an 8 because I believe the paper represents **an important step in formalizing mechanistic interpretability**. However, I have slightly reduced my confidence to a 4, as I did not have enough time to thoroughly review all the mathematical details of the proofs. That said, I still feel confident in recommending acceptance.
> > > > > > > >
> > > > > > > > Good luck with your acceptance process!

---

> > > > > > > > > ### Author Response · Authors · 2024-11-30
> > > > > > > > > **Response to final comment from Reviewer 8wLc**
> > > > > > > > >
> > > > > > > > > Many thanks for a very constructive review, and for updating the score.
> > > > > > > > >
> > > > > > > > > We agree and take onboard all suggestions as follows:
> > > > > > > > >
> > > > > > > > > 1. In the updated manuscript we have already included some discussion on the input space, and directed the reader to the appendix for the rest. But we will make an effort to move more of it to the main text, and add remarks reflecting the concerns in the reviewer’s comment.
> > > > > > > > > 2. We will look for ways to break theorems and proofs down where possible.
> > > > > > > > > 3. We will add schematics in the style of those currently in the appendix. It’s unlikely that they will fit in the main text but we will try. Otherwise, they will be included in the appendix before going on to theorems and proofs.
> > > > > > > > > 4. Thank you for catching this. We will go through the references and add any missing information.
> > > > > > > > >
> > > > > > > > > Thanks again for helping improve our work.

---

### Official Review · Reviewer_ZH6m · 2024-11-04

**Soundness:** 4
**Presentation:** 2
**Contribution:** 3
**Rating:** 8
**Confidence:** 4

**Summary:**

The authors frame a large number of circuit discovery problems in mechanistic interpretability as a set of formally defined queries about computational graphs.
They identify and prove the complexity classes for each of these queries. They also find the fixed-parameter complexity classes and the complexity of several types of approximations to the optimal algorithms.

**Strengths:**

This is the first formal characterization of the complexity of circuit discovery that I am aware of. It is very useful for the field to formalize problems such that they can be reasoned about with clarity. The formal characterization is also useful to clearly enumerate all the types of circuit queries that mechanistic interpretability researchers might want to perform to understand networks.

The paper is extremely thorough in covering a wide range of queries. The proofs that I read in detail (Appendix B on Minimum Locally Sufficient Circuits) appeared to be correct.

**Weaknesses:**

It is unclear to what extent these results should be major considerations for applied circuit discovery in real models. Heuristics such as gradient magnitude may be powerful enough to achieve close to optimal solutions with simple algorithms. The authors should include discussions of this to clarify.

The paper is primarily of use to researchers in mechanistic interpretability. Yet many researchers in the field will struggle to quickly understand much of the paper due to terse explanations of basic concepts in complexity theory. To some extent this is unavoidable as it would be arduous to include extensive tutorials on basic concepts from complexity, but a few well-chosen explanations could go a long way to bridge the gap for researchers who have some background in computer science. For example, more intuitive definitions of fixed parameter complexity, W[1]-hard and $\Sigma^{p}_{2}$-hard could be fairly concise and save many readers from needing to search for external resources.

Further, it would be useful to spell out more explicitly for less familiar readers how each proof connects to real world applications of circuit discovery. For example, since transformers contain MLPs, the proofs of complexities of queries on MLPs also apply to neuron level circuit discovery in transformers. Similarly, since binary vectors are a subset of the possible inputs to a real model, some proofs apply to models which are not constrained to binary inputs.

Since real models do not have binary activations, it is almost impossible for a circuit to achieve perfect faithfulness. In practice, we want our circuits to produce outputs that are sufficiently close to the full model output. But the problems considered by the authors ask for circuits that are perfectly faithful. So the authors should discuss whether their results apply in practice to realistic circuit discovery requirements.

Some smaller nitpicks:
- In Table 2: The notation “ “ is unclear because it looks like it could mean the same value as the previous row. It would be clearer to use a faded font or simply leave the box empty.
 - In Table 2: There is no explanation of "Coverage In" vs "Coverage Out".
 - I don’t understand why the coverage out of local circuits is empty.
 - In Appendix A.3 there is no explanation of function g(k).
 - In Appendix A.4.1 the variable I (instances) should be explained in the first paragraph.
 - In Appendix B, in the definition of MLSC, it’s unclear what the purpose of the constant c is.
 - In Appendix B, in Table 5, the third column is not labeled.
 - In Appendix B.2 make it clear which parts of proofs are being repeated from earlier parts.

**Questions:**

Some proofs rely on step functions being applied to neuron activations. To what extent do these proofs apply to neural networks with only ReLU activation functions? I believe that ReLU MLPs can approximate step functions, but not implement them exactly. Alternatively, could you extend your proofs such they don't rely on step functions?

---

> ### Author Response · Authors · 2024-11-18
> **Response to Reviewer ZH6m (1)**
>
> * Thank you for a great summary of our work and its strengths, and for acknowledging our contribution as the first of its kind. We agree with this characterization.
> * Below we reply to each comment and detail our changes to the manuscript to address each issue.
>
>
> * __Response to Weakness 1 (relevance to practice given potential of heuristics):__
>     * Thank you for bringing up the potentially good performance of heuristics in spite of the sometimes bleak theoretical results.
>     * We think this is an important issue to discuss in the paper, and our reasoning is as follows:
>         * (i) when heuristics have unproven properties, results such as ours can elucidate what heuristics are and are not likely capable of, and can help make sense of empirical results when these are inconclusive (as they often are, and as is the case so far in this field).
>         * (ii) the current performance of heuristics for circuit discovery is not at odds with our results.
>             * Experimental work with heuristics currently shows that in many cases the circuits found do not conform to the criteria defined in general circuit queries (e.g., minimality), or lack faithfulness (e.g., perform well on a dataset but do not generalize in the same manner as the network).
>             * This empirical evidence is in line with what we see theoretically, and our formalization of Quasi-minimal circuits, which we prove to be a tractable query, provides one possible theoretical description of the successes that are seen in practice.
>         * (iii) It is tempting to believe that heuristics can achieve good performance on hard problems. But we have principled reasons to think there is a hard limit to how good this performance can be. For instance, if heuristics for NP-hard problems could be as good as erring only on at most polynomially many instances with instance size, a whole set of highly implausible consequences would follow (Hemaspaandra and Williams, 2012).
>     * More generally, Reviewer 19hk (Strengths) correctly summarizes one rationale to interpret our computational complexity results for applied circuit discovery; namely, as a “guiding light as to what problems are tractable and might be solvable without requiring any additional assumptions, [which] might help guide practitioners search for tractable notions of interpretability.”
>         * Our results are useful to understand the scope of what heuristics can achieve unless they explicitly exploit problem restrictions, and to explore problem variants which some heuristics might be well equipped to solve.
>         * Heuristics used in practice such as gradient magnitude can be incorporated into algorithms such as the ones in our proofs of tractability of Quasi-Minimal Circuit (Problem 2 and Problem 7), yielding procedures with provable guarantees of yielding circuits with more modest properties which nevertheless retain useful affordances (Table 1).
>     * We now elaborate on all these points, as above, in the last section where these issues come up.

---

> > ### Author Response · Authors · 2024-11-18
> > **Response to Reviewer ZH6m (2)**
> >
> > * __Response to Weakness 2 (intuitive explanation of computational complexity concepts):__
> >     * Thank you for this comment. It is important for us that the concepts and techniques are explained to the community in an accessible way.
> >         * Therefore, we now expand the sections on classical and parameterized complexity, and approximation, with explanations of complexity classes, proof techniques, rationale for proving membership and hardness, the canonical interpretation of results.
> >         * This includes, among others, the following:
> >             * Parameterized complexity as a novel and more fine-grained analysis of the sources of complexity of the problems that can help identify aspects that make interpretability feasible in practice.
> >             * W[1]-hardness as roughly analogous to the more widely known NP-hardness but for parameterized complexity.
> >             * $\Sigma^p_2$-hardness as a level of complexity higher up in one  possible complexity hierarchy (the so-called Polynomial Hierarchy), one level above NP-hardness.

---

> > > ### Author Response · Authors · 2024-11-18
> > > **Response to Reviewer ZH6m (3)**
> > >
> > > * __Response to Weakness 3.1 (links between each result and real-world applications):__
> > >     * Thank you for this suggestion. We think it will improve the presentation of our results.
> > >     * As we present the problems and report the results, we add statements about the intuitive connection between the formal problems and the efforts and methods in the literature, and cite examples. For instance:
> > >         * In the Sufficient Circuit section, we connect it to notions of faithfulness in interpretability and link it to work by Wang et al., (2022).
> > >         * When presenting the Circuit Robustness problem, we add a conceptual link to Kramar et al. (2024) effort to diagnose nodes that are excluded during circuit discovery but can have an effect on the behavior (false positives).
> > >         * In the Necessary Circuit section, we now also connect it to the notion of circuit overlap, and link it to work by Merullo et al. (2024) who investigate circuits that overlap for a variety of tasks, and Hanna et al. (2024) that examines the relationship between overlap and faithfulness.
> > >         * In the Circuit Patching section we give a further example of work that uses this approach (Hanna et al., 2024).

---

> > > > ### Author Response · Authors · 2024-11-18
> > > > **Response to Reviewer ZH6m (4)**
> > > >
> > > > * __Response to Weakness 3.2 (highlight relevance to transfomers, and other input formats):__
> > > >     * Many thanks for highlighting these connections of our results to real-world applications of circuit discovery. We now discuss them in the implications section (and Appendix) as follows:
> > > >         * Our results are relevant beyond the specific formal context in which they were obtained.
> > > >             * For instance, since the transformer architecture contains MLPs, the complexity status of our circuit queries bears on neuron-level circuit discovery efforts in transfomers (Large Language/Vision/Audio models).
> > > >             * Likewise, in our constructions we sometimes simplify the network inputs to be binary vectors, but this is not a crucial assumption as it can be shown that the queries we examine contain a ‘hardness core’ due to the combinatorics of internal circuits independent of the input properties (see Appendix).
> > > >                 * APPENDIX: All of our proofs for local problem variants assume a particular input vector I, be it the all-0 or all-1 vector. Note that we can simulate these vectors by having zero weights on the input lines and putting appropriate 0 and 1 biases on the input neurons (a technique developed and used in our later-derived proofs but readily applicable to earlier ones). This causes the input to be “ignored”, which renders our proofs correct under both integer and continuous inputs. Note this construction is in line with previous work such as Barcelo et al. (2020; NeurIPS) and Bassan et al. (2024; ICML).
> > > >                     * Importantly, this highlights that the characteristics of the input are not important but rather there is a combinatorial "heart" beating at the center of our circuit problems; namely, the selection of a subcircuit from exponential number of subcircuits. This combinatorial core can, if not tamed by appropriate restrictions on network and input structure, give rise to non-polynomial worst-case algorithmic complexity.

---

> > > > > ### Author Response · Authors · 2024-11-18
> > > > > **Response to Reviewer ZH6m (5)**
> > > > >
> > > > > * __Response to Weakness 4 (faithfulness given binary output):__
> > > > >     * Thank you for raising this issue. We think it is important to discuss it in the manuscript and we reason about it as follows.
> > > > >     * Continuous input/output does not necessarily imply easier problem variants.
> > > > >         * Sometimes it does, as in Linear Programming (PTIME; Karmarkar, 1984) vs 0-1 Integer Programming (NP-complete; Garey and Johnson, 1979), and sometimes it does not, as in the case of Euclidean Steiner Tree (NP-hard, not in NP for technical reasons; Garey and Johnson, 1979) vs Rectilinear Steiner Tree (NP-complete; Garey and Johnson, 1979).
> > > > >         * This has to be established on a case-by-case basis with proof, as intutition and analogy are not reliable guides.
> > > > >     * We are operating within a standard discrete setting, established previously, and considered to be appropriate and relevant (Barcelo et al., 2020, NeurIPS; Bassan et al, 2024, ICML).
> > > > >         * This makes our results comparable to literature on the complexity of explainability (a central contribution of our paper, as correctly observed by the reviewer).
> > > > >     * Nevertheless, exploring, for instance, variants where the output is approximated is a great idea and we will both mention it as a future direction in the final section, and follow up on it in our future work.

---

> > > > > > ### Author Response · Authors · 2024-11-18
> > > > > > **Response to Reviewer ZH6m (6)**
> > > > > >
> > > > > > * __Responses to minor comments:__
> > > > > >     * Thank you for the careful reading and for pointing these out. We make the following changes to correct these errors:
> > > > > >         * (1) We now indicate empty strings with “__” in a light gray font as suggested.
> > > > > >         * (2) The “Coverage” variables belong to parts of the problem description that vary according to whether the requested circuit must have global or local coverage (e.g., sufficiency with respect to the behavior of the network). We now clarify this when we reference the table.
> > > > > >         * (3) The variable CoverageOUT of local circuit query variants is empty because for these versions, the input to the network is provided in the input to the computational problem, and therefore the description of the output of the problem needs only to reference this input ‘x’ in the sufficiency requirement of the circuit: C(x) = M(x). In contrast, for global versions, where no input to the network is provided in the input to the problem, we use a universal quantifier “\forall” to specify the global sufficiency requirement for the circuit. We now clarify this in the corresponding section.
> > > > > >         * (4) g() is an arbitrary function
> > > > > >         * (5) we added the definition of I in the first sentence
> > > > > >         * (6) c is not a constant but part of the notation for the parameter. This (admittedly cumbersome) notation is made transparent in Table 5 of the same section.
> > > > > >         * (7) We now label the column “Problem”.
> > > > > >         * (8) We originally opted to make each section self-contained to save the reader time visiting sections outside that of interest. We nevertheless now indicate which parts are similar, as requested.

---

> > > > > > > ### Author Response · Authors · 2024-11-18
> > > > > > > **Response to Reviewer ZH6m (7)**
> > > > > > >
> > > > > > > * __Response to Question 1 (step functions)__
> > > > > > >     * Thank you for this question, which allows us to highlight ways in which our results might be stronger than they seem (see Reviewer 19hk).
> > > > > > >     * Although some of our constructions use step functions (following the literature; Bassan et al., 2024, ICML; Barcelo et al., 2020, NeurIPS), note that for the following computational problems, our proofs use constructions which do not use step functions: Circuit Ablation; Circuit Patching; Circuit Robustness; Sufficient Circuit, the reduction from Clique does use a step function, but a second reduction from Vertex Cover we also provide does not; similarly for Minimal Circuit Counting.
> > > > > > >         * This suggests that the step function is not likely to be a significant source of complexity.
> > > > > > >     * We add these considerations to the manuscript.

---

> > > > > > > > ### Author Response · Authors · 2024-11-18
> > > > > > > > **Response to Reviewer ZH6m (8) - Closing remarks and References**
> > > > > > > >
> > > > > > > > * __Note:__ we believe we have addressed all weaknesses and concerns. Please consider raising the score if you believe we have done so satisfactorily, and we welcome requests for further changes..
> > > > > > > >
> > > > > > > > __References__
> > > > > > > >
> > > > > > > > Karmarkar, N. (1984). A new polynomial-time algorithm for linear programming. Combinatorica, 4(4), 373–395. https://doi.org/10.1007/BF02579150
> > > > > > > >
> > > > > > > > Garey, M. R., & Johnson, D. S. (1979). Computers and intractability: A guide to the theory of NP-completeness. W. H. Freeman.
> > > > > > > >
> > > > > > > > Friedman, D., Lampinen, A., Dixon, L., Chen, D., & Ghandeharioun, A. (2024). Interpretability Illusions in the Generalization of Simplified Models (No. arXiv:2312.03656). https://doi.org/10.48550/arXiv.2312.03656
> > > > > > > >
> > > > > > > > Barceló, P., Monet, M., Pérez, J., & Subercaseaux, B. (2020). Model Interpretability through the lens of Computational Complexity. Advances in Neural Information Processing Systems, 33, 15487–15498. https://proceedings.nips.cc/paper_files/paper/2020/hash/b1adda14824f50ef24ff1c05bb66faf3-Abstract.html
> > > > > > > >
> > > > > > > > Bassan, S., Amir, G., & Katz, G. (2024). Local vs. Global Interpretability: A Computational Complexity Perspective. Proceedings of the 41st International Conference on Machine Learning, 3133–3167. https://proceedings.mlr.press/v235/bassan24a.html
> > > > > > > >
> > > > > > > > Kramár, J., Lieberum, T., Shah, R., & Nanda, N. (2024). AtP*: An efficient and scalable method for localizing LLM behaviour to components (No. arXiv:2403.00745). arXiv. https://doi.org/10.48550/arXiv.2403.00745
> > > > > > > >
> > > > > > > > Merullo, J., Eickhoff, C., & Pavlick, E. (2024). Circuit Component Reuse Across Tasks in Transformer Language Models (No. arXiv:2310.08744). arXiv. https://doi.org/10.48550/arXiv.2310.08744
> > > > > > > >
> > > > > > > > Hanna, M., Pezzelle, S., & Belinkov, Y. (2024). Have Faith in Faithfulness: Going Beyond Circuit Overlap When Finding Model Mechanisms (No. arXiv:2403.17806). arXiv. https://doi.org/10.48550/arXiv.2403.17806
> > > > > > > >
> > > > > > > > Wang, K. R., Variengien, A., Conmy, A., Shlegeris, B., & Steinhardt, J. (2022, September 29). Interpretability in the Wild: A Circuit for Indirect Object Identification in GPT-2 Small. The Eleventh International Conference on Learning Representations. https://openreview.net/forum?id=NpsVSN6o4ul

---

> > > > > > > > > ### Comment · Reviewer_ZH6m · 2024-11-23
> > > > > > > > >
> > > > > > > > > Thanks to the authors for their insightful comments and for addressing feedback. I am keeping my score at 8 and raising the confidence to 4.

---

### Official Review · Reviewer_19hk · 2024-11-04

**Soundness:** 3
**Presentation:** 4
**Contribution:** 3
**Rating:** 8
**Confidence:** 3

**Summary:**

The paper studies the computational challenges surrounding the fields of circuit discovery and mechanistic interpretability. Achieving circuit discovery or accurate and precise interpretability at scale is a major empirical challenge for deep learning today. This paper sheds light on why that goal might be computationally out of reach by showing that many of the tools we use today end up having to solve computationally hard problems. Some of the specific problems the paper studies computational complexity of are:

1. Sufficient circuit – finding a small sub-circuit which reproduces the behavior of the network on a specific input or all inputs.
2. Gnostic neurons – a set of neurons with maximally distant activations of certain subsets of inputs we are trying to classify between.
3. Circuit Ablation and Clamping
4. Circuit patching
5. Circuit robustness
6. Sufficient reasons

For all these different problems, the paper studies the computational hardness of the general problem, fixed-parameter tractability, hardness of approximation (additive, multiplicative and probabilistic).
They show that while some of these problems are solvable in poly time (like gnostic neurons) most remain hard even to approximate. However, some are NP-complete and have reductions from 3SAT which leaves the promise of SAT-solver heuristic being applicable for these problems.
In addition, for some problems the results show a separation between local notion of interpretability (which captures a certain property only on a small set of inputs) and a global notion (which captures the property over the entire input range). Such separations are proposed as a reason for why local explanations of faithfulness in the past have been shown to be at times fickle.

**Strengths:**

-	The paper tackles an important conceptual question facing empirical deep learning today. Although its results are theoretical and computational complexity has faced criticism in the past of being overly pessimistic, it is nevertheless a stellar guiding light as to what problems are tractable and might be solvable without requiring any additional assumptions.
-	The paper provides some transformations/additional assumptions with which some of the problems to be solved start becoming tractable.
-	The paper is quite comprehensive in its coverage of different notions and tools used in mechanistic interpretability literature.
- Even if computational hardness for many of the problems being studied is to be expected by theorists, it is important to compile the results and clearly state the message as it might help guide practitioners search for tractable notions of interpretability.

**Weaknesses:**

-	Computational complexity of PAC learning MLPs is a well-studied problem by the learning theory community and has connections to many of the results you study here. A comparison to some of the hardness results already known there is in order.
-	The assumption of applying a step function on the output is quite stringent. It is known that being able to see a real-valued output rather than a Boolean output can make some learning problems on neural nets tractable [1]. The results of the paper would be stronger and more significant if they hold without the step function being applied at the end.
-	For most of the interpretability tasks, the notion of approximation considered to measure the quality of the sub-circuit found allows for some amount of flexibility in capturing the output behavior. This axis of approximation doesn’t seem to be addressed in your work (apologize if I missed it). Could the landscape change for any of the problems when considering such an approximation.


[1] Reliably Learning the ReLU in Polynomial Time. Goel et. al.

**Questions:**

-	What is the valid criticism of zero-ablation? (line 285)
-	Can you add a brief discussion on how your results relate to some computational hardness results known for PAC-learning variants of neural networks. In particular, I’m curious to understand if any of your results follow more directly from some of the hardness reductions considered in these works. Some example works (a non-exhaustive list) are cited below ([1-3]).
-	What happens if we relax the assumption of having a step function at the output layer? Would most of the problems studied continue to remain intractable. (note that now we can also allow for an approximation in the quality of the output. i.e. the sub-circuit doesn’t have to exactly match the behavior of the original circuit, it can be approximately close)


1. On the computational efficient of training neural nets. Livni et. al.
2. On the complexity of learning neural networks. Song et. al.
3. Learning deep neural networks is fixed-parameter tractable. Chen et. al.

---

> ### Author Response · Authors · 2024-11-18
> **Response to Reviewer 19hk (1)**
>
> Thank you for a great summary and description of strengths. We agree with this characterization.
>
> Below we reply to each comment and detail our changes to the manuscript to address each issue.
>
> * __Response to Weakness 1 and Question 2 (discussion of links to PAC-learning):__
>     * Thank you for encouraging us to discuss these potential links to PAC-learning.
>     * Circuit queries are conceptually closer to the goals and methods of interpretability practitioners than the typical PAC-learning setting.
>     * Furthermore, there are a number of conceptual and formal differences between the PAC-learning setting and our circuit queries:
>         * For instance, the former often involves learning the weights of a neural network that computes a function out of a large hypothesis class, whereas circuit queries ‘merely’ select a circuit out of a learned network.
>             * Also, the former imposes the constraints of the neural network architecture whereas circuits are not full architectures but are constrained by the structure of existing learned weights.
>         * These and other differences make it hard, as the reviewer remarks, to predict our results from existing results in PAC learning.
>             * For instance, one might intuitively speculate that between picking a suitable neural network from a large hypothesis class and finding an existing circuit within an already trained neural network (e.g., pruning an already suitable network), the latter should be easier.
>             * But our results, perhaps counterintuitively, demonstrate this intuition would be mistaken (unless restrictions such as quasi-minimality are imposed on the circuit queries).
>     * We think this potential connection to PAC-learning is worth speculating on in the Related Work section (including Livni et al., 2014; Song et al., 2017 Chen et al., 2020) and we look forward to exploring these links in our future work.

---

> > ### Author Response · Authors · 2024-11-18
> > **Response to Reviewer 19hk (2)**
> >
> > * __Response to Weakness 2 and Question 3 (step function and real-valued output):__
> >     * Thank you for suggesting ways to strengthen our results, which we are able to do as follows.
> >     * (i) Although some of our constructions use step functions (following the literature; Bassan et al., 2024, ICML; Barcelo et al., 2020, NeurIPS), note that for the following computational problems, our proofs use constructions which do not use step functions: Circuit Ablation; Circuit Patching; Circuit Robustness; Sufficient Circuit, the reduction from Clique does use a step function, but a second reduction from Vertex Cover we also provide does not; similarly for Minimal Circuit Counting.
> >         * This suggests that the step function is not likely to be a significant source of complexity.
> >     * (ii) More generally, continuous input/output does not necessarily imply easier problem variants.
> >         * Sometimes it does, as in the case of Linear Programming (PTIME; Karmarkar, 1984) versus 0-1 Integer Programming (NP-complete; Garey and Johnson, 1979), and sometimes it does not, as in the case of Euclidean Steiner Tree (NP-hard, not in NP for technical reasons; Garey and Johnson, 1979) versus Rectilinear Steiner Tree (NP-complete; Garey and Johnson, 1979).
> >         * As with other possible variants to interpretability queries, this has to be established on a case-by-case basis with proof, as intutition and analogy are not reliable guides.
> >     * We add these considerations to the discussion of our results as we agree with the Reviewer that they strengthen them.
> >     * (Please see also response to Weakness 3, which is relevant to considerations of real-valued output).

---

> > > ### Author Response · Authors · 2024-11-18
> > > **Response to Reviewer 19hk (3)**
> > >
> > > * __Response to Weakness 3 (approximation of the output):__
> > >     * Thank you for suggesting this approximation variant.
> > >     * Such variants could, in principle, be easier, but there is no particular reason to expect they will, for the reasons described in (ii) of the response to Weakness 2.
> > >     * But more importantly, as this is the first work to study the complexity of inner interpretability queries (as acknowledged by Reviewer ZH6m), we are operating within a standard discrete setting, established previously and considered to be appropriate and relevant to interpretability/explainability work (Bassan et al., 2024, ICML; Barcelo et al., 2020, NeurIPS).
> > >         *  This makes our results comparable to literature on the complexity of explainability (a central contribution of our paper, as observed by Reviewer 8wLc).
> > >     * In any case, the approximation variant described by the reviewer is a great suggestion and we will both discuss it as a future direction in the final section, and follow up on it in our future work.

---

> > > > ### Author Response · Authors · 2024-11-18
> > > > **Response to Reviewer 19hk (4)**
> > > >
> > > > * __Response to Question 1 (explanation of criticism of zero-ablation):__
> > > >     * The criticism of zero-ablation as a method (e.g., to find sufficient circuits) is that the patched value (zero) is somewhat arbitrary and therefore can mischaracterize the functioning of the neuron/circuit when operating in the context of the rest of the network during inference (Yu et al., 2024).
> > > >     * This gives rise to alternative methods such as activation patching with activation means or specific input activations.
> > > >     * This was briefly explained in the description of the circuit patching problem, but we now make this clearer earlier when we mention zero-ablation.
> > > >
> > > > (Replies to Questions 2 and 3 are given above)

---

> > > > > ### Author Response · Authors · 2024-11-18
> > > > > **Response to Reviewer 19hk (5) - Closing remarks and References**
> > > > >
> > > > > * __Note:__ we believe we have addressed all weaknesses and concerns. Please consider raising the score if you believe we have done so satisfactorily, and we welcome requests for further changes.
> > > > >
> > > > >
> > > > > __References__
> > > > >
> > > > > Chen, S., Klivans, A. R., & Meka, R. (2020). Learning Deep ReLU Networks Is Fixed-Parameter Tractable (No. arXiv:2009.13512). arXiv. https://doi.org/10.48550/arXiv.2009.13512
> > > > >
> > > > > Livni, R., Shalev-Shwartz, S., & Shamir, O. (2014). On the Computational Efficiency of Training Neural Networks. Advances in Neural Information Processing Systems, 27. https://papers.nips.cc/paper_files/paper/2014/hash/3a0772443a0739141292a5429b952fe6-Abstract.html
> > > > >
> > > > > Song, L., Vempala, S., Wilmes, J., & Xie, B. (2017). On the Complexity of Learning Neural Networks. Advances in Neural Information Processing Systems, 30. https://proceedings.neurips.cc/paper_files/paper/2017/hash/a78482ce76496fcf49085f2190e675b4-Abstract.html
> > > > >
> > > > > Barceló, P., Monet, M., Pérez, J., & Subercaseaux, B. (2020). Model Interpretability through the lens of Computational Complexity. Advances in Neural Information Processing Systems, 33, 15487–15498. https://proceedings.nips.cc/paper_files/paper/2020/hash/b1adda14824f50ef24ff1c05bb66faf3-Abstract.html
> > > > >
> > > > > Bassan, S., Amir, G., & Katz, G. (2024). Local vs. Global Interpretability: A Computational Complexity Perspective. Proceedings of the 41st International Conference on Machine Learning, 3133–3167. https://proceedings.mlr.press/v235/bassan24a.html
> > > > >
> > > > > Yu, L., Niu, J., Zhu, Z., & Penn, G. (2024). Functional Faithfulness in the Wild: Circuit Discovery with Differentiable Computation Graph Pruning (No. arXiv:2407.03779). arXiv. https://doi.org/10.48550/arXiv.2407.03779

---

> > > > > > ### Comment · Reviewer_19hk · 2024-11-25
> > > > > > **Thanks for your response**
> > > > > >
> > > > > > The authors' response has addressed some of the major concerns. I updated my score accordingly.

---

### Meta-Review · Area_Chair_33DA · 2024-12-23

**Metareview:**

This paper presents a framework for examining the computational complexity of generating various types of "interpretability queries" within the field of mechanistic interpretability. Unlike previous work that explored the complexity of post-hoc explanations for different ML models, primarily focusing on explanations in the input domain, this paper extends these findings by providing similar forms of explanations (specifically for neural networks) at the neural level rather than the input level. The paper also introduces complexity results for different mechanistic interpretability queries, demonstrating the intractability of some queries while proving that others are tractable, especially under certain relaxations.

All reviewers believe this paper made meaningful contributions. The AC agrees and thus recommends acceptance.

**Additional Comments On Reviewer Discussion:**

There were concerns about writing and the implications of the results. They are generally minor and do not change the decision of this paper.

---

### Decision · Program_Chairs · 2025-01-22

Accept (Spotlight)